# Vertical substitution strategy to enable cooperation between spin–orbit coupling and transition dipoles for organic phosphorescence

**Kikuya Hayashi, Riku Shimura, Ryo Miyashita & Shuzo Hirata** [✉]

The emission from organic molecules can be broadly classified as fluorescence or phosphorescence, and it is used in a wide range of applications from optoelectronics to bioimaging. The common method to enhance the fluorescence from organic chromophores is to introduce horizontal substituents into long conjugated structures. However, this concept does not allow the phosphorescence efficiency in the visible range to be increased to the fluorescence efficiency. Here we introduce a molecular design of conjugated molecules that effectively cooperate spin-orbit and transition dipoles. Increasing the through-space substitution of main-group elements perpendicular to the conjugated plane produces a cooperative effect of the spin–orbit and large transition dipoles of the conjugated plane to enhance the organic phosphorescence. This molecular design, which is different from the fluorescence, will increase the efficiency of the organic phosphorescence to the same level as the fluorescence at all visible wavelengths.

The emission from organic dyes is used in a wide range of applications, from bioimaging and other fields of life to lighting and photoelectron devices, such as electroluminescent devices[1–10], and it is essential for our lives. The photoluminescence from organic chromophores can be broadly classified as fluorescence without spin conversion of the quantum states or phosphorescence with spin conversion of the quantum states[11]. The molecular designs for obtaining highly efficient fluorescence include a conjugated π plane in which the donor and acceptor groups are horizontally attached to the rigid π plane, and these designs have been used in the design of laser dyes and two-photon absorption dyes[12–14]. Heavy metal complexes have been used to implement highly efficient phosphorescence across the entire visible light range in organic light-emitting diodes[15–17], and proposals for afterglow emitting imaging using the long lifetime of organic phosphorescence have rapidly increased in the past decade[18–22]. Although molecules with similar skeletons and substituent characteristics to those that exhibit fluorescence have been reported for organic phosphorescence, the phosphorescence is inefficient, especially in the

long-wavelength range[23–26]. If the fundamental design guidelines that differ between organic fluorescence and phosphorescence could be clarified, it would be greatly assist in reaching 100% yield of the organic phosphorescence in the visible to near-infrared range.

Here, we report molecular design concentrating on the organic phosphorescence process. Organic π backbones with horizontally introduced hetero-substituents and organic π backbones with vertically introduced hetero-substituents are synthesized. The introduction of multiple horizontal hetero-substituents induces inefficient phosphorescence, whereas the introduction of multiple vertical hetero-substituents leads to more efficient phosphorescence. Cooperative theoretical and experimental analysis confirms that the multiple vertical hetero-substituents can utilize the large transition dipole along the π plane, which allows for the cooperation of the large transition dipoles and spin–orbit coupling to enhance the phosphorescence radiation. Designed chromophores with enhanced red organic phosphorescence radiation allow high-resolution afterglow imaging using the emission colour and delay time, which contributes to break the

Department of Engineering Science, The University of Electro-Communications, 1-5-1 Chofugaoka, Chofu, Tokyo, Japan. [✉]e-mail: shuzohirata@uec.ac.jp

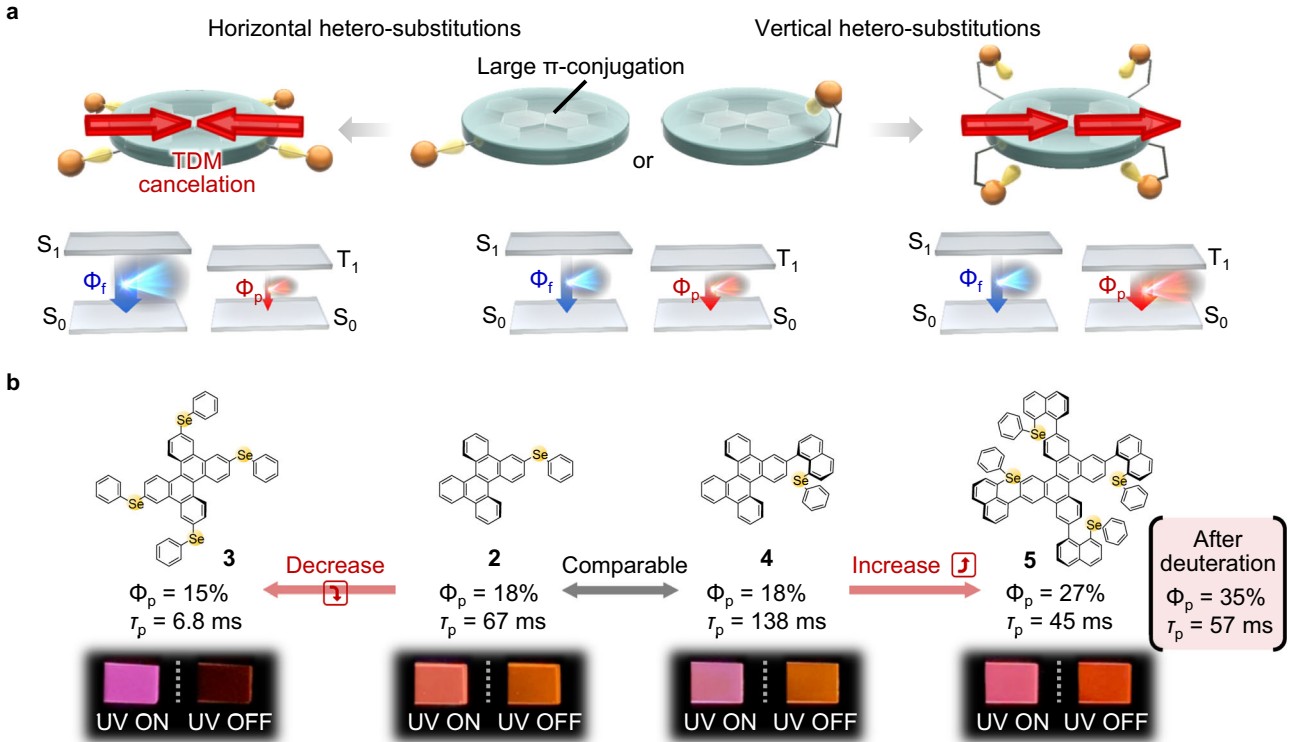

**Fig. 1 | Molecular design strategy and emission behaviour. a** Illustration of the concept of horizontal hetero-substitutions (left) and vertical hetero-substitutions (right) differentially changing the fluorescence quantum yield ($\Phi_f$) and phosphorescence quantum yield ($\Phi_p$). **b** Molecular structures of chromophores whose $\Phi_p$ and RTP lifetime ($\tau_p$) change depending on horizontal hetero-substitutions and vertical hetero-substitutions, and luminescence photographs of their dispersed solid thin films under excitation light and immediately after the excitation light is stopped.

limitation of the number of labels functioning in high-resolution and rapid bioimaging.

## Results

### Molecular design and emission behaviour

The expansion of the π-conjugation is a basic strategy to facilitate radiative transitions from excited states. One rational design strategy to extend the π conjugation is to introduce multiple hetero-substituents horizontally into an aromatic backbone. Multiple horizontal hetero-substituents generate a large transition dipole moment (TDM) associated with the singlet radiation process, thereby generally improving the fluorescence yield ($\Phi_f$). However, the TDM originating from the multiple substituents often cancels the TDM related to the triplet radiation process along the extended conjugation axis (Fig. 1a, left). When the large TDM originating from the extended π system can be used for the triplet radiation process, efficient room-temperature phosphorescence (RTP) can be obtained. Here, we report a molecular design that allows a large TDM associated with triplet radiation along the extended conjugation axis and efficient RTP by multiple vertical hetero-substitutions (Fig. 1a, right).

Dibenzo[*g,p*]chrysene (DBC) (**1**) derivatives with one or four phenylselane substituent (**2** and **3**) and one or four naphthalene–phenylselane substituent (**4** and **5**) were synthesized (Supplementary Figs. 1–14). Computational simulations suggested that the selenium atoms are placed vertically on DBC in **4** and **5** because of the steric hindrance between DBC and the naphthalene unit (Supplementary Fig. 15). Each chromophore was doped into *β*-estradiol at a concentration of 0.3 wt% to prepare an amorphous film (Supplementary Fig. 16). Soon after ceasing excitation, **2** and **4** showed red afterglow RTP with comparable brightness and RTP yield ($\Phi_p$) (Fig. 1b, middle). However, **3** showed almost no afterglow capability (Fig. 1b, left). In contrast, chromophore **5** showed the brightest afterglow and

highest $\Phi_p$ among chromophores **2**–**5** (Fig. 1b, right). The opposite behaviour of the RTP between **3** and **5** suggests that each substituent affects the phosphorescence process differently.

### Optical properties

To clarify the mechanism, the emission properties of **1**–**5** and deuterated **5** (**5d**) in amorphous *β*-estradiol (Fig. 2a) were investigated. Chromophores **1**–**5d** showed blue fluorescence and red RTP under excitation (Fig. 2b), but they showed only red RTP soon after ceasing excitation (Fig. 2c). Integration sphere measurements showed that the $\Phi_f$ values of **1**–**5d** were 20%, 1.1%, 1.7%, 1.2%, 1.0%, and 0.8%, respectively. The $\Phi_p$ values of **1**–**5d** were 2.9%, 18%, 15%, 18%, 27%, and 35%, respectively (Supplementary Figs. 17–19). These results confirm that increasing the number of horizontal hetero-substituents enhances $\Phi_f$ while reducing $\Phi_p$, whereas $\Phi_p$ is selectively enhanced with increasing number of vertical hetero-substituents. The RTP decay of **1**–**5d** approached a single exponential nature, and the average RTP lifetimes ($\tau_p$) of **1**–**5d** were 580, 67, 6.8, 138, 45, and 57 ms, respectively (Fig. 2d). Transient absorption measurements confirmed that the intersystem crossing yields ($\Phi_{isc}$) of **1**–**5d** were ≥87% (Supplementary Figs. 20–23 and Supplementary Tables 1 and 2). The comparable $\Phi_{isc}$ values indicate that the photophysical parameters after formation of the lowest triplet excited state ($T_1$) are responsible for the difference in $\Phi_p$.

Generally, $\Phi_p$ depends on the rate constant of the radiative transition from $T_1$ ($k_r^T$) and the rate constant of the nonradiative transition from $T_1$ ($k_{nr}^T$). $\Phi_p$ and $\tau_p$ are expressed as $\Phi_{isc}k_r^T\tau_p$ and $1/(k_r^T + k_{nr}^T)$, respectively. Using the relationship $\Phi_p = \Phi_{isc}k_r^T\tau_p$, the $k_r^T$ values of **1**–**5d** in amorphous *β*-estradiol were determined to be 0.063, 2.8, 23, 1.3, 6.1, and 6.3 s$^{-1}$, respectively (Supplementary Table 2). By substituting $\tau_p$ and $k_r^T$ into $\tau_p = 1/(k_r^T + k_{nr}^T)$, the $k_{nr}^T$ values of **1**–**5d** were determined to be 1.7, 12, 124, 5.9, 16, and 11 s$^{-1}$, respectively. The selective suppression of $k_{nr}^T$ from **5** to **5d** can be explained by a

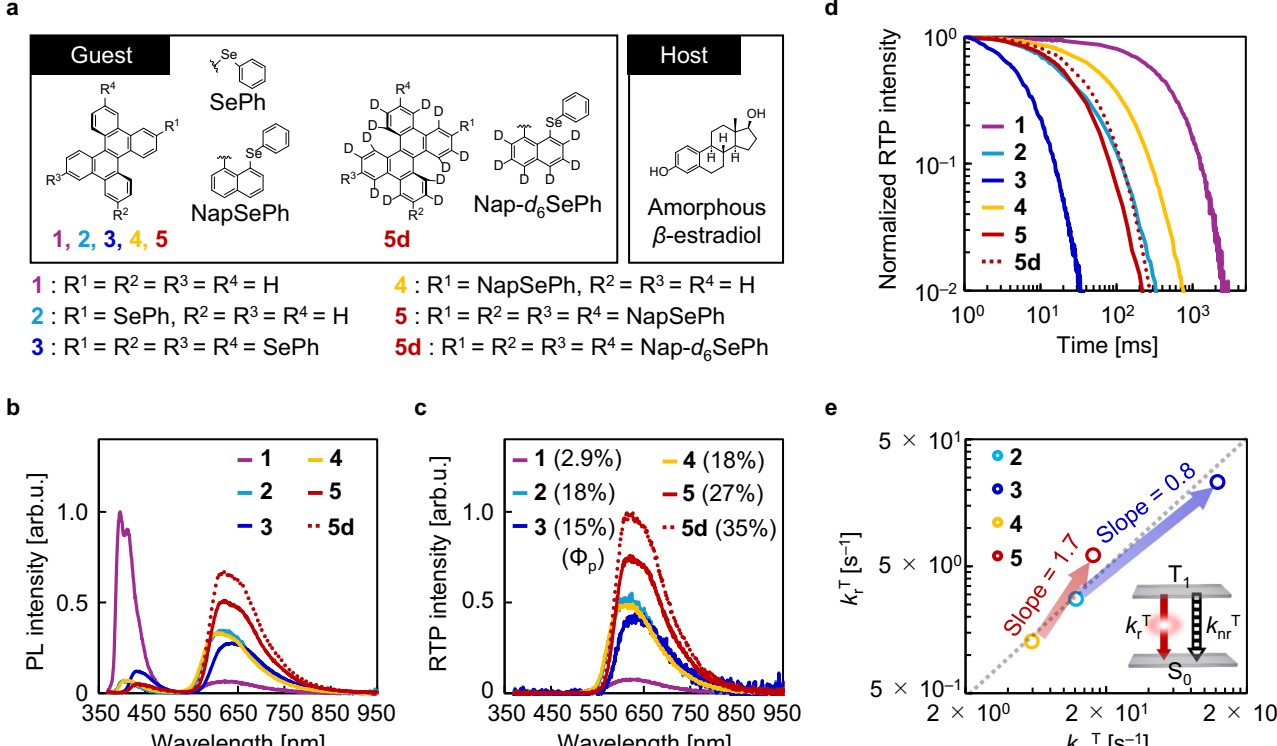

**Fig. 2 | Photophysical properties. a** Chemical structures of **1**–**5d** (left) and β-estradiol (right). **b** Emission spectra under excitation. **c** Room-temperature phosphorescence (RTP) spectra soon after ceasing excitation. The graph was constructed so that the area of the spectrum was proportional to the quantum yield of RTP ($\Phi_p$). **d** RTP decay characteristics. **e** Relationship between the rate constant of phosphorescence from the lowest triplet excited state ($k_r^T$) and the rate constant of nonradiative transition from the lowest triplet excited state ($k_{nr}^T$) among **2**–**5**. The blue and red arrows represent the magnitude of the increase in $k_r^T$ compared with that in $k_{nr}^T$ from **2** to **3** and from **4** to **5**, respectively.

common isotopic effect. However, the relationship between $k_r^T$ and $k_{nr}^T$ among **2**–**5** showed that $k_r^T$ and $k_{nr}^T$ greatly changed depending on the substituent type (Fig. 2e). From **2** to **3**, the slope of $k_r^T$ against $k_{nr}^T$ was less than one (Fig. 2e, blue arrow), confirming that the magnitude of the $k_r^T$ enhancement was smaller than that of the $k_{nr}^T$ enhancement. In contrast, from **4** to **5,** $k_r^T$ was enhanced by approximately 1.7 times compared with $k_{nr}^T$ (Fig. 2e, red arrow). Therefore, **5** has a mechanism that enhances $k_r^T$ relative to $k_{nr}^T$, causing efficient RTP of **5**.

### Effect of vertical hetero-substituents for $k_{nr}^T$ suppression

One reason for the enhancement of $k_r^T$ compared with $k_{nr}^T$ for vertical hetero-substitutions is that increasing the number of vertical hetero-substitutions makes $k_{nr}^T$ less likely to increase compared with common horizontal hetero-substitutions. $k_{nr}^T$ is generally expressed as[22,27]

$$k_{nr}^T = \frac{2\pi}{\hbar} \langle T_1 | \mathbf{H}_{SO} | S_0 \rangle^2 FC, \qquad (1)$$

where $\hbar$ is the Dirac constant, $\langle T_1 | \mathbf{H}_{SO} | S_0 \rangle$ is the spin–orbit coupling (SOC) between $T_1$ and the ground state ($S_0$), $\mathbf{H}_{SO}$ is the spin–orbit Hamiltonian, and $FC$ is the Franck–Condon factor between $T_1$ and $S_0$. $FC$ contains the energy gap law, where $k_{nr}^T$ increases as the phosphorescence is red-shifted[28]. Because the phosphorescence energies of **1**–**5** are comparable (Fig. 2c), the variation of the $k_{nr}^T$ values among **1**–**5** is expected to originate from differences in the $\langle T_1 | \mathbf{H}_{SO} | S_0 \rangle^2$ values. Indeed, quantum chemical calculations showed a good correlation between the optically determined $k_{nr}^T$ values and calculated $\langle T_1 | \mathbf{H}_{SO} | S_0 \rangle^2$ values for **1**–**5** (Supplementary Fig. 24a). Therefore, it is reasonable to attribute the difference in the $k_{nr}^T$ values between **3** and **5** to their respective $\langle T_1 | \mathbf{H}_{SO} | S_0 \rangle^2$ values. The first factor that changes $\langle T_1 | \mathbf{H}_{SO} | S_0 \rangle^2$ is the overlap of the orbitals.

$\langle T_1 | \mathbf{H}_{SO} | S_0 \rangle^2$ increases when the overlap of the orbitals involved in the two transitions is large (Fig. 3a(i)). $\mathbf{H}_{SO}$ is expressed as[6,29,30]

$$\mathbf{H}_{SO} = \alpha^2 \sum_A \sum_i \frac{Z_A}{r_{Ai}^3} \vec{\mathbf{l}}_{Ai} \cdot \vec{\mathbf{s}}_i, \qquad (2)$$

where $\alpha$ is the fine structure constant, $A$ indicates the nucleus, $i$ indicates the electron, $Z_A$ is the atomic number, $r$ is the electron–nucleus distance, $\vec{\mathbf{l}}$ is the orbital angular momentum, and $\vec{\mathbf{s}}$ is the spin angular momentum. The second factor that changes $\langle T_1 | \mathbf{H}_{SO} | S_0 \rangle^2$ is the distance between the heavy atoms and the overlap site of the orbitals involved in the $T_1$–$S_0$ transition. The $\frac{Z_A}{r_{Ai}^3}$ term in Eq. (2) means that $\langle T_1 | \mathbf{H}_{SO} | S_0 \rangle^2$ increases when the heavy atoms are closer to the overlap site of the orbitals involved in the $T_1$–$S_0$ transition (Fig. 3a(ii)). This is the contribution of the increase in $\langle T_1 | \mathbf{H}_{SO} | S_0 \rangle^2$ owing to the introduction of heavier atoms, that is, the heavy atom effect (HAE). The third factor that changes $\langle T_1 | \mathbf{H}_{SO} | S_0 \rangle^2$ is the relative orientation of the molecular orbitals. $\vec{\mathbf{l}}_{Ai} \vec{\mathbf{s}}_i$ in Eq. (2) means that $\langle T_1 | \mathbf{H}_{SO} | S_0 \rangle^2$ increases when the axes of the two orbitals involved in the overlapping region point in different directions (Fig. 3a(iii)). This enhancement of the SOC by this term is known as the El-Sayed rule[31]. When the two orbital axes are in the same direction, no torque is generated to flip the spin because the electron spin and magnetic moment are parallel (left of Fig. 3a(iii), Supplementary Note 5-2, and Supplementary Fig. 25). Conversely, when the axes of the two orbitals are in different directions, a torque that flips the spin is generated because the electron spin and magnetic moment are not parallel (Fig. 3a(iii), right)[32]. Thus, the three factors of the orbital overlap, HAE, and relative orientation of the molecular orbitals can be individually considered to discuss the difference in $\langle T_1 | \mathbf{H}_{SO} | S_0 \rangle^2 \propto k_{nr}^T$ between **3** and **5**.

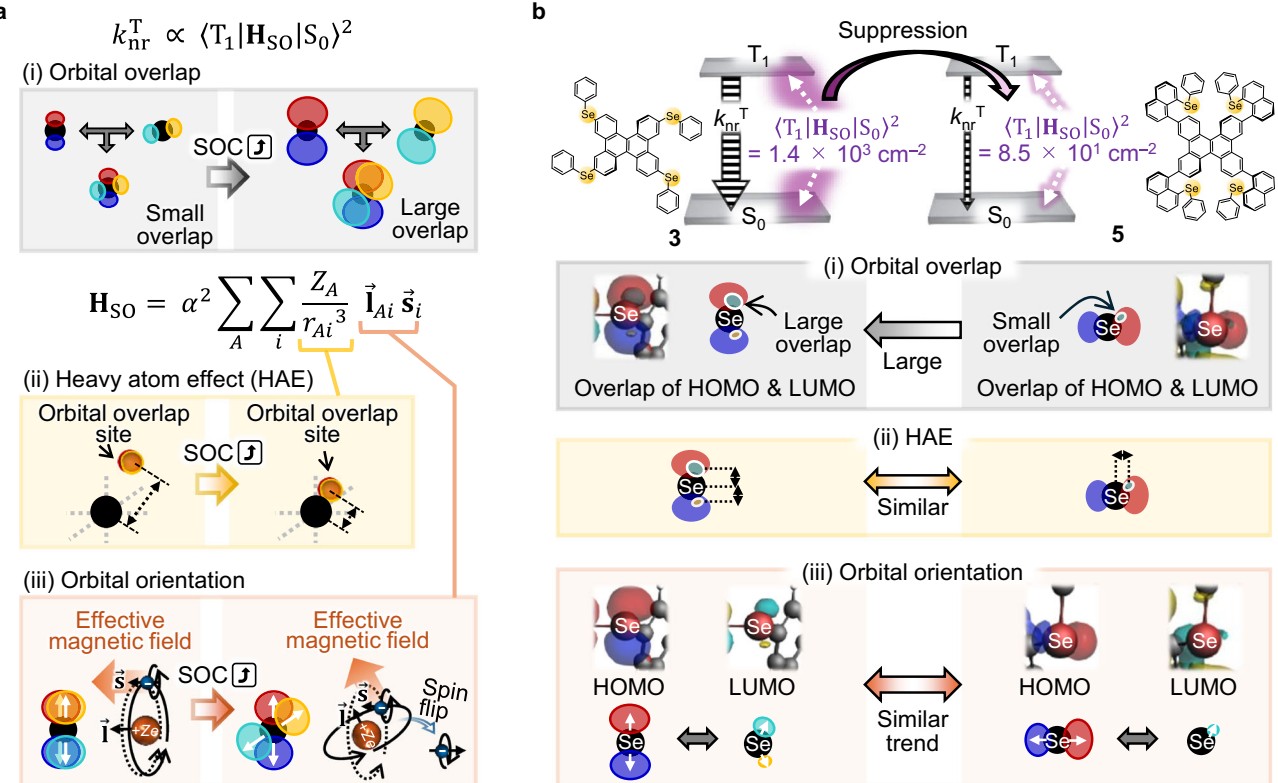

**Fig. 3 | Difference in the parameters affecting the nonradiative transition rate from the lowest triplet excited state ($k_{nr}^T$) between horizontal and vertical hetero-substituents. a** Schematic illustration of the parameters affecting the spin-orbit coupling between $T_1$ and $S_0$ ($< T_1 | \mathbf{H}_{SO} | S_0 >^2$): the (i) orbital overlap, (ii) heavy atom effect (HAE), and (iii) orbital orientation. **b** Molecular orbitals and schematic illustrations related to the (i) orbital overlap, (ii) HAE, and (ii) orbital

orientation to explain the smaller $k_{nr}^T$ of **5** than **3**. In (iii) of **a** and **b**, the white arrows represent the orbital orientation axes. The isovalues of the highest occupied molecular orbital (HOMO) of **3**, the lowest unoccupied molecular orbital (LUMO) of **3**, and the HOMO of **5** are 0.018. The isovalue of 0.0072 is used for the LUMO of **5**. Because the probability of the LUMO being near Se in **5** is extremely small, the isovalue is set low to reflect its very small presence.

The suppression of the increase in $k_{nr}^T$ with increasing number of vertical hetero-substitutions is due to the small overlap of the orbitals involved in the $T_1$–$S_0$ transition. Regarding the orbital overlap, the first factor that changes $< T_1 | \mathbf{H}_{SO} | S_0 >^2$, the $T_1$–$S_0$ transition is predominantly composed of the highest occupied molecular orbital (HOMO) and the lowest unoccupied molecular orbital (LUMO) for both **3** and **5** (Supplementary Fig. 26). Because the spatial overlap between the HOMO and LUMO on Se of **5** is much smaller than that of **3** (Fig. 3b(i)), the different amounts of orbital overlap contribute to the different $< T_1 | \mathbf{H}_{SO} | S_0 >^2$ values between **3** and **5**. Regarding the HAE, the second factor that changes $< T_1 | \mathbf{H}_{SO} | S_0 >^2$ for **3** and **5**, the Se atom is often considered to be the heavy atom source for the HAE, and it is widely used for organic phosphorescence materials[33–35]. However, the large difference in the $< T_1 | \mathbf{H}_{SO} | S_0 >^2$ values between **3** and **5** cannot be well explained by the HAE contribution because the distance between the HOMO and LUMO overlap sites and Se does not significantly differ between **3** and **5** (Fig. 3b(ii)). Regarding the orbital orientation factor, the third factor that changes $< T_1 | \mathbf{H}_{SO} | S_0 >^2$, the orbital orientation on the Se atoms differs between the HOMO and LUMO of **3** (Fig. 3b(iii), left). Because the HOMO and LUMO on the Se atoms also exhibit different orientations for **5** (Fig. 3b(iii), right), the orbital orientation factor does not well explain the different $< T_1 | \mathbf{H}_{SO} | S_0 >^2$ values between **3** and **5**. Therefore, the spatial overlap between the HOMO and LUMO on Se mainly causes the smaller $< T_1 | \mathbf{H}_{SO} | S_0 >^2$ of **5** than **3**, suppressing the increase of $k_{nr}^T$ for **5** compared with that for **3**.

### Effect of vertical hetero-substituents for $k_r^T$

Another reason for the enhancement of $k_r^T$ relative to $k_{nr}^T$ by vertical hetero-substitution is that an increase in the number of vertical hetero-substitutions tends to increase $k_r^T$ to a similar level as an increase in the number of horizontal hetero-substitutions. The equation for $k_r^T$ based on first-order perturbation theory is expressed as[6,29,30]

$$k_r^T = \frac{E_{T_1-S_0}^{\,3}}{3\varepsilon_0 \pi \hbar^4 c^3} \left| \mu_{T_1-S_0} \right|^2, \qquad (3)$$

$$\mu_{T_1-S_0} = \sum_{n \geq 1} \frac{\langle S_n | \mathbf{H}_{SO} | T_1 \rangle}{E_{S_n - T_1}} \mu_{S_n - S_0} + \sum_{m \geq 2} \frac{\langle T_m | \mathbf{H}_{SO} | S_0 \rangle}{E_{T_m - S_0}} \mu_{T_m - T_1}, \quad (4)$$

where $E_{T_1-S_0}$ is the energy gap between $T_1$ and $S_0$, $\varepsilon_0$ is the vacuum permittivity, c is the speed of light, $\mu_{T_1-S_0}$ is the TDM between $T_1$ and $S_0$, $< S_n | \mathbf{H}_{SO} | T_1 >$ is the SOC between the $n$th-order singlet excited state ($S_n$, $n \geq 1$) and $T_1$, $E_{S_n - T_1}$ is the energy gap between $S_n$ and $T_1$, $\mu_{S_n - S_0}$ is the TDM between $S_n$ and $S_0$, $< T_m | \mathbf{H}_{SO} | S_0 >$ is the SOC between the $m$th-order triplet excited state ($T_m$, $m \geq 2$) and $S_0$, $E_{T_m - S_0}$ is the energy gap between $T_m$ and $S_0$, and $\mu_{T_m - T_1}$ is the TDM between $T_m$ and $T_1$. Equation (3) indicates that the enhancement of $k_r^T$ becomes increasingly difficult as the phosphorescence is red-shifted. From Eq. (4), however, $k_r^T$ can still be facilitated by the cooperative contributions of $< S_n | \mathbf{H}_{SO} | T_1 >^2$ and $\mu_{S_n - S_0}^2$, and of $< T_m | \mathbf{H}_{SO} | S_0 >^2$ and $\mu_{T_m - T_1}^2$ even in the long-wavelength region (Fig. 4a). In terms of SOC elements, Eqs. (1), (3), and (4) indicate that the selective enhancement of $k_r^T$ relative to $k_{nr}^T$ tends to occur when $< S_n | \mathbf{H}_{SO} | T_1 >^2$ and $< T_m | \mathbf{H}_{SO} | S_0 >^2$ are larger than $< T_1 | \mathbf{H}_{SO} | S_0 >^2$. For **1**–**5**, the calculated $k_r^T$ values based on the first-order perturbation showed a good correlation with the optically determined $k_r^T$ values (Supplementary Fig. 24b). For **3**, $n = 5$ and $m = 7$ mostly contribute to enhance $k_r^T$, while for **5**, $n = 2$ and $m = 9$ mainly

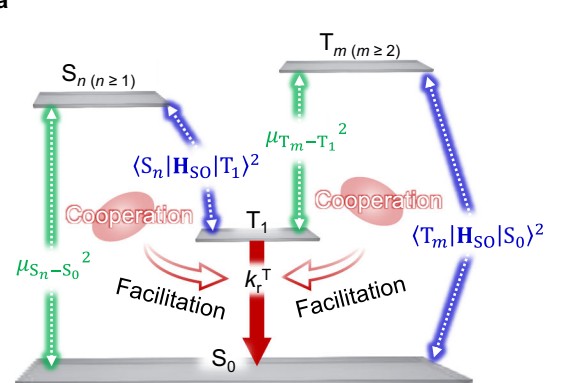

**Fig. 4 | Different spin–orbit coupling (SOC) enhancement levels between different singlet and triplet state pairs. a** Schematic energy diagram illustrating the transition dipole moment between $S_n$ and $S_0$ ($\mu_{S_n-S_0}$), the SOC between $S_n$ and $T_1$ ($\langle S_n | \mathbf{H}_{SO} | T_1 \rangle$), the transition dipole moment between $T_m$ and $T_1$ ($\mu_{T_m-T_1}$), and the

SOC between $T_m$ and $S_0$ ($\langle T_m | \mathbf{H}_{SO} | S_0 \rangle$). **b** Histograms of the $\langle T_1 | \mathbf{H}_{SO} | S_0 \rangle^2$, $\langle S_5 | \mathbf{H}_{SO} | T_1 \rangle^2$, and $\langle T_7 | \mathbf{H}_{SO} | S_0 \rangle^2$ values for **3** (left) and $\langle T_1 | \mathbf{H}_{SO} | S_0 \rangle^2$, $\langle S_2 | \mathbf{H}_{SO} | T_1 \rangle^2$, and $\langle T_9 | \mathbf{H}_{SO} | S_0 \rangle^2$ values for **5** (right).

contribute to $k_r^T$ (Supplementary Figs. 27 and 28). For **3**, $\langle S_5 | \mathbf{H}_{SO} | T_1 \rangle^2$ for the driving force of $k_r^T$ greatly decreases compared with the large $\langle T_1 | \mathbf{H}_{SO} | S_0 \rangle^2$ (Fig. 4b(i)). Thus, while multiple horizontal hetero-substitutions of Se increases $\langle T_1 | \mathbf{H}_{SO} | S_0 \rangle^2$, the increase of $\langle S_n | \mathbf{H}_{SO} | T_1 \rangle^2$ leading to $k_r^T$ enhancement is weak. This characteristic was also confirmed for chromophores with a horizontal phenylseleno substituent at the 3-position of DBC, and $\Phi_p$ of the chromophore was indeed small (Supplementary Figs. 14 and 29–31, and Table 3). In contrast, for **5**, $\langle S_2 | \mathbf{H}_{SO} | T_1 \rangle^2$ for the driving force of $k_r^T$ is large relative to the small $\langle T_1 | \mathbf{H}_{SO} | S_0 \rangle^2$ (Fig. 4b(ii)), and it is comparable with $\langle S_5 | \mathbf{H}_{SO} | T_1 \rangle^2$ of **3**. Similarly, $\langle T_9 | \mathbf{H}_{SO} | S_0 \rangle^2$ of **5** is comparable with $\langle T_7 | \mathbf{H}_{SO} | S_0 \rangle^2$ of **3** (Fig. 4b(iii)). Thus, multiple vertical hetero-substitutions suppress the increase in $\langle T_1 | \mathbf{H}_{SO} | S_0 \rangle^2$, which causes an increase in $k_{nr}^T$, compared with multiple horizontal hetero-substitutions, but they do not stop the increases in $\langle S_n | \mathbf{H}_{SO} | T_1 \rangle^2$ and $\langle T_m | \mathbf{H}_{SO} | S_0 \rangle^2$, which leads to an increase in $k_r^T$ based on the relationship from Eqs. (3) and (4) in Fig. 5a. As a result, $\langle S_n | \mathbf{H}_{SO} | T_1 \rangle^2$ and $\langle T_m | \mathbf{H}_{SO} | S_0 \rangle^2$ of **5** increase to levels close to those of **3**.

$\langle S_n | \mathbf{H}_{SO} | T_1 \rangle^2$ of the multiple vertical hetero-substituted chromophore approaching that of the multiple horizontal hetero-substituted chromophore can be understood by considering the following three factors, which are the same factors as those explained the SOC regarding $k_{nr}^T$ in Fig. 3a. First, regarding the overlap of the orbitals before and after the transition that increases $\langle S_n | \mathbf{H}_{SO} | T_1 \rangle$ in Eq. (4), $\langle S_n | \mathbf{H}_{SO} | T_1 \rangle$ logically increases as the overlap between the orbital overlap density related to the $S_n-S_0$ transition and the orbital overlap density related to the $T_1-S_0$ transition increases. For **3**, $\langle S_5 | \mathbf{H}_{SO} | T_1 \rangle$ significantly contributes to the increase in $k_r^T$, but the overlap density of the $S_5-S_0$ transition [(HOMO-2)×(LUMO)] only slightly overlaps with that of the $T_1-S_0$ transition [(HOMO)×(LUMO)] between C and Se (green open circles at the top of Fig. 5b(i) and Supplementary Fig. 32a). Although $\langle S_2 | \mathbf{H}_{SO} | T_1 \rangle$ significantly contributes to the increase in $k_r^T$ for **5**, similarly, the overlap density of the $S_2-S_0$ transition [(HOMO-1)×(LUMO)] only has a small overlap with that of the $T_1-S_0$ transition [(HOMO)×(LUMO)] as a through-space interaction between C and Se (green open circle at the bottom of Fig. 5b(i) and Supplementary Fig. 32b). Therefore, the comparable poor orbital overlap hardly changes $\langle S_n | \mathbf{H}_{SO} | T_1 \rangle$ between **3** and **5**. Second, regarding the contribution of the HAE to increasing $\langle S_n | \mathbf{H}_{SO} | T_1 \rangle$ in Eq. (4), the distance between the overlapping site and Se does not significantly change between **3** and **5** (Fig. 5b(ii)). Therefore, the effect of the HAE does not result in a driving force that significantly differentiates between $\langle S_5 | \mathbf{H}_{SO} | T_1 \rangle$ of **3** and $\langle S_2 | \mathbf{H}_{SO} | T_1 \rangle$ of **5**. Third, regarding the

contribution of the orbital axis orientation to the change in $\langle S_n | \mathbf{H}_{SO} | T_1 \rangle$, for **3**, the orbital axes of (HOMO-2)×(LUMO) and (HOMO)×(LUMO) are in different directions (Fig. 5b(iii), top). Similarly, for **5**, the orbital axis of the through-space type (HOMO-1)×(LUMO) formed between Se and C is different from that of the (HOMO)×(LUMO) on DBC (Fig. 5b(iii), bottom). Therefore, the orbital axis orientation hardly changes $\langle S_n | \mathbf{H}_{SO} | T_1 \rangle$ between **3** and **5**. Therefore, these similar tendencies regarding the spatial overlap characteristics, HAE, and orbital orientation increase $\langle S_2 | \mathbf{H}_{SO} | T_1 \rangle^2$ of **5** to a comparable value to $\langle S_5 | \mathbf{H}_{SO} | T_1 \rangle^2$ of **3**. When the Se atoms in **5** are replaced with S atoms, the enhancement of $\langle S_n | \mathbf{H}_{SO} | T_1 \rangle^2$ becomes small owing to the lack of an orbital overlap factor (Supplementary Figs. 13 and 33–35, and Table 4).

The approach of $\langle T_m | \mathbf{H}_{SO} | S_0 \rangle$ of the multiple vertical hetero-substituted chromophore to that of the multiple horizontal hetero-substituted chromophore can also be understood from the three perspectives of the orbital overlap, HAE effect, and orbital orientation (Fig. 5c). Regarding the orbital overlap (Fig. 5c(i)), the $T_7-S_0$ transition in **3** is mainly composed of HOMO-3 and LUMO (Supplementary Fig. 36a), and the $T_9-T_1$ transition in **5** is composed of HOMO-2 and LUMO (Supplementary Fig. 36b). The spatial overlap density between HOMO-3 and LUMO on Se in **3** (green open circles at the top of Fig. 5c(i)) is larger than the through-space overlap density between HOMO-2 and LUMO formed between Se and C in **5** (green open circle at the bottom of Fig. 5c(i)). Therefore, the contribution of the orbital overlap term to the enhancement of $\langle T_7 | \mathbf{H}_{SO} | S_0 \rangle$ for **3** is larger than that to the enhancement of $\langle T_9 | \mathbf{H}_{SO} | S_0 \rangle$ for **5**. Regarding the contribution from the HAE, the distance between the HOMO-3 and LUMO overlap sites and Se for **3** is smaller than that for **5** (Fig. 5c(ii)). Therefore, for **3**, the HAE plays a role in increasing $\langle T_m | \mathbf{H}_{SO} | S_0 \rangle$. Regarding the orbital orientation (Fig. 5c(iii)), the orbital axis directions of the HOMO-3 and LUMO on the Se atoms slightly differ for **3** (Fig. 5c(iii), top). However, for **5**, the orbital axis direction of the HOMO-2 of Se significantly differs from that of the LUMO of C on DBC (Fig. 5c(iii), bottom). Therefore, the contribution of the orbital orientation term to the enhancement of $\langle T_7 | \mathbf{H}_{SO} | S_0 \rangle$ for **3** is smaller than that to the enhancement of $\langle T_9 | \mathbf{H}_{SO} | S_0 \rangle$ for **5**. Considering these three factors, the decrease in $\langle T_9 | \mathbf{H}_{SO} | S_0 \rangle$ due to the poor orbital overlap and HAE for **5** is compensated for by the increase in $\langle T_9 | \mathbf{H}_{SO} | S_0 \rangle$ due to the orbital orientation, which results in $\langle T_9 | \mathbf{H}_{SO} | S_0 \rangle$ for **5** being equivalent to $\langle T_7 | \mathbf{H}_{SO} | S_0 \rangle$ for **3**.

The cooperation between the high-order TDM and the maintained large $\langle S_n | \mathbf{H}_{SO} | T_1 \rangle$ and $\langle T_m | \mathbf{H}_{SO} | S_0 \rangle$ contributes an additional factor to the maintenance of the large $k_r^T$ in the suppressed $k_{nr}^T$ state in the

**a**

$$k_r^T \propto \left[ \sum_{n \geq 1} \frac{\langle S_n | \mathbf{H}_{SO} | T_1 \rangle}{E_{S_n - T_1}} \times \mu_{S_n - S_0} + \sum_{m \geq 2} \frac{\langle T_m | \mathbf{H}_{SO} | S_0 \rangle}{E_{T_m - S_0}} \times \mu_{T_m - T_1} \right]^2$$

**b**

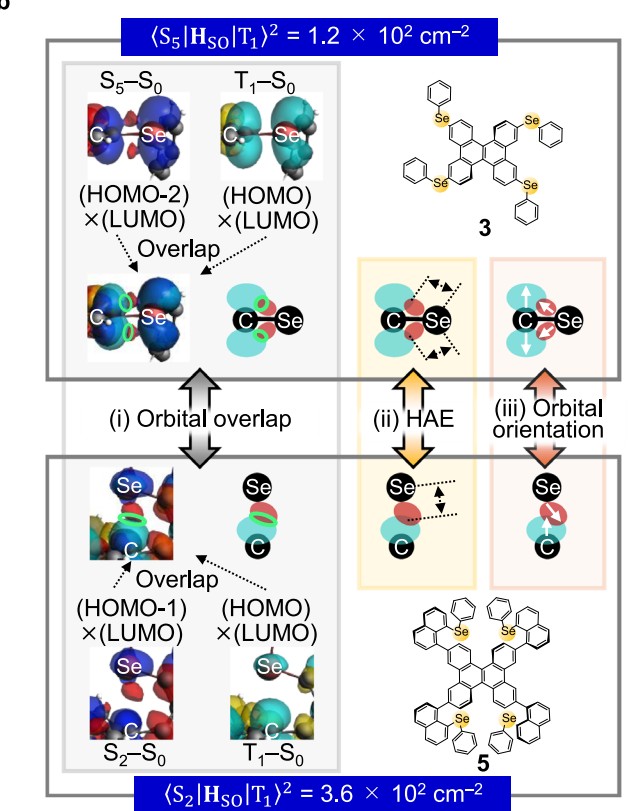

**c**

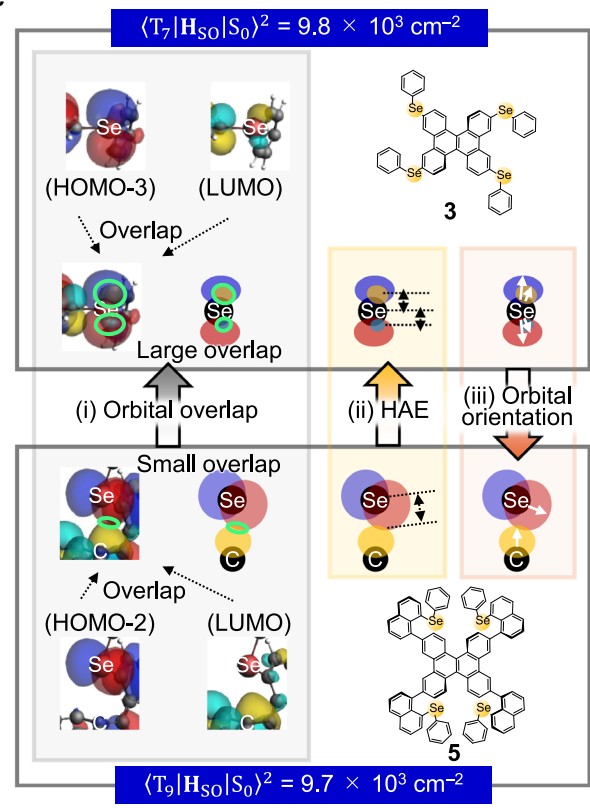

**Fig. 5 | Difference between horizontal and vertical hetero-substituents regarding the factors governing the spin-orbit coupling between $S_n$ and $T_1$ ($\langle S_n | \mathbf{H}_{SO} | T_1 \rangle$) and the spin-orbit coupling between $T_m$ and $S_0$ ($\langle T_m | \mathbf{H}_{SO} | S_0 \rangle$). a** Simplified expression of the equation of the rate constant of phosphorescence from the lowest triplet excited state ($k_r^T$). **b** A schematic diagram illustrating the equivalence of $\langle S_5 | \mathbf{H}_{SO} | T_1 \rangle^2$ of **3** (top) and $\langle S_2 | \mathbf{H}_{SO} | T_1 \rangle^2$ of **5** (bottom) using molecular orbitals, including the highest occupied molecular orbital (HOMO) and the lowest unoccupied molecular orbital (LUMO). The

isovalues of the overlap density regarding (HOMO-2)×(LUMO) and (HOMO)×(LUMO) for **3** and (HOMO-1)×(LUMO) and (HOMO)×(LUMO) for **5** are 0.00011. **c** Molecular orbitals and schematic illustrations explaining the comparable magnitudes of $\langle T_7 | \mathbf{H}_{SO} | S_0 \rangle^2$ for **3** (top) and $\langle T_9 | \mathbf{H}_{SO} | S_0 \rangle^2$ for **5** (bottom). The isovalue of 0.013 is used for HOMO-3 and LUMO of **3** and HOMO-2 and LUMO of **5**. In **b** and **c**, panels (i), (ii), and (iii) represent the orbital overlap, heavy atom effect (HAE), and orbital orientation effect, respectively. The white arrows in (iii) represent the orbital axis orientations.

multiple vertical hetero-substituted chromophores. For instance, for **3**, $\mu_{S_5 - S_0}{}^2$ is small because of the symmetry-forbidden nature of (HOMO-2)×(LUMO) as the overlap density of the $S_5 - S_0$ transition along the extended π-conjugation direction in the $x$ axis (Fig. 6a(i)). As well as the $S_5 - S_0$ transition, **3** has a symmetry-forbidden nature along the extended π-conjugation in the $T_7 - T_1$ transition, which prevents a large enhancement of $\mu_{T_7 - T_1}{}^2$ (Fig. 6a(ii)). The horizontal hetero-substituents therefore lead to cancellation of the large TDMs originating from the π-extended framework, weakening the $\langle S_5 | \mathbf{H}_{SO} | T_1 \rangle^2$ and $\mu_{S_5 - S_0}{}^2$ cooperation and $\langle T_7 | \mathbf{H}_{SO} | S_0 \rangle^2$ and $\mu_{T_7 - T_1}{}^2$ cooperation. In contrast, for **5**, $\mu_{S_2 - S_0}{}^2$ and $\mu_{T_9 - T_1}{}^2$ are large because both the $S_2 - S_0$ and $T_9 - T_1$ transitions are constructed from the symmetrically allowed nature along the long π-conjugation direction (Fig. 6b(i) and (ii)). The large TDMs enable strong cooperativities between $\mu_{S_2 - S_0}{}^2$ and $\langle S_2 | \mathbf{H}_{SO} | T_1 \rangle^2$, and between $\mu_{T_9 - T_1}{}^2$ and $\langle T_9 | \mathbf{H}_{SO} | S_0 \rangle^2$. As a result of the strong cooperativities and suppressed $\langle T_1 | \mathbf{H}_{SO} | S_0 \rangle^2$, selective enhancement of $k_r^T$ compared with $k_{nr}^T$ is realized for **5**. In previous strategies featuring horizontal hetero-substitutions, the extended π-conjugation could not be effectively utilized for the triplet radiation. The vertical hetero-substituent strategy can activate the long π

conjugation system for triplet radiation, which serves as a guideline for the design of new organic phosphorescence π-conjugated molecules.

### Demonstration of multi-label afterglow imaging

High-resolution multi-label imaging utilizing the colour and afterglow time (Fig. 7a) was demonstrated using **5d** with vertically introduced hetero-substituents as a chromophore with bright-red persistent RTP. Four water-dispersible crystalline particles with green RTP and short $\tau_p$ (particle A), green RTP and long $\tau_p$ (particle B), red RTP and short $\tau_p$ (particle C), and red RTP and long $\tau_p$ (particle D) were prepared (Fig. 7b, and Supplementary Note 6-1). The transmittance image confirmed that the existence of cells (Fig. 7c, and Fig. 7d, left) and the morphology of the cell membrane remained essentially unchanged for at least 90 min in the presence of the particles (Supplementary Fig. 37). Under irradiation with 360-nm excitation, significant autofluorescence was observed (Fig. 7d, right). However, 20–60 ms after ceasing excitation, only afterglow signals from the particles could be detected with a spatial resolution approaching the diffraction limit (Supplementary Fig. 38). The use of an optical bandpass filter in the green region and a long-pass filter above 685 nm confirmed the existence of green RTP

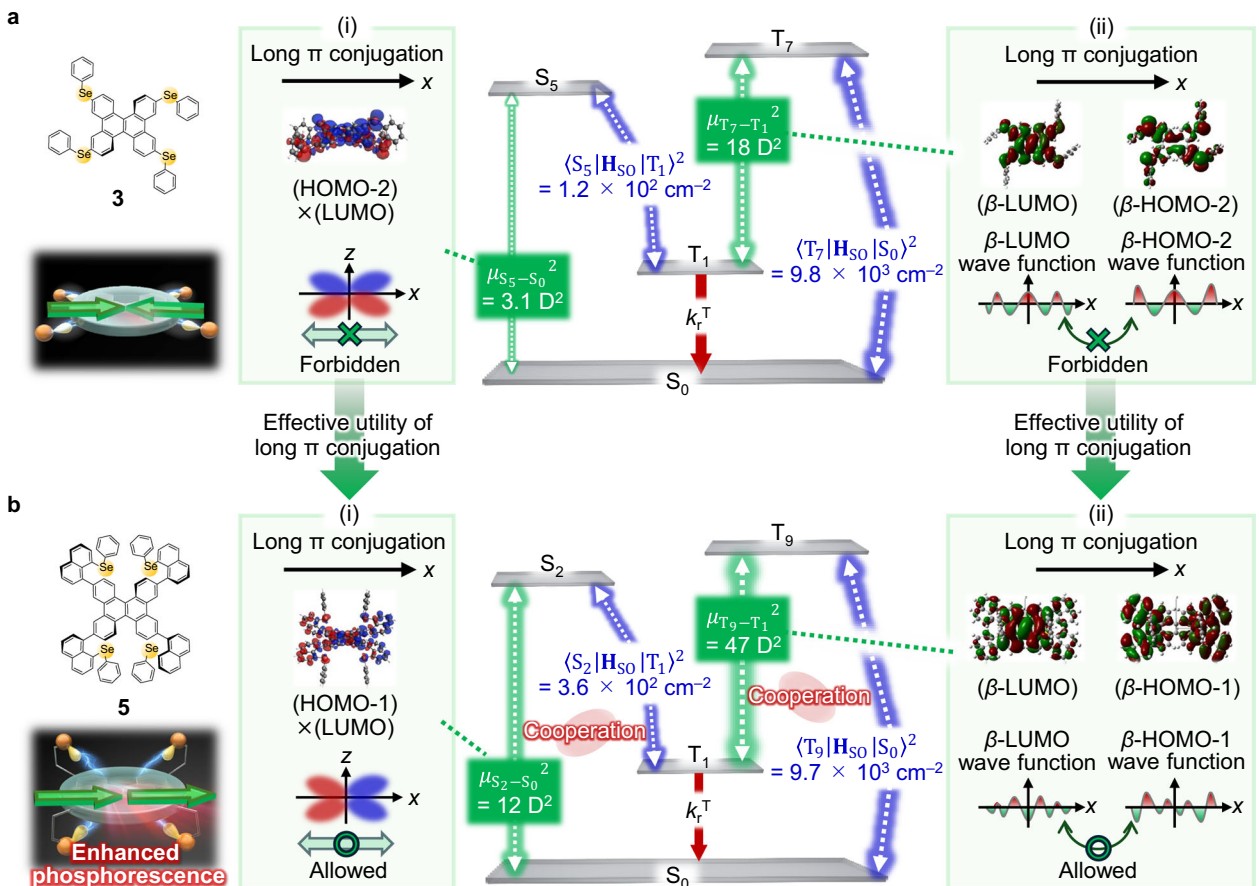

**Fig. 6 | Cooperativity between the transition dipole moment and spin-orbit coupling (SOC) in the triplet radiation process. a** The molecular orbitals and excitation energy levels associated with (i) $\mu_{S_5-S_0}{}^2$ and (ii) $\mu_{T_7-T_1}{}^2$, which are the main contributors to the rate constant of phosphorescence ($k_r^T$), are the $S_n$–$S_0$ transition dipole moment ($\mu_{S_n-S_0}$) and the $T_m$–$T_1$ transition dipole moment ($\mu_{T_m-T_1}$) in **3**. **b** The molecular orbitals and excited energy levels related to (i) $\mu_{S_2-S_0}{}^2$ and (ii) $\mu_{T_9-T_1}{}^2$ that mainly contribute to $k_r^T$ for **5**. In **a** and **b**, $\mu_{S_n-S_0}{}^2$, $<S_n|\mathbf{H}_{SO}|T_1>^2$, $\mu_{T_m-T_1}{}^2$, and $<T_m|\mathbf{H}_{SO}|S_0>^2$ in the energy diagram are

values calculated with the Amsterdam Density Functional 2025 package using the PBE0 functional and TZP basis set. The molecular orbitals shown in panel (ii) were calculated with the Gaussian09 program using the PBEPBE functional and 6-311 G(d,p) basis set. Regarding molecular orbitals, including the highest occupied molecular orbital (HOMO) and the lowest unoccupied molecular orbital (LUMO), the isovalue of 0.00016 is used for the overlap density of (HOMO-2)×(LUMO) in **3** and the overlap density of (HOMO-1)×(LUMO) in **5**. $\beta$-HOMO-2 and $\beta$-LUMO of **3** and $\beta$-HOMO-1 and $\beta$-LUMO of **5** are shown with an isovalue of 0.01.

particles (i.e., particles A or B) (Fig. 7e(i)) and red RTP particles (i.e., particles C or D) (Fig. 7e(ii)), respectively. Additionally, by detecting the afterglow at a later time, the green RTP from particle B (Fig. 7e(iii)) and red RTP from particle D (Fig. 7e(iv)) was identified (Supplementary Fig. 39).

## Discussion

Organic π-conjugated structures with different numbers of horizontal and vertical hetero-substituents were synthesized. The introduction of multiple horizontal substituents decreased the phosphorescence, while multiple vertical substituents enhanced the phosphorescence. Multiple hetero-substituents vertical to the π plane allow for the effective use of the transition dipole extended to the π plane to cooperate with the spin–orbit coupling to enhance the phosphorescence yield. So far, through-space interactions have been discussed as a factor related to the rigidity and $\Phi_{isc}$[36,37], where $\Phi_{isc}$ has been discussed only in terms of calculated values of $<S_1|\mathbf{H}_{SO}|T_m>^2$. In contrast, this study demonstrates that vertically introduced through-space substituents selectively enhance $<S_n|\mathbf{H}_{SO}|T_1>^2$ and $<T_m|\mathbf{H}_{SO}|S_0>^2$, and they are not directly related to $<S_1|\mathbf{H}_{SO}|T_m>^2 \propto \Phi_{isc}$. Additionally, increasing the number of vertical hetero-substituents generates strong cooperativity between $<S_n|\mathbf{H}_{SO}|T_1>^2$ and the TDM, as well as between $<T_m|\mathbf{H}_{SO}|S_0>^2$ and the TDM, for enhanced phosphorescence. Using

chromophores that emit persistent red organic phosphorescence, this molecular design demonstrated afterglow imaging without auto-fluorescence using nanoparticles with different colours and phosphorescence delay times. The molecular design methodology presented in this paper to increase the organic persistent RTP of various colours creates a new path to overcome the limitation of the number of labels in high-resolution bioimaging using two-dimensional photodetectors.

## Methods

### Synthesis and film preparation

Chromophore **1** was commercially available (Tokyo Chemical Industry (TCI), Tokyo, Japan). The synthesis procedures of **2**–**5d** are described in the Supplementary Note 1. To prepare the amorphous film, each chromophore was dissolved in melted $\beta$-estradiol (TCI) at a concentration of 0.3 wt%, and the mixture was sandwiched between two quartz substrates at 240 °C. The substrates were quenched to room temperature to produce an amorphous $\beta$-estradiol film doped with the chromophore.

### Photophysical characteristics

The emission spectrum and RTP decay characteristics were collected using a photonic multichannel analyser (PMA-12, Hamamatsu

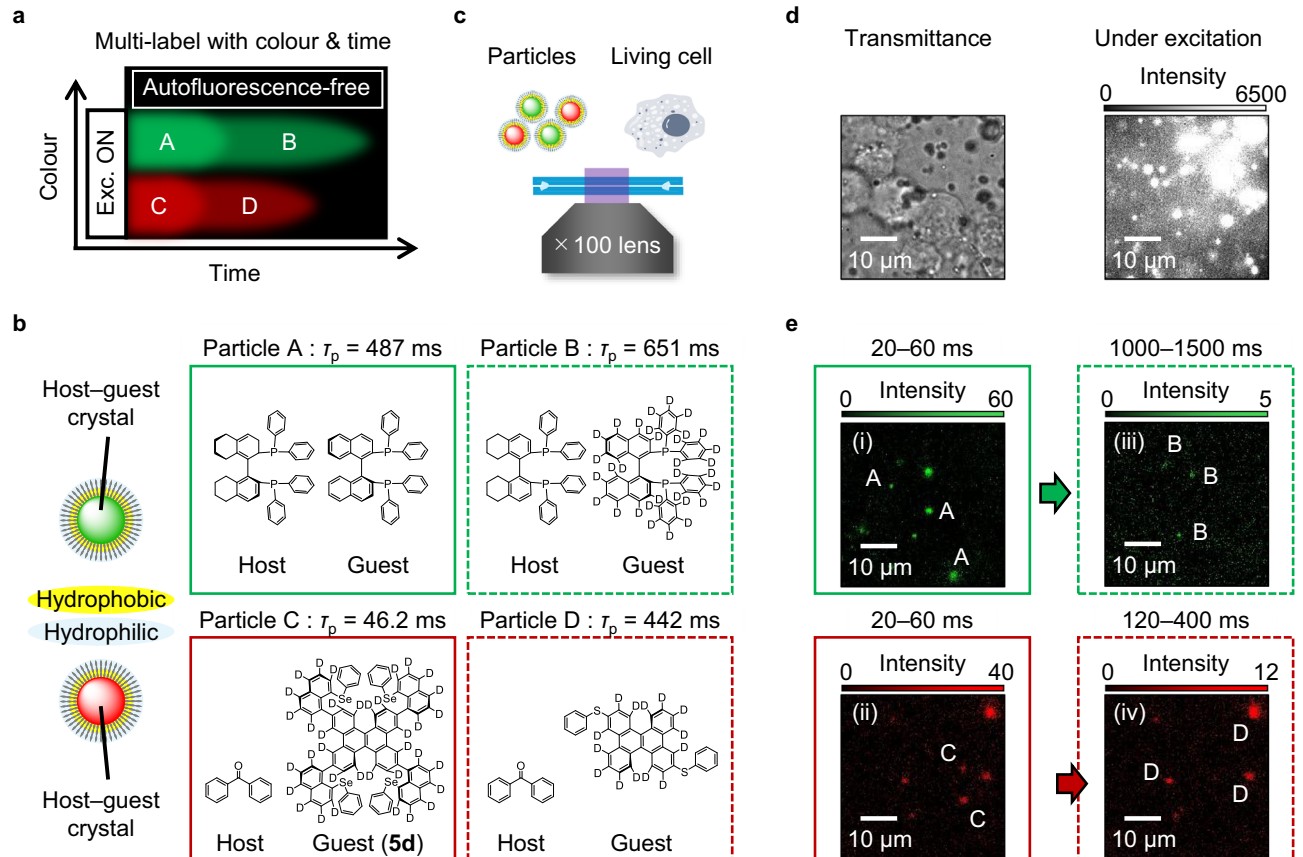

**Fig. 7 | High-resolution bioimaging using the RTP colour and decay characteristics. a** Concept. **b** Chemical structures of the host and guest molecules used for particles A–D. **c** Condition of the in vitro imaging. **d** Transmittance image of a HEK293 cell culture sample in the presence of particles A–D (left) and an emission image under 360-nm excitation (right). **e** Afterglow images in different time ranges after ceasing excitation using (i) and (iii) a bandpass filter in the green region and (ii) and (iv) a long-pass filter above 685 nm. In (i) and (iii), the images were collected 20–60 ms after ceasing excitation. In (ii) and (iv), the images were collected 1000–1500 and 120–400 ms after ceasing excitation, respectively.

Photonics, Shizuoka, Japan) and monochromatic light from the excitation unit of a fluorimeter (FP-8300, JASCO, Tokyo, Japan). An absolute quantum yield measurement system (C9920-02G, Hamamatsu Photonics) was used to determine the emission quantum yields. The intersystem crossing yields of the synthesized chromophores were determined based on a transient absorption tequnique[38,39] using a subnanosecond transient absorption spectrophotometer (picoTAS, Unisoku, Osaka, Japan) with a 355 nm Q-switched microchip laser (PNV-M02510-1×0, Teem Photonics, Meylan, France).

**Theoretical calculations**

The optimized geometry of $T_1$ was determined by the Gaussian09 program based on density functional theory (DFT). The optimization was performed using the B3LYP functional and 6-311 + G(d,p) basis set. Using the optimized $T_1$ geometry, the SOC elements, $\mu_{S_n - S_0}$, and $\mu_{T_m - T_1}$ were calculated using the Amsterdam Density Functional (ADF) 2025 package with the PBE0 functional and TZP basis set. The molecular orbitals associated with $\mu_{T_m - T_1}$ were estimated by time-dependent DFT (TD-DFT) calculation with the Gaussian09 program using the PBEPBE functional and 6-311 G(d,p) basis set. In the TD-DFT calculations, the option iop(3/76 = 1000002500) was applied to adjust the fraction of the Hartree–Fock exchange to be consistent with the PBE0 functional.

**Preparation of the crystalline particles**

Water-dispersible afterglow particles A–D were prepared in a similar manner by changing the host and guest molecules. For example, for particle A, (S)-(−)-2,2'-bis(diphenylphosphino)-1,1'-binaphthyl ((S)-

BINAP) (Sigma-Aldrich, St Louis, MO, USA) and (S)-(−)-2,2'bis(diphenylphosphino)-5,5',6,6',7,7'8,8'-octahydro-1,1'-binaphthyl ((S)-H$_8$-BINAP) (Sigma-Aldrich) were used as the guest and host molecules, respectively (Fig. 4b)[40]. (S)-BINAP was dissolved in melted (S)-H$_8$-BINAP at a concentration of 1.0 wt% at 240 °C. The mixture was then annealed at 160 °C for 5 min to prepare the 1.0 wt%-(S)-BINAP-doped (S)-H$_8$-BINAP crystalline solid. The crystalline solid (1.0 mg) was added to an aqueous solution of F127 (Sigma-Aldrich) (3.3 g L$^{-1}$, 1.0 mL), and then the aqueous solution was stirred with an ultrasonic homogenizer (UR-20P, Tomy Seiko Co., Ltd., Tokyo, Japan) for 10 min in an ice bath. The mixture was then filtered through a hydrophilic membrane filter (Millex-AP, glass fiber, 2.0 μm, Merck Millipore Ltd., Tullagreen, Carrigtwohill, Ireland) with a specific pore size. After the filtered solution was centrifuged, the aqueous solution was replaced with pure water to produce particle A dispersed in water. The preparation procedures for particles B–D are described in the Supplementary Note 6-1.

**Sample preparation for in vitro imaging**

Human embryonic kidney (HEK) 293tSA[41] cells were plated on polyethyleneimine-coated glass coverslips and cultured in Dulbecco's Modified Eagle's Medium high glucose (Sigma-Aldrich) supplemented with 10% foetal bovine serum, 2 mM glutamine (Gibco) (Thermo Fisher Scientific, Waltham, MA, USA) and 1% penicillin–streptomycin solution (Gibco) (Thermo Fisher Scientific) at 37 °C under 10% CO$_2$ (MCO-19AIC PHCbi, Tokyo, Japan). For the microscopic experiments, each of the aqueous solution (0.5 μL) dispersed crystalline particles A, B, C, and D was injected into the cultured cells.

## Microscopic measurements

The high-resolution multi-label imaging was performed with an inverted optical microscope (IX73, Olympus, Tokyo, Japan) using an oil immersion objective lens (UPlan FLN×100/1.3 NA, oil, Olympus). A continuous-wave laser emitting at 360 nm (UV-F-360, CNI, Changchun, China) was used as the excitation source. The two-dimensional images were captured by an electron multiplying charge-coupled device (iXon Ultra/life 897, Oxford Instruments, Abingdon, UK). The transmittance image and emission image under excitation were collected through a 405 nm dichroic mirror (Di02-R405, Semrock, NY, USA) and a 405 nm long-pass filter (BLP01-405R, Semrock). A bandpass filter (FF01-520/44-25, Semrock) and a 470 nm dichroic mirror (FF470-Di01, Semrock) were used to detect the green afterglow from particle A and particle B. To detect the red afterglow from particle C and particle D, a long-pass filter (FF01-685/LP-25, Semrock) and a 470 nm dichroic mirror (FF470-Di01, Semrock) were used. The afterglow images of the green channel were captured in the time ranges of 20–60 and 1000–1500 ms after stopping excitation to distinguish the afterglow signals from particle A and particle B, respectively. Similarly, the afterglow images of the red channel were captured in the time range of 20–60 and 120–400 ms after ceasing excitation to distinguish the afterglow signals from particle C and particle D, respectively. In the high-resolution imaging experiments, the excitation intensity was 200 mW cm$^{-2}$.

## Data availability

All relevant data in the main text or the Supplementary materials are available from the corresponding authors upon request. Atomic coordinates for quantum chemical calculations are provided as Supplementary Data Files. Source data are provided with this paper.

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

## Acknowledgements

We thank Dr. Shinji Matsuda for advice and support with cell culture. S.H. acknowledges the support of JSPS KAKENHI grant numbers: JP25H01260 (Grant-in-Aid for Transformative Research Areas, "π-molecular complexity") and JP24K01567, and the Fusion Oriented Research for disruptive Science and Technology (FOREST) of the Japan Science and Technology Agency: JPMJFR201T.

## Author contributions

K.H. wrote the paper. K.H., R.S., and R.M. synthesized chemicals. K.H. and S.H. contributed to the theoretical calculations and analyses. R.S. prepared a cell-cultured sample. K.H. performed microscopic and photophysical measurements. S.H. planned and supervised the project. All authors contributed to the writing of this manuscript and have approved the final version.

## Competing interests

The authors declare no competing interests.
