## [Transparent Peer Review file · Nature Communications]

Vertical substitution strategy to enable cooperation between spin–orbit coupling and transition dipoles for organic phosphorescence

Corresponding Author: Professor Shuzo Hirata

Version 0:

Reviewer comments:

Reviewer #1

(Remarks to the Author)

This manuscript reports a molecular design strategy for enhancing organic room-temperature phosphorescence (RTP) by introducing “vertical” hetero-substituents. The authors demonstrate that cooperative effects between transition dipole moments (TDM) and spin–orbit coupling (SOC) selectively enhance the radiative rate of the $T_1 \rightarrow S_0$ transition, leading to efficient RTP. With further discussion on the role of higher triplet states, a more explicit dissection of heavy-atom versus orbital-overlap effects, and a stronger connection to established EHE concepts, the work will have even greater impact. I recommend acceptance after minor revision.

(1) The discussion mainly focuses on SOC between the lowest triplet state T_1 and higher singlet states (S_n). However, the possible contribution of higher triplet states (T_n , $n > 1$) to the radiative transition process has not been addressed. Since multi-state interactions may play a role in modulating $T_1 \rightarrow S_0$ phosphorescence, a more systematic consideration of such contributions would strengthen the mechanistic interpretation.

(2) The manuscript attributes the increase in k_r^T to the cooperative interplay between $\mu_{S_n-S_0}$ and $\langle S_n | H_{SO} | T_1 \rangle$. A critical question remains: is the main driving force the intrinsic heavy-atom effect of Se, or the optimization of orbital overlap and spatial distribution of frontier orbitals? From Fig. 3 it is clear that HOMO–LUMO overlap differs significantly between substituent patterns. I recommend further quantitative comparison, for instance by replacing Se with lighter congeners (S or O) to clarify whether the enhanced k_r^T originates predominantly from heavy-atom SOC or orbital overlap effects.

(3) The vertical substitution strategy may be conceptually related to the “intramolecular external heavy atom effect” (EHE), where the heavy atom is not directly conjugated with the emissive core but exerts SOC through spatial proximity. It would be valuable for the authors to elaborate on this analogy and position their design within the broader context of heavy-atom effect literature.

(4) The definition of “cooperative effect” should be more precisely articulated—whether it refers to a simple multiplicative relationship between $\mu_{S_n-S_0}$ and $\langle S_n | H_{SO} | T_1 \rangle$, or if energy matching and symmetry considerations are also involved.

(5) The discussion would benefit from a clearer comparison with previous “horizontal substitution” strategies, highlighting why they fail to enhance RTP while the vertical substitution succeeds.

Reviewer #2

(Remarks to the Author)

In this paper, the author presents that enhanced phosphorescence was achieved by adjusting vertical hetero-substituents into π -conjugated structures. Multiple hetero-substituents vertical to the π plane allow for the effective use of the transition dipole extended to the π plane to cooperate with the spin-orbit coupling to enhance the phosphorescence yield. Considering these thorough results and the potential importance of this research, I believe this work should be published with improved organization and a clear summary.

1. The phosphorescence radiative rate (k_r) and lifetime (τ_P) are intrinsically governed by the energy gap (Δ_{EST}) between

the triplet state (T1) and the ground state (S0), as well as the magnitude of the transition dipole moment for the T1→S0 transition. A larger transition dipole moment and a smaller ΔE_{ST} generally favor a higher k_r , leading to a shorter τ_P . This fundamental relationship should be explicitly discussed in the manuscript to provide a more complete photophysical rationale for the observed phosphorescence performance.

2. The superior performance of compound 5 against compound 3 is likely attributable to a stronger through-space spin-orbit coupling effect based on the heavy atom. This enhancement is probably due to the optimized spatial proximity and orientation of the heavy selenium (Se) atom relative to the central π -core, allowing its potent heavy-atom effect to more effectively perturb the spin densities and facilitate the T1→S0 transition. It seems that explanations regarding this aspect are not mentioned in the article.

3. The proposed cooperation between strong SOC from high-lying singlets (S5 and S3) and a large transition dipole moment is an intriguing and less common mechanism for enhancing phosphorescence. However, is there any experimental observation evidence to support this hypothesis?

4. The claim regarding the vertical distribution of substituents relative to the central plane should ideally be corroborated by single-crystal X-ray diffraction data. If such crystal structures are available for the discussed compounds, referencing them would provide direct and unambiguous experimental evidence for this asserted conformation.

5. The manuscript's language will be thoroughly refined to enhance precision, clarity, and conciseness.

Reviewer #3

(Remarks to the Author)

In this work, Hayashi and colleagues present an innovative strategy to enhance organic RTP: cooperative spin-orbit and transition dipoles effect. They point out that the horizontally introduced substituent strategies commonly used in fluorescent materials are not suitable for phosphorescence. By employing a unique Se-containing DBC molecular system, the authors found that multiple vertical hetero-substituents lead to more efficient phosphorescence than horizontal substitution.

Theoretical calculations reveal that the reason lies in the larger transition dipole moment along the π plane enabled by vertical substitution, which facilitates the cooperative effect of a large transition dipole moment (TDM) and strong spin-orbit coupling (SOC). They further utilize their molecule in multi-label bioimaging to show its practical application. This work presents an effective and compelling strategy, making it well-suited for publication in Nat Commun. However, some minor revisions, particularly regarding theoretical analysis, should be addressed prior to acceptance.

1. The author may consider add the term through-space substitution or vertical substitution in the title to emphasize its importance in bringing cooperative spin-orbit and transition dipole and enhancing RTP.

2. On page 2, line 25, the authors claim that this molecular design can enhance the efficiency of organic phosphorescence to a level comparable to that of fluorescence across the entire visible spectrum. However, there is a lack of discussion regarding fluorescence enhancement in the article. Please clarify this point.

3. In Figure 2c, comparing RTP intensity through photoluminescence (PL) measurements for solid samples may be challenging due to multiple variables such as film inhomogeneity, doping concentration, and measurement distance. Therefore, it is recommended to supplement the figure with phosphorescence quantum yield values.

4. Although the authors cite some references in the Supporting Information (Section 5-3), which empirically illustrate that $k_r T$ is related to TDM(Sn-S0) and SOC(Sn-T1), the structure-property relationship remains unclear. Why the transition dipole moment of Sn and the SOC between Sn and T1 will affect the rate constant of T1? Could the authors provide an explanation from the electronic effect perspective instead of an empirical rule?

5. In Figure 3b, panels iii and iv show overlap between Ω_{Sn} and Ω_{T1} , but only Ω_{Sn} (red and blue) is clearly visible, while Ω_{T1} (orange and blue) is difficult to distinguish. Please clarify this more explicitly and include additional representations of Ω_{T1} in the Supporting Information. Moreover, for consistency, it is suggested that overlap density be used uniformly in panels i-iv, rather than switching between HOMO/LUMO and Ω_{Sn} representations when comparing compounds 3 and 5.

6. Multi-label imaging is an interesting application. Have the authors conducted cytotoxicity tests to demonstrate the biocompatibility of the material?

7. An efficient RTP system achieved through similar through-space substitution of main-group elements was recently reported (10.1021/jacs.4c17142), which should be cited. In addition, other recently developed high-performance red RTP systems (e.g., 10.1002/agt.2.70124, 10.1038/s41467-025-61714-0) should also be referenced in the Introduction.

Version 1:

Reviewer comments:

Reviewer #1

(Remarks to the Author)

All the issues have been well solved, the manuscript can be published as it is.

Reviewer #2

(Remarks to the Author)

The manuscript has been greatly improved in response to the reviewers' comments. It is recommended to be accepted.

Reviewer #3

(Remarks to the Author)

The authors have adequately revised their manuscript according to my previous comments and suggestions. The quality of the manuscript has been improved after the revision. The revised manuscript is recommended for publication in its present form.

Response letter to reviewers

Response to Reviewer 1

(Overall comments from Reviewer 1):

This manuscript reports a molecular design strategy for enhancing organic room-temperature phosphorescence (RTP) by introducing “vertical” hetero-substituents. The authors demonstrate that cooperative effects between transition dipole moments (TDM) and spin-orbit coupling (SOC) selectively enhance the radiative rate of the $T_1 \rightarrow S_0$ transition, leading to efficient RTP. With further discussion on the role of higher triplet states, a more explicit dissection of heavy-atom versus orbital-overlap effects, and a stronger connection to established EHE concepts, the work will have even greater impact. I recommend acceptance after minor revision.

(Response)

We sincerely appreciate your constructive comments. We have addressed your comments, as outlined in the following responses. Below, our response to the comments from Reviewer 1 is in blue, and the specific text and supporting information that we revised are in red.

(Comment 1 from Reviewer 1)

The discussion mainly focuses on SOC between the lowest triplet state T_1 and higher singlet states (S_n). However, the possible contribution of higher triplet states (T_n , $n > 1$) to the radiative transition process has not been addressed. Since multi-state interactions may play a role in modulating $T_1 \rightarrow S_0$ phosphorescence, a more systematic consideration of such contributions would strengthen the mechanistic interpretation.

(Response)

Thank you for your important comment. As Reviewer 1 pointed out, the rate constant of the radiation transition from T_1 (k_r^T) is enhanced by contribution of higher triplet states (T_m , $m \geq 2$). The k_r^T based on the first order perturbation is expressed as (*Chem. Rev.* **66**, 199–241 (1966); *J. Phys. Chem. A* **111**, 10490–10499 (2007); *Chem. Rev.* **117**, 6500–6537 (2017)),

$$k_r^T = \frac{E_{T_1-S_0}^3}{3\varepsilon_0\pi\hbar^4c^3} |\mu_{T_1-S_0}|^2, \quad (\text{R1-1})$$

$$\mu_{T_1-S_0} = \sum_{n \geq 1} \frac{\langle S_n | \mathbf{H}_{SO} | T_1 \rangle}{E_{S_n-T_1}} \mu_{S_n-S_0} + \sum_{m \geq 2} \frac{\langle T_m | \mathbf{H}_{SO} | S_0 \rangle}{E_{T_m-S_0}} \mu_{T_m-T_1}, \quad (\text{R1-2})$$

where $E_{T_1-S_0}$ is the energy gap between T_1 and S_0 , ε_0 is the vacuum permittivity, \hbar is the Dirac constant, c is the speed of light, $\mu_{T_1-S_0}$ is the transition dipole moment (TDM) between T_1 and S_0 , $\langle S_n | \mathbf{H}_{SO} | T_1 \rangle$ is the spin-orbit coupling (SOC) between the n th-order singlet excited state (S_n , $n \geq 1$) and T_1 , $E_{S_n-T_1}$ is the energy gap between S_n and T_1 , $\mu_{S_n-S_0}$ is the TDM between S_n and S_0 , $\langle T_m | \mathbf{H}_{SO} | S_0 \rangle$ is the SOC between T_m and S_0 , $E_{T_m-S_0}$ is the energy gap between T_m and S_0 , $\mu_{T_m-T_1}$ is the TDM between T_m and T_1 . These parameters are visualized in Fig. R1a. Equation (R1-2) shows that the $\langle T_m | \mathbf{H}_{SO} | S_0 \rangle$ and $\mu_{T_m-T_1}$ contribute to k_r^T as well as $\langle S_n | \mathbf{H}_{SO} | T_1 \rangle$ and $\mu_{S_n-S_0}$. In original manuscript, the Amsterdam Density Function (ADF) 2018 package was used to calculate k_r^T values. However, the calculation version could not evaluate the contribution of T_m , and the obtained k_r^T reflected only first term of equation (R1-2). Therefore, we performed quantum chemical calculations with considerations of second term of equation (R1-2) using the ADF 2025 package. Because the calculation methods are changed from original manuscript, the parameters for SOC elements, TDM values, and calculated k_r^T values are also changed. The calculated k_r^T values still have good positive correlation with the optically measured k_r^T values (Fig. R1b(i)). Furthermore, the calculated $\langle T_1 | \mathbf{H}_{SO} | S_0 \rangle^2$, which proportional to the rate constant of the nonradiative transition from T_1 (k_{nr}^T), was consistent with optically determined k_{nr}^T (Fig. R1b(ii)). Therefore, we could discuss the changes in k_r^T and k_{nr}^T based on ADF 2025 calculation. In particular, the magnitude of k_r^T of **5** was approximately maintained up to that of **3** when compared with the difference in the k_{nr}^T between **3** and **5** (arrows in Fig. R1b(i) and (ii)). Due to the maintenance of k_r^T relative to the suppressed k_{nr}^T , **5** has efficient RTP capability than **3**.

Fig. R1 a, Schematic energy diagram illustrating the key factors contributing to k_r^T . **b**, Relationship (i) between optically measured and calculated k_r^T and (ii) between optically measured k_{nr}^T and calculated $\langle T_1 | \mathbf{H}_{SO} | S_0 \rangle^2$. The quantum chemical calculations were performed using ADF 2025 package with considerations of $\langle T_m | \mathbf{H}_{SO} | S_0 \rangle$, $E_{T_m-S_0}$, and $\mu_{T_m-T_1}$ contributions in addition to the $\langle S_n | \mathbf{H}_{SO} | T_1 \rangle$, $E_{S_n-T_1}$, and $\mu_{S_n-S_0}$ contributions.

The maintenance of k_r^T of **5** relative to its suppressed k_{nr}^T involves contribution from $\langle T_m | \mathbf{H}_{SO} | S_0 \rangle^2$ as well as from $\langle S_n | \mathbf{H}_{SO} | T_1 \rangle^2$. In terms of SOC elements, equations (R1-1) and (R1-2) indicate that the relative enhancement of k_r^T to k_{nr}^T tends to occur when $\langle S_n | \mathbf{H}_{SO} | T_1 \rangle^2$ and $\langle T_m | \mathbf{H}_{SO} | S_0 \rangle^2$ are larger than $\langle T_1 | \mathbf{H}_{SO} | S_0 \rangle^2$. For **3**, $n = 5$ and $m = 7$ mostly contribute to enhance k_r^T , while for **5**, $n = 2$ and $m = 9$ mainly contribute to k_r^T . For **3**, $\langle S_5 | \mathbf{H}_{SO} | T_1 \rangle^2$ for the driving force of k_r^T greatly decreases compared with the large $\langle T_1 | \mathbf{H}_{SO} | S_0 \rangle^2$ (Fig. R2(i)). In contrast, for **5**, $\langle S_2 | \mathbf{H}_{SO} | T_1 \rangle^2$ for the driving force of k_r^T is large relative to the small $\langle T_1 | \mathbf{H}_{SO} | S_0 \rangle^2$ (Fig. R2(ii)), and it is maintained up to the $\langle S_5 | \mathbf{H}_{SO} | T_1 \rangle^2$ of **3**. Similarly, $\langle T_9 | \mathbf{H}_{SO} | S_0 \rangle^2$ of **5** is comparable with $\langle T_7 | \mathbf{H}_{SO} | S_0 \rangle^2$ of **3** (Fig. R2(iii)). Thus, the **5** suppresses the increase in $\langle T_1 | \mathbf{H}_{SO} | S_0 \rangle^2$, which leads to an increase in k_{nr}^T , but it does not stop the increases in $\langle T_m | \mathbf{H}_{SO} | S_0 \rangle^2$ as well as $\langle S_n | \mathbf{H}_{SO} | T_1 \rangle^2$ to enhance k_r^T .

Fig. R2 Histograms of $\langle T_1 | H_{SO} | S_0 \rangle^2$, $\langle S_5 | H_{SO} | T_1 \rangle^2$, and $\langle T_7 | H_{SO} | S_0 \rangle^2$ values for **3** (left) and $\langle T_1 | H_{SO} | S_0 \rangle^2$, $\langle S_2 | H_{SO} | T_1 \rangle^2$, and $\langle T_9 | H_{SO} | S_0 \rangle^2$ values for **5** (right).

Furthermore, high-order TDMs collaborating with the maintained SOC in T_m contribution as well as S_n contribution serve as an additional factor that sustains the large k_r^T relative to suppressed k_{nr} in **5**. For instance, for **3**, $\mu_{S_5-S_0}^2$ is small because of the symmetry-forbidden nature of the overlap density between HOMO-2 and LUMO for the S_5-S_0 transition along the extended π -conjugation direction in the x axis (Fig. R3a(i)). As well as the S_5-S_0 transition, **3** has a symmetry forbidden nature along the extended π -conjugation in the T_7-T_1 transition, which prevents a large enhancement of $\mu_{T_7-T_1}^2$ (Fig. R3a(ii)). The horizontal hetero-substituents therefore lead to cancellation of the large TDMs originating from the π -extended framework, weakening the $\langle S_5 | H_{SO} | T_1 \rangle^2$ and $\mu_{S_5-S_0}^2$ cooperation and $\langle T_7 | H_{SO} | S_0 \rangle^2$ and $\mu_{T_7-T_1}^2$ cooperation. In contrast, for **5**, $\mu_{S_2-S_0}^2$ and $\mu_{T_9-T_1}^2$ are large because both the S_2-S_0 and T_9-T_1 transitions are constructed from the symmetrically allowed nature along the long π -conjugation direction (Fig. R3b(i) and (ii)). The large TDMs enable strong cooperativities between $\mu_{S_2-S_0}^2$ and $\langle S_2 | H_{SO} | T_1 \rangle^2$, and between $\mu_{T_9-T_1}^2$ and $\langle T_9 | H_{SO} | S_0 \rangle^2$. As a result of the strong cooperativities and suppressed $\langle T_1 | H_{SO} | S_0 \rangle^2$, selective enhancement of k_r^T relative to k_{nr}^T is realized for **5**. Thus, the cooperative effect of TDM and SOC is confirmed not only in the S_n contribution but also in the T_m contribution in vertically hetero-substituents, leading to enhanced organic phosphorescence process.

Fig. R3 a, Molecular orbitals and energy diagram relating (i) $\mu_{S_5-S_0}^2$ and (ii) $\mu_{T_7-T_1}^2$ for **3**. **b**, Molecular orbitals and energy diagram relating (i) $\mu_{S_2-S_0}^2$ and (ii) $\mu_{T_9-T_1}^2$ for **5**. In **a** and **b**, $\mu_{S_n-S_0}^2$, $\langle S_n | H_{SO} | T_1 \rangle^2$, $\mu_{T_m-T_1}^2$, and $\langle T_m | H_{SO} | S_0 \rangle^2$ in energy diagram values calculated with the Amsterdam Density Functional (ADF) 2025 package using the PBE0 functional and TZP basis set. The molecular orbitals shown in panel (ii) were calculated with Gaussian09 program using the PBE/PBE functional and 6-311G(d,p) basis set. The isovalue of 0.00016 is used for the overlap density between HOMO-2 and LUMO [(HOMO-2) \times (LUMO)] in **3** and the overlap density between HOMO-1 and LUMO [(HOMO-1) \times (LUMO)] in **5**. β -HOMO-2 and β -LUMO of **3** and β -HOMO-1 and β -LUMO of **5** are shown with an isovalue of 0.01.

From the above, we changed the display items and sentences due to the addition of the T_m contribution. Although the sentences related to discussion on $\langle T_1 | H_{SO} | S_0 \rangle$, $\langle S_n | H_{SO} | T_1 \rangle$, and $\langle T_m | H_{SO} | S_0 \rangle$ was also changed in revised manuscript, the discussion on the SOC will describe in the next response.

First of all, we revised the illustration in Fig. 1a to revise the diagram to simply show the use of π -plane transition dipoles for triplet radiation as follow.

Before revision: Fig. 1a in original manuscript

Caption of Figure 1a, Illustration of the concept and energy diagrams of chromophores with horizontal hetero-substitution (left) and vertical hetero-substitution (right).”

After revision: Fig. 1a in revised manuscript

Caption of Figure 1a, Illustration of the concept and energy diagrams of chromophores with horizontal hetero-substitution (left) and vertical hetero-substitution (right).”

In addition, we revised the following sentences due to the modification of Fig. 1.

Before revision: Lines 8–17 in p. 4 in original manuscript

“The multiple horizontal hetero-substituents cause a large transition dipole moment (TDM) relating the singlet radiation process (TDM-S) along the conjugated plane, which generally improves the fluorescence yield (Φ_f) (Fig. 1a, blue arrow in the left figure). However, the multiple substituents often induce a TDM relating the triplet radiation process (TDM-T) in a direction orthogonal to the aromatic backbone (Fig. 1a, red arrow in the left figure). When the TDM-T direction can be aligned with the long-conjugation plane, efficient room-temperature phosphorescence (RTP) can be obtained. Here we report a molecular design that allows large TDM-T along the long-conjugation plane and efficient RTP by multiple vertical hetero-substitution (Fig. 1a, right).”

After revision: Lines 8–17 in p. 4 in revised manuscript

“Such multiple horizontal hetero-substituents cause a large transition dipole moment (TDM) related to the singlet radiation process, which generally improves the fluorescence yield (Φ_f). However, the TDM originating from the multiple substituents often cancels the TDM related to the triplet radiation process along the extended conjugation axis (Fig.

1a, left). When the large TDM originating from the extended π system can be used for the triplet radiation process, efficient room-temperature phosphorescence (RTP) can be obtained. Here we report a molecular design that allows a large TDM associated with triplet radiation along the extended conjugation axis and efficient RTP by multiple vertical hetero-substitutions (Fig. 1a, right).”

Next, we changed Fig. 3a of original manuscript to Fig. 4 of revised manuscript to illustrate the T_m contribution to k_r^T .

Before revision: Fig. 3a in original manuscript

Caption of Figure 3a, Schematic energy diagram illustrating (i) $\langle T_1 | H_{so} | S_0 \rangle$, (ii) $\langle S_n | H_{so} | T_1 \rangle$, and (iii) $\mu_{S_n-S_0}$.”

After revision: Fig. 4 in revised manuscript

Fig. 4 Selectivity of the spin-orbit coupling (SOC) enhancement among $\langle T_1|H_{SO}|S_0\rangle^2$, $\langle S_n|H_{SO}|T_1\rangle^2$, and $\langle T_m|H_{SO}|S_0\rangle^2$. **a**, Schematic energy diagram illustrating $\mu_{S_n-S_0}^2$, $\langle S_n|H_{SO}|T_1\rangle^2$, $\mu_{T_m-T_1}^2$, and $\langle T_m|H_{SO}|S_0\rangle^2$. **b**, Histograms of the $\langle T_1|H_{SO}|S_0\rangle^2$, $\langle S_5|H_{SO}|T_1\rangle^2$, and $\langle T_7|H_{SO}|S_0\rangle^2$ values for **3** (left) and $\langle T_1|H_{SO}|S_0\rangle^2$, $\langle S_2|H_{SO}|T_1\rangle^2$, and $\langle T_9|H_{SO}|S_0\rangle^2$ values for **5** (right)."

Furthermore, we changed the sentences regarding factors that associate with k_r^T because of addition of the T_m contribution.

Before revision: Lines 12–16 in p. 6 in original manuscript

“ k_r^T is approximated by $k_r^T \propto [\sum_n (\mu_{S_n-S_0} \langle S_n|H_{SO}|T_1\rangle)]^2$ ($n \geq 1$), where $\langle S_n|H_{SO}|T_1\rangle$ is the SOC between the n th-order singlet excited state (S_n) and T_1 (Fig. 3a(ii)), and $\mu_{S_n-S_0}$ is the TDM in the S_n-S_0 transition (Fig. 3a(iii))^{6,26}. Therefore, selective enhancement of k_r^T compared with k_{nr}^T requires the strong cooperation of $\langle S_n|H_{SO}|T_1\rangle^2$ and $\mu_{S_n-S_0}^2$ without a significant increase in $\langle T_1|H_{SO}|S_0\rangle^2$.”

After revision: From line 7 in p. 9 to line 1 in p. 10 in revised manuscript

“Another reason for the enhancement of k_r^T relative to k_{nr}^T by vertical hetero-substitution is that an increase in the number of vertical hetero-substitutions tends to increase k_r^T to a similar level as an increase in the number of horizontal hetero-substitutions. The equation for k_r^T based on first-order perturbation theory is expressed as^{6,29,30}

$$k_r^T = \frac{E_{T_1-S_0}^3}{3\varepsilon_0\pi\hbar^4c^3} |\mu_{T_1-S_0}|^2, \quad (3)$$

$$\mu_{T_1-S_0} = \sum_{n \geq 1} \frac{\langle S_n | \mathbf{H}_{SO} | T_1 \rangle}{E_{S_n-T_1}} \mu_{S_n-S_0} + \sum_{m \geq 2} \frac{\langle T_m | \mathbf{H}_{SO} | S_0 \rangle}{E_{T_m-S_0}} \mu_{T_m-T_1}, \quad (4)$$

where $E_{T_1-S_0}$ is the energy gap between T_1 and S_0 , ε_0 is the vacuum permittivity, \hbar is the Dirac constant, c is the speed of light, $\mu_{T_1-S_0}$ is the TDM between T_1 and S_0 , $\langle S_n | \mathbf{H}_{SO} | T_1 \rangle$ is the SOC between the n th-order singlet excited state (S_n , $n \geq 1$) and T_1 , $E_{S_n-T_1}$ is the energy gap between S_n and T_1 , $\mu_{S_n-S_0}$ is the TDM between S_n and S_0 , $\langle T_m | \mathbf{H}_{SO} | S_0 \rangle$ is the SOC between the m th-order triplet excited state (T_m , $m \geq 2$) and S_0 , $E_{T_m-S_0}$ is the energy gap between T_m and S_0 , and $\mu_{T_m-T_1}$ is the TDM between T_m and T_1 . Equation (3) indicates that the enhancement of k_r^T becomes increasingly difficult as the phosphorescence is red-shifted. From equation (4), however, k_r^T can still be facilitated by the cooperative contributions of $\langle S_n | \mathbf{H}_{SO} | T_1 \rangle^2$ and $\mu_{S_n-S_0}^2$, and of $\langle T_m | \mathbf{H}_{SO} | S_0 \rangle^2$ and $\mu_{T_m-T_1}^2$ even in the long-wavelength region (Fig. 4a).”

In addition, we added the sentences regarding comparison of $\langle T_1 | \mathbf{H}_{SO} | S_0 \rangle^2$, $\langle S_n | \mathbf{H}_{SO} | T_1 \rangle^2$, and $\langle T_m | \mathbf{H}_{SO} | S_0 \rangle^2$ values between **3** and **5**.

After revision: Lines 1–20 in p. 10 in revised manuscript

“In terms of SOC elements, equations (1), (3), and (4) indicate that the selective enhancement of k_r^T relative to k_{nr}^T tends to occur when $\langle S_n | \mathbf{H}_{SO} | T_1 \rangle^2$ and $\langle T_m | \mathbf{H}_{SO} | S_0 \rangle^2$ are larger than $\langle T_1 | \mathbf{H}_{SO} | S_0 \rangle^2$. For **1–5**, the calculated k_r^T values based on the first-order perturbation showed a good correlation with the optically determined k_r^T values (supplementary Fig. S24b). For **3**, $n = 5$ and $m = 7$ mostly contribute to enhance k_r^T , while for **5**, $n = 2$ and $m = 9$ mainly contribute to k_r^T (supplementary Figs. S27 and S28). For **3**, $\langle S_5 | \mathbf{H}_{SO} | T_1 \rangle^2$ for the driving force of k_r^T greatly decreases compared with the large $\langle T_1 | \mathbf{H}_{SO} | S_0 \rangle^2$ (Fig. 4b(i)). Thus, while multiple horizontal hetero-substitutions of Se increases $\langle T_1 | \mathbf{H}_{SO} | S_0 \rangle^2$, the increase of $\langle S_n | \mathbf{H}_{SO} | T_1 \rangle^2$ leading to k_r^T enhancement is weak. This characteristic was also confirmed for chromophores with a horizontal phenylseleno substituent at the 3-position of DBC, and Φ_p of the chromophore was indeed small (supplementary Figs. S14 and S29–S31, and Table S3). In contrast, for **5**, $\langle S_2 | \mathbf{H}_{SO} | T_1 \rangle^2$ for the driving force of k_r^T is large relative to the small $\langle T_1 | \mathbf{H}_{SO} | S_0 \rangle^2$ (Fig. 4b(ii)), and it is comparable with $\langle S_5 | \mathbf{H}_{SO} | T_1 \rangle^2$ of **3**. Similarly, $\langle T_9 | \mathbf{H}_{SO} | S_0 \rangle^2$ of **5** is comparable with $\langle T_7 | \mathbf{H}_{SO} | S_0 \rangle^2$ of **3** (Fig. 4b(iii)). Thus, multiple vertical hetero-substitutions suppress the increase in $\langle T_1 | \mathbf{H}_{SO} | S_0 \rangle^2$, which causes an increase in k_{nr}^T , compared with multiple horizontal hetero-substitutions, but they do not stop the increases in $\langle S_n | \mathbf{H}_{SO} | T_1 \rangle^2$ and

$\langle T_m | \mathbf{H}_{SO} | S_0 \rangle^2$, which leads to an increase in k_r^T based on the relationship from equations (3) and (4) in Fig. 5a. As a result, $\langle S_n | \mathbf{H}_{SO} | T_1 \rangle^2$ and $\langle T_m | \mathbf{H}_{SO} | S_0 \rangle^2$ of **5** increase to levels close to those of **3**.”

Next, we changed Fig. 3b(v) and (vi) of original manuscript to Fig. 6 of revised manuscript due to addition of the $\mu_{T_m-T_1}^2$ contribution.

Before revision: Fig. 3b(v) and (vi) in original manuscript

Caption of Figure 3b, Molecular orbitals and their calculated values relating (i) and (ii) $\langle T_1 | \mathbf{H}_{SO} | S_0 \rangle^2$, (iii) and (iv) $\langle S_n | \mathbf{H}_{SO} | T_1 \rangle^2$, and (v) and (vi) $\mu_{S_n-S_0}^2$. In (i) and (ii), the red-blue orbitals and orange-light blue orbitals are the highest occupied molecular orbital (HOMO) and lowest unoccupied molecular orbital (LUMO), respectively. In (iii) and (iv), the red-blue orbitals are Ω_{S_3} or Ω_{S_5} and the orange-light blue orbitals are Ω_{T_1} . In (i)–(iv), the white arrows represent the orbital angular momentum (OAM) vectors. The isovalues of the HOMO of **3**, LUMO of **3**, and HOMO of **5** are 0.02. The isovalue of 0.0085 is used for the LUMO of **5**. The isovalues of Ω_{S_3} , Ω_{S_5} , and Ω_{T_1} are 0.00016.”

After revision: Fig. 6 in revised manuscript

Fig. 6 Cooperativity between the transition dipole moment and SOC in the triplet radiation process. a, Molecular orbitals and energy diagram relating (i) $\mu_{S_5-S_0}^2$ and (ii) $\mu_{T_7-T_1}^2$ for **3**. **b,** Molecular orbitals and energy diagram relating (i) $\mu_{S_2-S_0}^2$ and (ii) $\mu_{T_9-T_1}^2$ for **5**. In **a** and **b**, $\mu_{S_n-S_0}^2$, $\langle S_n | H_{SO} | T_1 \rangle^2$, $\mu_{T_m-T_1}^2$, and $\langle T_m | H_{SO} | S_0 \rangle^2$ in the energy diagram are values calculated with the Amsterdam Density Functional 2025 package using the PBE0 functional and TZP basis set. The molecular orbitals shown in panel (ii) were calculated with the Gaussian09 program using the PBEPBE functional and 6-311G(d,p) basis set. The isovalue of 0.00016 is used for the overlap density of (HOMO-2) × (LUMO) in **3** and the overlap density of (HOMO-1) × (LUMO) in **5**. β-HOMO-2 and β-LUMO of **3** and β-HOMO-1 and β-LUMO of **5** are shown with an isovalue of 0.01.”

Furthermore, we changed the sentences in revised manuscript regarding the TDM contribution to k_r^T because of the addition of the $\mu_{T_m-T_1}^2$ contribution.

Before revision: From lines 19 in p. 7 to line 9 in p. 8 in original manuscript

“Finally, most importantly, the vertically introduced hetero-substituents have the characteristics of $\mu_{S_n-S_0}^2$ collaborating with the maintained large $\langle S_n | H_{SO} | T_1 \rangle^2$ for the enhanced phosphorescence (Fig. 3b, C). For instance, in **3**, $\mu_{S_5-S_0}$ mostly contributing to k_r^T is small along the z axis because of the symmetry-forbidden nature of the overlap density for the S₅–S₀ transition (Ω_{S_5}) in the x axis direction (Fig. 3b(v)). Owing to the small $\mu_{S_5-S_0}$, the cooperativity between $\langle S_5 | H_{SO} | T_1 \rangle^2$ and $\mu_{S_5-S_0}^2$ becomes weak. In contrast, $\mu_{S_3-S_0}^2$ mostly contributing to k_r^T of **5** is large because Ω_{S_3} has a symmetrically allowed transition nature along the long-conjugated direction in the x axis (Fig. 3b(vi)). The large $\mu_{S_3-S_0}^2$ leads to strong $\mu_{S_3-S_0}^2$ and $\langle S_3 | H_{SO} | T_1 \rangle^2$ cooperation. As a result of the small $\langle T_1 | H_{SO} | S_0 \rangle^2$ and strong cooperation between $\langle S_3 | H_{SO} | T_1 \rangle^2$ and $\mu_{S_3-S_0}^2$, selective

enhancement of k_r^T compared with k_{nr}^T is realized in **5**. Previous horizontal substitution in the π plane direction was unable to contribute to the TDM of the conjugated plane to triplet radiation. The vertically introduced hetero-substituents have the characteristics to effectively utilize the conjugated plane for triplet radiation, which serves as a guideline for the design of new organic phosphorescence π -conjugated molecules.”

After revision: Lines 1–20 in p. 13 in revised manuscript

“The cooperation between the high-order TDM and the maintained large $\langle S_n | \mathbf{H}_{SO} | T_1 \rangle$ and $\langle T_m | \mathbf{H}_{SO} | S_0 \rangle$ contributes an additional factor to the maintenance of the large k_r^T in the suppressed k_{nr}^T state in the multiple vertical hetero-substituted chromophores. For instance, for **3**, $\mu_{S_5-S_0}^2$ is small because of the symmetry-forbidden nature of (HOMO-2)×(LUMO) as the overlap density of the S_5-S_0 transition along the extended π -conjugation direction in the x axis (Fig. 6a(i)). As well as the S_5-S_0 transition, **3** has a symmetry-forbidden nature along the extended π -conjugation in the T_7-T_1 transition, which prevents a large enhancement of $\mu_{T_7-T_1}^2$ (Fig. 6a(ii)). The horizontal hetero-substituents therefore lead to cancellation of the large TDMs originating from the π -extended framework, weakening the $\langle S_5 | \mathbf{H}_{SO} | T_1 \rangle^2$ and $\mu_{S_5-S_0}^2$ cooperation and $\langle T_7 | \mathbf{H}_{SO} | S_0 \rangle^2$ and $\mu_{T_7-T_1}^2$ cooperation. In contrast, for **5**, $\mu_{S_2-S_0}^2$ and $\mu_{T_9-T_1}^2$ are large because both the S_2-S_0 and T_9-T_1 transitions are constructed from the symmetrically allowed nature along the long π -conjugation direction (Fig. 6b(i) and (ii)). The large TDMs enable strong cooperativities between $\mu_{S_2-S_0}^2$ and $\langle S_2 | \mathbf{H}_{SO} | T_1 \rangle^2$, and between $\mu_{T_9-T_1}^2$ and $\langle T_9 | \mathbf{H}_{SO} | S_0 \rangle^2$. As a result of the strong cooperativities and suppressed $\langle T_1 | \mathbf{H}_{SO} | S_0 \rangle^2$, selective enhancement of k_r^T compared with k_{nr}^T is realized for **5**. In previous strategies featuring horizontal hetero-substitutions, the extended π -conjugation could not be effectively utilized for the triplet radiation. The vertical hetero-substituent strategy can activate the long π conjugation system for triplet radiation, which serves as a guideline for the design of new organic phosphorescence π -conjugated molecules.”

Next, we changed the calculation procedure in Methods section because of the change in the calculation methods.

Before revision: Lines 5–7 in p. 12 in original manuscript

“Using the optimized T_1 geometry, the SOC elements and $\mu_{S_n-S_0}$ were calculated using the Amsterdam Density Functional (ADF) 2018 package with the PBE0 functional and TZP basis set (see the Supplementary Information).”

After revision: Lines 5–11 in p. 19 in revised manuscript

“Using the optimized T_1 geometry, the SOC elements, $\mu_{S_n-S_0}$, and $\mu_{T_m-T_1}$ were calculated

using the Amsterdam Density Functional (ADF) 2025 package with the PBE0 functional and TZP basis set. The molecular orbitals associated with $\mu_{T_m-T_1}$ were estimated by time-dependent DFT (TD-DFT) calculation with the Gaussian09 program using the PBE0 functional and 6-311G(d,p) basis set. In the TD-DFT calculations, the option `iop(3/76=1000002500)` was applied to adjust the fraction of the Hartree–Fock exchange to be consistent with the PBE0 functional.”

In addition, we changed the calculation result regarding T_1 optimized geometry in supplementary information because of the change in the calculation methods.

Before revision: Section 2 in original supplementary information

“2. T_1 optimized geometry of chromophores estimated by theoretical calculation

The optimized structures of the lowest triplet excited state (T_1) of **1–5** (Fig. S13) were determined by Gaussian09 program based on the density functional theory (DFT) using T_1 optfreq mode. In the calculation, B3LYP and 6-311G+(d,p) were used as a functional and a basis set, respectively. In T_1 optimized structure of **2** and **3**, Se atom is horizontally attached to DBC moiety. In **4** and **5**, where naphthalene is introduced as a spacer unit, Se atom is vertically placed on DBC because DBC and naphthalene unit are twisted by the steric hindrance.

Fig. S13. T_1 optimized structures of 1–5.”

After revision: Section 2 in revised supplementary information

“2. T_1 optimized geometry of chromophores estimated by theoretical calculation (Figure S15)

The optimized structures of the lowest triplet excited state (T_1) of 1–5 (Fig. S15) were determined by Gaussian09 program based on the density functional theory (DFT). In the calculation, B3LYP and 6-311G+(d,p) were used as a functional and a basis set, respectively. In T_1 optimized structure of 2 and 3, Se atom is horizontally attached to dibenzo[*g,p*]chrysene (DBC) moiety. In 4 and 5, where naphthalene is introduced as a

spacer unit, Se atom is vertically placed on DBC because DBC and naphthalene unit are twisted by the steric hindrance

Fig. S15. T_1 optimized structures of 1–5.”

Then, we updated the calculation results in revised supplementary information regarding the correlation between the measured and calculated k_r^T as well as the measured k_{nr}^T and calculated $\langle T_1 | H_{SO} | S_0 \rangle^2$ because of the change in the calculation methods.

Before revision: Section 5-1 in original supplementary information

“5-1. Calculated values of k_r^T and $\langle T_1 | H_{SO} | S_0 \rangle^2$ ”

Calculation procedure of T_1 optimized structure for 1–5 were described in supplementary section 2. Using the T_1 optimized structure (Fig. S13), single-point calculations were

performed using the time-dependent (TD) DFT with the Amsterdam Density Functional (ADF) 2018 package to calculate k_r^T and the spin-orbit coupling (SOC) between T_1 and S_0 ($\langle T_1 | \mathbf{H}_{SO} | S_0 \rangle$). The parameter $\langle S_n | \mathbf{H}_{SO} | T_1 \rangle$ was treated as a perturbation based on scalar relativistic orbitals using the PBE0 as a functional and TZP as a basis sets. The scalar relativistic-time-dependent DFT calculations included 10 singlet and 10 triplet excitations, which were used as the basis for the perturbative expansions in the calculations.

From the above calculations, the good correlation between calculated k_r^T and measured k_r^T (Fig. S21a) and between calculated $\langle T_1 | \mathbf{H}_{SO} | S_0 \rangle^2$ and measured k_{nr}^T (Fig. S21b) were observed. From **2** to **3**, the magnitude of calculated k_r^T enhancement (Fig. S21a, blue arrow) was smaller than that of $\langle T_1 | \mathbf{H}_{SO} | S_0 \rangle^2$ enhancement (Fig. S21b, blue arrow). In contrast, from **4** to **5**, calculated k_r^T was more enhanced compared with $\langle T_1 | \mathbf{H}_{SO} | S_0 \rangle^2$ (Figs. S21a and S21b, red arrows). Thus, the more selective enhancement of k_r^T compared with k_{nr}^T of **5** was observed from theoretical calculations as well as optical measurements. Therefore, the discussion on the differences in measured k_r^T and k_{nr}^T based on the quantum chemical calculation was reasonable.

Fig. S21. Relationship between measured and calculated parameters relating phosphorescence process. **a**, Correlation between measured and calculated k_r^T . **b**, Correlation between measured k_{nr}^T and calculated $\langle T_1 | \mathbf{H}_{SO} | S_0 \rangle^2$.”

After revision: Section 5-1 in revised supplementary information

“5-1. Calculated values of k_r^T and $\langle T_1 | \mathbf{H}_{SO} | S_0 \rangle^2$ (Figure S24)

Calculation procedure of T_1 optimized structure for **1–5** were described in supplementary Section 2. Using the T_1 optimized structure (Fig. S15), single-point calculations were performed using the time-dependent (TD) DFT with the Amsterdam Density Functional

(ADF) 2025 package to calculate k_r^T and the spin-orbit coupling (SOC) between T_1 and S_0 ($\langle T_1 | \mathbf{H}_{SO} | S_0 \rangle$). The parameter $\langle S_n | \mathbf{H}_{SO} | T_1 \rangle$ and $\langle T_m | \mathbf{H}_{SO} | S_0 \rangle$ were treated as a perturbation based on scalar relativistic orbitals using the PBE0 as a functional and TZP as a basis sets. The scalar relativistic-time-dependent DFT calculations included 10 singlet and 10 triplet excitations, which were used as the basis for the perturbative expansions in the calculations.

From the above calculations, the good correlation between calculated $\langle T_1 | \mathbf{H}_{SO} | S_0 \rangle^2$ and measured k_{nr}^T (Fig. S24a) and between calculated k_r^T and measured k_r^T (Fig. S24b) were observed among synthesized chromophores. From **2** to **3**, the magnitude of calculated k_r^T enhancement (Fig. S24b, blue arrow) was smaller than that of $\langle T_1 | \mathbf{H}_{SO} | S_0 \rangle^2$ enhancement (Fig. S24a, blue arrow). In contrast, from **4** to **5**, calculated k_r^T was more enhanced compared with $\langle T_1 | \mathbf{H}_{SO} | S_0 \rangle^2$ (Figs. S24a and S24b, red arrows). Thus, the more selective enhancement of k_r^T relative to k_{nr}^T of **5** was observed from theoretical calculations as well as optical measurements. Therefore, the discussion on the differences in measured k_r^T and k_{nr}^T based on the quantum chemical calculation is reasonable.

Fig. S24. Relationship between measured and calculated parameters relating phosphorescence process. **a**, Correlation between measured k_{nr}^T and calculated $\langle T_1 | \mathbf{H}_{SO} | S_0 \rangle^2$. **b**, Correlation between measured and calculated k_r^T .

Additionally, we changed the calculated parameters in revised supplementary information relating to the k_r^T due to the change of calculation methods and addition of the T_m contribution.

Before revision: Section 5-3 in original supplementary information

“5-3. Calculated parameters relating to k_r^T ”

The formula for k_r^T can be expressed approximately as the following equations^{6,7}.

$$k_r^T \approx \sum_n (\mu_{S_n-S_0} \times \lambda_n)^2, \quad (\text{S5})$$

$$\lambda_n = \langle S_n | \mathbf{H}_{SO} | T_1 \rangle / E_{S_n-T_1}, \quad (\text{S6})$$

where $\mu_{S_n-S_0}$ is the transition dipole moment between n^{th} order singlet excited states (S_n) and S_0 , $\langle S_n | \mathbf{H}_{SO} | T_1 \rangle$ is the SOC between S_n and T_1 , $E_{S_n-T_1}$ is the energy difference between S_n and T_1 . Based on the equations (S5) and (S6), good correlation between calculated and measured k_r^T was reported in many π -conjugated molecules^[1,3,5,7-13]. Therefore, the contributions of $\mu_{S_n-S_0}^2$ and $\langle S_n | \mathbf{H}_{SO} | T_1 \rangle^2$ to k_r^T were discussed to explain the selective k_r^T enhancement of **5** in main text.

The calculated values of $\mu_{S_n-S_0}^2 \lambda_n^2$ (Figs. S23a–S23d, top) indicate that $n = 6$, $n = 5$, $n = 9$, $n = 3$ are mainly contributing to the k_r^T for **2**, **3**, **4**, and **5**, respectively. In **2** with a horizontal selenium substitution, $\mu_{S_6-S_0}^2 \lambda_6^2$ is small due to the small $\langle S_6 | \mathbf{H}_{SO} | T_1 \rangle^2$ (Fig. S23a). In **3** with the multiple introductions of horizontal substituents, $\langle S_n | \mathbf{H}_{SO} | T_1 \rangle^2$ for each n were increased overall compared with that of **2** (Fig. S23b). Additionally, the $\mu_{S_n-S_0}^2$ for $n = 1, 4, 6$ are large compared with that of **2** due to expansion of π -conjugation horizontally. However, the cooperativity of $\mu_{S_n-S_0}^2$ and $\langle S_n | \mathbf{H}_{SO} | T_1 \rangle^2$ at $n = 5$ where mainly contributes to k_r^T is weak because of the small $\mu_{S_5-S_0}^2$. For multiple vertical substituents, $\langle S_3 | \mathbf{H}_{SO} | T_1 \rangle^2$ of **5** is still large and not drastic decreased compared with $\langle S_5 | \mathbf{H}_{SO} | T_1 \rangle^2$ of **3** (Fig. S23d). Meanwhile, $\mu_{S_3-S_0}^2$ of **5** is larger than $\mu_{S_5-S_0}^2$ of **3**. Thus, quantum chemical calculations confirm the strong cooperation between $\mu_{S_n-S_0}^2$ and $\langle S_n | \mathbf{H}_{SO} | T_1 \rangle^2$ in **5**.

Fig. S23. Histograms of $\mu_{S_n-S_0}^2 \lambda_n^2$ (top), $\langle S_n | H_{SO} | T_1 \rangle^2$ (middle), and $\mu_{S_n-S_0}^2$ (bottom) for each $n = 1$ to 10. **a**, Chromophore 2. **b**, Chromophore 3. **c**, Chromophore 4, **d**, Chromophore 5.”

After revision: Section 5-4 in revised supplementary information

“5-4. Calculated parameters relating to k_r^T (Figure S27)

The calculated values of $\langle S_n | H_{SO} | T_1 \rangle^2 E_{S_n-T_1}^{-2} \mu_{S_n-S_0}^2$ are the largest at $n = 6$, $n = 5$, $n = 7$, and $n = 2$ for **2**, **3**, **4**, and **5**, respectively (Fig. S27a). Accordingly, these n mostly contribute to the first term of equation (4) for chromophores **2–5**, respectively. For the second term of equation (4), the largest value of $\langle T_m | H_{SO} | S_0 \rangle^2 E_{T_m-S_0}^{-2} \mu_{T_m-T_1}^2$ indicate that $m = 9$, $m = 7$, $m = 5$, and $m = 9$ dominantly contribute to k_r^T for **2**, **3**, **4**, and **5**, respectively (Fig. S27b).

Fig. S27. Histograms of calculated parameters relating to k_r^T . **a**, Histograms of $\langle S_n | H_{SO} | T_1 \rangle^2 E_{S_n-T_1}^{-2} \mu_{S_n-S_0}^{-2}$ (top), $\langle S_n | H_{SO} | T_1 \rangle^2$ (middle), and $\mu_{S_n-S_0}^{-2}$ (bottom) for 2–5. **b**, Histograms of $\langle T_m | H_{SO} | S_0 \rangle^2 E_{T_m-S_0}^{-2} \mu_{T_m-T_1}^{-2}$ (top), $\langle T_m | H_{SO} | S_0 \rangle^2$ (middle), and $\mu_{T_m-T_1}^{-2}$ (bottom) for 2–5.”

(Comment 2 from Reviewer 1)

The manuscript attributes the increase in k_r^T to the cooperative interplay between $\mu_{S_n-S_0}$ and $\langle S_n | H_{SO} | T_1 \rangle$. A critical question remains: is the main driving force the intrinsic heavy-atom effect of Se, or the optimization of orbital overlap and spatial distribution of frontier orbitals? From Fig. 3 it is clear that HOMO–LUMO overlap differs significantly between substituent patterns. I recommend further quantitative comparison, for instance by replacing Se with lighter congeners (S or O) to clarify whether the enhanced k_r^T originates predominantly from heavy-atom SOC or orbital overlap effects.

(Response)

Thank you for your critical comments. We consider that the heavy atom effect (HAE) is not the driving force behind the different RTP properties between substituent patterns. The formula for spin-orbit coupling itself expresses the HAE and other effects separately.

However, it may be often intended for mathematicians and physicists, and does not include diagrams or other explanations to make it easy to understand for people in a wide range of fields. Therefore, we first revised it to explain the effects of increasing SOC from three different perspectives: the orbital overlap term, the HAE term, and the orbital orientation term related to the orbital angular momentum vector.

First, we would like to introduce the factors governing magnitude of SOC, using k_{nr}^T as an example. The k_{nr}^T is generally expressed as (*Chem. Phys. Lett.* **16**, 353–358 (1972)),

$$k_{nr}^T = \frac{2\pi}{\hbar} \langle T_1 | \mathbf{H}_{SO} | S_0 \rangle^2 FC, \quad (R1-3)$$

where, FC is the Franck-Condon factor between T_1 and S_0 . The first factor that changes $\langle T_1 | \mathbf{H}_{SO} | S_0 \rangle^2$ is the overlap of the orbitals. $\langle T_1 | \mathbf{H}_{SO} | S_0 \rangle^2$ increases when the overlap of the orbitals involved in the two transition is large (Fig. R4(i)). \mathbf{H}_{SO} is expressed as (*Chem. Rev.* **66**, 199–241 (1966); *J. Phys. Chem. A* **111**, 10490–10499 (2007); *Chem. Rev.* **117**, 6500–6537 (2017)),

$$\mathbf{H}_{SO} = \alpha^2 \sum_A \sum_i \frac{Z_A}{r_{Ai}^3} \vec{l}_{Ai} \vec{s}_i, \quad (R1-4)$$

where α is the fine structure constant, A indicates the nucleus, i indicates the electron, Z is the atomic number, r is the electron–nucleus distance, \vec{l} is the orbital angular momentum, and \vec{s} is the spin angular momentum. The second factor that changes $\langle T_1 | \mathbf{H}_{SO} | S_0 \rangle^2$ is the distance between the heavy atoms and the overlap site of the orbitals involved in the T_1 – S_0 transition. The $\frac{Z_A}{r_{Ai}^3}$ term in equation (R1-4) means that $\langle T_1 | \mathbf{H}_{SO} | S_0 \rangle^2$ increases when the heavy atoms are closer to the overlap site of the orbitals involved in the T_1 – S_0 transition (Fig. R4(ii)). This is the contribution of the increase in $\langle T_1 | \mathbf{H}_{SO} | S_0 \rangle^2$ owing to the introduction of heavier atoms, that is, HAE. The third factor that changes $\langle T_1 | \mathbf{H}_{SO} | S_0 \rangle^2$ is the relative orientation of the molecular orbitals. $\vec{l}_{Ai} \vec{s}_i$ in equation (R1-4) means that $\langle T_1 | \mathbf{H}_{SO} | S_0 \rangle^2$ increases when the axes of the two orbitals involved in the overlapping region point in different directions (Fig. R4(iii)). This

enhancement of the SOC by this term is known as the El-Sayed rule (*J. Chem. Phys.* **38**, 2384–2938 (1963)). Thus, the three factors of the orbital overlap, HAE, and the orbital orientation characteristics govern the magnitude of $\langle T_1 | \mathbf{H}_{SO} | S_0 \rangle^2$.

Fig. R4 Schematic illustration of the parameters affecting $\langle T_1 | \mathbf{H}_{SO} | S_0 \rangle^2$: the (i) orbital overlap, (ii) heavy atom effect (HAE), and (iii) orbital orientation.

Although we mentioned that the orbital orientation feature is an important factor for the SOC strength, the following points should be clarified to avoid potential misinterpretation. Transitions between singlet and triplet states necessarily involve a change in \vec{s} . According to the conservation of total angular momentum, such a spin conversion must be accompanied by the change in \vec{l} (ΔL). The resulting ΔL exerts a torque on the electron, thereby promoting spin flipping. The magnitude of ΔL depends on the vector cross product of the respective \vec{l} (*Acc. Chem. Res.* **55**, 1573–1585 (2022)). Therefore, understanding the direction of the \vec{l} is essential when discussing the electronic transitions that require spin flipping.

The orbital orientation axis is directly related to orientation of \vec{l} . In case of p_z orbital, \vec{l} exhibits an effective component within the xy -plane (Fig. R5a(i)). When the projection component is considered, \vec{l} is oriented perpendicular to the orbital orientation axis (Fig. R5a(ii)). When two orbitals are oriented along the same axis, their corresponding \vec{l} are aligned in the same direction (Fig. R5b(i)). In this case, ΔL becomes

negligible, and no torque is generated to induce spin flip (Fig. R5b(ii)). Conversely, ΔL increases when the two orbitals are oriented in different directions and reaches a maximum when their orbital orientation axes are perpendicular (Fig. R5b(iii)). Under these conditions, the electron experiences a large torque, which strongly promotes the spin-flip process (Fig. R5b(iv)). Therefore, discussing changes in orbital orientation axis is effectively equivalent to discussing the orientation and magnitude of ΔL , providing an intuitive approach for visualizing the magnitude of SOC.

Fig. R5 a, Relationship between (i) atomic orbital and effective component of \vec{l} and (ii) orbital orientation axis and projected component of \vec{l} . **b**, Schematic illustrations of (i) two orbitals have the same orientation axis, (ii) the absence of torque generation for spin flipping, (iii) two orbitals oriented along different axes, and (iv) the electron experiencing a large torque that promotes spin flipping. In panels (ii) and (iv), B_{eff} and μ_s represent the effective magnetic field generated by \vec{l} and the electron spin magnetic momentum, respectively.

The use of the three factors of the orbital overlap, HAE, and the orbital orientation, helps us understand the reason for the significantly suppressed k_{nr}^{T} of **5** compared with that of **3** (energy diagrams in Fig. R6, left). First, in terms of the orbital overlap, different

Fig. R6 a, Molecular orbitals and schematic illustrations related to the (i) orbital overlap, (ii) HAE, and (iii) orbital orientation to explain the smaller k_{nr}^T of **5** than **3**. In (iii), white arrows represent the orbital orientation axes. The isovalues of the HOMO of **3**, LUMO of **3**, and HOMO of **5** are 0.018. The isovalue of 0.0072 is used for the LUMO of **5**. Because the probability of the existence of the LUMO near Se in **5** is extremely small, the isovalue is set small to express the very small existence.

amounts of orbital overlap are confirmed between **3** and **5**. In both **3** and **5**, HOMO and LUMO mainly contribute to the T_1 – S_0 transition. In **3**, HOMO and LUMO largely overlap around Se atom (Fig. R6(i), top). Meanwhile, the much smaller overlap between HOMO and LUMO was confirmed in **5** (Fig. R6(i), bottom). Therefore, the contribution of orbital overlap factor to $\langle T_1 | H_{SO} | S_0 \rangle^2$ is suppressed in **5** compared with **3**. Second, regarding HAE factor, the Se atom is often considered to be the heavy atom source for the HAE and it is widely used for organic phosphorescence materials. However, the difference in the $\langle T_1 | H_{SO} | S_0 \rangle^2$ values between **3** and **5** cannot be well explained by the HAE contribution because both the distance between Se atom and HOMO-LUMO overlap sites does not significantly differ between **3** and **5** (Fig. R6(ii)). Third, regarding the orbital orientation factor that changes $\langle T_1 | H_{SO} | S_0 \rangle$, the orbital orientation on the Se atoms differs between the HOMO and LUMO of **3** (Fig. R6(iii), top). Because the HOMO and LUMO on the Se atoms also exhibit different orientations for **5** (Fig. R6(iii), bottom), the orbital

orientation factor does not well explain the different $\langle T_1 | \mathbf{H}_{SO} | S_0 \rangle^2$ values between **3** and **5**. Therefore, the spatial overlap between the HOMO and LUMO on Se mainly causes the smaller $\langle T_1 | \mathbf{H}_{SO} | S_0 \rangle^2$ of **5** than **3**, suppressing the increase k_{nr}^T for **5** compared with that for **3**. Thus, consideration of the orbital overlap, HAE, and orbital orientation factors provide us with an understanding of the different characteristics of $\langle T_1 | \mathbf{H}_{SO} | S_0 \rangle^2$.

The argument for dividing SOC into three parts, orbital overlap, HAE, and orbital orientation, also worked for the $\langle S_n | \mathbf{H}_{SO} | T_1 \rangle$ related to k_r^T as explained below. First, regarding the orbital overlap factor, magnitude of $\langle S_n | \mathbf{H}_{SO} | T_1 \rangle$ logically increases as the overlap between the orbital overlap density related to the S_n-S_0 transition and the orbital overlap density related to the T_1-S_0 transition increases. For **3**, $\langle S_5 | \mathbf{H}_{SO} | T_1 \rangle$ significantly contributes to the increase in k_r^T , but the overlap density of the S_5-S_0 transition [(HOMO-2)×(LUMO)] only slightly overlaps with that of the T_1-S_0 transition [(HOMO)×(LUMO)] between C and Se (green open circle at the top of Fig. R7(i)). Although $\langle S_2 | \mathbf{H}_{SO} | T_1 \rangle$ significantly contributes to the increase in k_r^T for **5**, similarly, the overlap density of the S_2-S_0 transition [(HOMO-1)×(LUMO)] only has a small overlap with that of the T_1-S_0 transition [(HOMO)×(LUMO)] as a through-space interaction between C and Se (green open circle at the bottom of Fig. R7(i)). Therefore, the comparable poor orbital overlap hardly changes $\langle S_n | \mathbf{H}_{SO} | T_1 \rangle$ between **3** and **5**. Second, regarding the contribution of the HAE to increasing $\langle S_n | \mathbf{H}_{SO} | T_1 \rangle$, the distance between the overlapping site and Se does not significantly change between **3** and **5** (Fig. R7(ii)). Therefore, the effect of the HAE does not result in a driving force that significantly differentiates between $\langle S_5 | \mathbf{H}_{SO} | T_1 \rangle$ of **3** and $\langle S_2 | \mathbf{H}_{SO} | T_1 \rangle$ of **5**. Third, regarding the contribution of the orbital axis orientation to the change in $\langle S_n | \mathbf{H}_{SO} | T_1 \rangle$, for **3**, the orbital axes of the (HOMO-2)×(LUMO) and (HOMO)×(LUMO) are in different directions (Fig. R7(iii), top). Similarly, for **5**, the orbital axis of the through-space type (HOMO-1)×(LUMO) formed between Se and C is different from that of the (HOMO)×(LUMO) on DBC (Fig. R7(iii), bottom). Therefore, the orbital axis orientation hardly changes $\langle S_n | \mathbf{H}_{SO} | T_1 \rangle$ between **3** and **5**. Therefore, these similar tendencies regarding the spatial overlap characteristics, HAE, and orbital orientation increases $\langle S_2 | \mathbf{H}_{SO} | T_1 \rangle^2$ of **5** to a comparable value to $\langle S_5 | \mathbf{H}_{SO} | T_1 \rangle^2$ of **3**.

Fig. R7 Molecular orbitals and schematic illustrations showing that $\langle S_5 | H_{SO} | T_1 \rangle^2$ of **3** (top) and $\langle S_2 | H_{SO} | T_1 \rangle^2$ of **5** (bottom) have comparable magnitudes. Panels (i), (ii), and (iii) represent the orbital overlap, HAE, and orbital orientation effect, respectively. The white arrows in (iii) represent the orbital axis orientations. The isovalues of the overlap density regarding (HOMO-2) \times (LUMO) and (HOMO) \times (LUMO) for **3** and (HOMO-1) \times (LUMO) and (HOMO) \times (LUMO) for **5** are 0.00011.

In addition, the argument for dividing SOC into three parts, orbital overlap, HAE, and orbital orientation, also worked for the $\langle T_m | H_{SO} | S_0 \rangle$ related to k_r^T as following explanation. First, the orbital overlapping feature, the T_7-S_0 transition in **3** is mainly composed of HOMO-3 and LUMO, and the T_9-T_1 transition in **5** is composed of HOMO-2 and LUMO. The spatial overlap between HOMO-3 and LUMO on Se in **3** (green open circles at the top of Fig. R8a(i)) is larger than the through-space overlap density between HOMO-2 and LUMO formed between Se and C in **5** (green open circle at the bottom of Fig. R8(i)). Therefore, the contribution of the orbital overlap term to the enhancement of $\langle T_7 | H_{SO} | S_0 \rangle$ for **3** is larger than that to the enhancement of $\langle T_9 | H_{SO} | S_0 \rangle$ for **5**. Second, in terms of HAE contribution, the distance between the HOMO-3 and LUMO overlap sites and Se for **3** is smaller than that between HOMO-2 and LUMO overlap site and Se for **5** (Fig. R8(ii)). Therefore, for **3**, the HAE plays a role in increasing $\langle T_m | H_{SO} | S_0 \rangle$. Third, in

respect to orbital orientation, the orbital axis directions of the HOMO-3 and LUMO on the Se atoms slightly differ for **3** (Fig. R8(iii), top). However, for **5**, the orbital axis direction of the HOMO-2 of Se significantly differs from that of the LUMO of C on DBC (Fig. R8(iii), bottom). Therefore, the contribution of the orbital orientation term to the enhancement of $\langle T_7 | H_{SO} | S_0 \rangle$ for **3** is smaller than that to the enhancement of $\langle T_9 | H_{SO} | S_0 \rangle$ in **5**. Considering these three factors, the decrease in $\langle T_9 | H_{SO} | S_0 \rangle$ due to the poor orbital overlap and HAE for **5** is compensated for by the increase in $\langle T_9 | H_{SO} | S_0 \rangle$ due to the orbital orientation, which results in $\langle T_9 | H_{SO} | S_0 \rangle^2$ for **5** being equivalent to $\langle T_7 | H_{SO} | S_0 \rangle^2$ for **3**.

Fig. R8 Molecular orbitals and schematic illustrations explaining the comparable magnitudes of $\langle T_7 | H_{SO} | S_0 \rangle^2$ for **3** (top) and $\langle T_9 | H_{SO} | S_0 \rangle^2$ for **5** (bottom). Panels (i), (ii), and (iii) represent the orbital overlap, HAE, and orbital orientation effect, respectively. The white arrows in (iii) represent the orbital axis orientations. The isovalue of 0.013 is used for HOMO-3 and LUMO of **3** and HOMO-2 and LUMO of **5**.

As explained in our response on pages 23 to 25, the factors that change the SOC related to k_{nr}^T can be considered separately as HAE and other two terms. The SOC related to k_r^T can also be considered separately as HAE and other two terms. In vertical hetero-substitutions such as **5**, the overlap of orbitals with two different orientations related to

the T_1 - S_0 transition cannot occur near the heteroatom, making it difficult to enhance $\langle T_1 | H_{SO} | S_0 \rangle^2 \propto k_{nr}^T$. On the other hand, in vertical hetero-substitution, orbitals with two different orientation axes related to $\langle S_n | H_{SO} | T_1 \rangle^2$ and $\langle T_m | H_{SO} | S_0 \rangle^2$ overlap as through-space states near the heteroatom, resulting in a more selective enhancement of $\langle S_n | H_{SO} | T_1 \rangle^2$ and $\langle T_m | H_{SO} | S_0 \rangle^2$ over $\langle T_1 | H_{SO} | S_0 \rangle^2$. Thus, because the overlap of orbitals and the orientation of the orbitals that make up the overlap cause the difference in the increase in k_r^T and k_{nr}^T , the HAE effect does not directly contribute significantly to the increase in k_r^T relative to k_{nr}^T .

However, as Reviewer 1 pointed out, the comparison between Se and S is crucial for evaluating the HAE contribution. Following your suggestion, we synthesized **R1** as a reference chromophore with replacement of Se atoms of **5** with S atoms (Fig. R9, right). The Φ_p of **R1** was 10.4%, which is lower than that of **5** (= 27%). The k_r^T and k_{nr}^T of **R1** were determined to be 0.28 s^{-1} and 2.1 s^{-1} , respectively. Since k_r^T and k_{nr}^T of **5** are 6.1 s^{-1} and 16 s^{-1} , respectively, k_r^T largely decreases compared with k_{nr}^T depending on the change from **5** to **R1** (Fig. R9).

Fig. R9 Chemical structures and optically determined values of k_r^T and k_{nr}^T for **5** (left) and **R1** (right).

The decrease in k_r^T relative to k_{nr}^T of **R1** compared with **5** could be confirmed from the differences in the balance between $\langle T_1 | H_{SO} | S_0 \rangle^2$ and $\langle S_n | H_{SO} | T_1 \rangle^2$. Quantum chemical calculation revealed that $\langle S_3 | H_{SO} | T_1 \rangle^2$ mainly contributes to k_r^T of **R1**. In **R1**,

the $\langle S_3 | \mathbf{H}_{SO} | T_1 \rangle^2$ is slightly smaller than $\langle T_1 | \mathbf{H}_{SO} | S_0 \rangle^2$ (arrow in Fig. R10a), while in **5**, $\langle S_2 | \mathbf{H}_{SO} | T_1 \rangle^2$ is much larger than $\langle T_1 | \mathbf{H}_{SO} | S_0 \rangle^2$ (arrow in Fig. R10b). This calculation results regarding balance of $\langle T_1 | \mathbf{H}_{SO} | S_0 \rangle^2$ and $\langle S_n | \mathbf{H}_{SO} | T_1 \rangle^2$ indicate that some factors are preventing $\langle S_3 | \mathbf{H}_{SO} | T_1 \rangle^2$ from increasing in **R1**.

The perspective of the orbital overlap, HAE, orbital orientation factor provided a reason for the small $\langle S_3 | \mathbf{H}_{SO} | T_1 \rangle^2$ of **R1**. First, regarding orbital overlap as a factor that changing $\langle S_n | \mathbf{H}_{SO} | T_1 \rangle$, for **R1**, the overlap density of the S_3-S_0 transition [(HOMO) \times (LUMO+1)] hardly overlaps with that of the T_1-S_0 transition [(HOMO) \times (LUMO)] at through-space region between S and DBC (green open circle at the top of Fig. R10c(i)). On the other hand, for **5**, the overlap density of the S_2-S_0 transition [(HOMO-1) \times (LUMO)] well overlaps with that of the T_1-S_0 transition [(HOMO) \times (LUMO)] at through-space site between Se and DBC (green open circle at the bottom of Fig. R10c(i)). Therefore, the contribution of the orbital overlapping term to enhancement of $\langle S_3 | \mathbf{H}_{SO} | T_1 \rangle$ for **R1** is significantly smaller than that to enhancement of $\langle S_2 | \mathbf{H}_{SO} | T_1 \rangle$ for **5**. Second, the contribution from HAE, the distance between the overlap site and Se does not significantly change between **R1** and **5** (Fig. R10c(ii)). However, the effect of the HAE on **5** is larger than that on **R1** due to the heavier atoms. Third, regarding orbital orientation factor, for **R1**, the orbital axes of (HOMO) \times (LUMO+1) and (HOMO) \times (LUMO) are in different directions at through-space site (Fig. R10c(iii), top). Similarly, for **5**, the orbital axis of the through-space type (HOMO-1) \times (LUMO) is different from that of the (HOMO) \times (LUMO) on DBC (Fig. R10c(iii), bottom). Therefore, the orbital axis orientation hardly changes $\langle S_n | \mathbf{H}_{SO} | T_1 \rangle$ between **R1** and **5**. When considering these three factors, in **5**, a through-space orbital overlap forms to selectively increases $\langle S_n | \mathbf{H}_{SO} | T_1 \rangle$. Therefore, k_r^T increases relative to k_{nr}^T , acting as the driving force for the enhancement of the phosphorescence yield. On the other hand, in **R1**, the orbital overlap effect to selectively enhance $\langle S_n | \mathbf{H}_{SO} | T_1 \rangle$ is not fully achieved. Therefore, a large $\langle S_3 | \mathbf{H}_{SO} | T_1 \rangle$ is not obtained, and k_r^T does not increase significantly relative to k_{nr}^T . Therefore, the phosphorescence yield does not improve significantly. Thus, the experimental and computational results regarding the replacement of Se with S indicates that the orbital overlap term is a key to maintenance of large $\langle S_n | \mathbf{H}_{SO} | T_1 \rangle$.

Fig. R10 a, Histogram of $\langle T_1 | H_{SO} | S_0 \rangle^2$ and $\langle S_3 | H_{SO} | T_1 \rangle^2$ for **R1**. **b**, Histogram of $\langle T_1 | H_{SO} | S_0 \rangle^2$ and $\langle S_2 | H_{SO} | T_1 \rangle^2$ for **5**. **c**, Molecular orbitals and schematic illustrations related to (i) orbital overlap, (ii) HAE, and (iii) orbital orientation to explain the smaller $\langle S_3 | H_{SO} | T_1 \rangle^2$ of **R1** (top) than $\langle S_2 | H_{SO} | T_1 \rangle^2$ of **5** (bottom). The white arrows in (iii) represent the orbital axes orientations. The isovalues of the overlap density regarding (HOMO) \times (LUMO+1) and (HOMO) \times (LUMO) for **R1** are 0.000055. The isovalue of 0.0001 is used for the overlap density regarding (HOMO-1) \times (LUMO) and (HOMO) \times (LUMO) for **5**. When isovalue = 0.0001 is used for the overlap density regarding (HOMO) \times (LUMO+1) and (HOMO) \times (LUMO) for **R1**, no overlap region appears between Se and C.

From the above, first of all, we added following Fig. 3a in revised manuscript to illustrate the three factors governing SOC.

After revision: Fig. 3a in revised manuscript

Caption of Figure 3a, Schematic illustration of the parameters affecting $\langle T_1 | \mathbf{H}_{SO} | S_0 \rangle^2$: the (i) orbital overlap, (ii) heavy atom effect (HAE), and (iii) orbital orientation.”

Furthermore, we added the sentences in revised manuscript to explain the three factors governing SOC.

After revision: From line 15 in p. 6 to line 8 in p. 8 in revised manuscript

“One reason for the enhancement of k_r^T compared with k_{nr}^T for vertical hetero-substitutions is that increasing the number of the vertical hetero-substitutions makes k_{nr}^T less likely to increase compared with common horizontal hetero-substitutions. k_{nr}^T is generally expressed as^{22,27}

$$k_{nr}^T = \frac{2\pi}{\hbar} \langle T_1 | \mathbf{H}_{SO} | S_0 \rangle^2 FC, \quad (1)$$

where $\langle T_1 | \mathbf{H}_{SO} | S_0 \rangle$ is the spin–orbit coupling (SOC) between T_1 and the ground state (S_0), \mathbf{H}_{SO} is the spin–orbit Hamiltonian, and FC is the Franck–Condon factor between T_1 and S_0 . FC contains the energy gap law, where k_{nr}^T increases as the phosphorescence is red-shifted²⁸. Because the phosphorescence energies of **1–5** are comparable (Fig. 2c), the variation of the k_{nr}^T values among **1–5** is expected to originate from differences in the

$\langle T_1 | \mathbf{H}_{SO} | S_0 \rangle^2$ values. Indeed, quantum chemical calculations showed a good correlation between the optically determined k_{nr}^T values and calculated $\langle T_1 | \mathbf{H}_{SO} | S_0 \rangle^2$ values for **1–5** (supplementary Fig. S24a). Therefore, it is reasonable to attribute the difference in the k_{nr}^T values between **3** and **5** to their respective $\langle T_1 | \mathbf{H}_{SO} | S_0 \rangle^2$ values. The first factor that changes $\langle T_1 | \mathbf{H}_{SO} | S_0 \rangle^2$ is the overlap of the orbitals. $\langle T_1 | \mathbf{H}_{SO} | S_0 \rangle^2$ increases when the overlap of the orbitals involved in the two transitions is large (Fig. 3a(i)). \mathbf{H}_{SO} is expressed as^{6,29,30}

$$\mathbf{H}_{SO} = \alpha^2 \sum_A \sum_i \frac{Z_A}{r_{Ai}^3} \vec{l}_{Ai} \vec{s}_i, \quad (2)$$

where α is the fine structure constant, A indicates the nucleus, i indicates the electron, Z is the atomic number, r is the electron–nucleus distance, \vec{l} is the orbital angular momentum, and \vec{s} is the spin angular momentum. The second factor that changes $\langle T_1 | \mathbf{H}_{SO} | S_0 \rangle^2$ is the distance between the heavy atoms and the overlap site of the orbitals involved in the T_1 – S_0 transition. The $\frac{Z_A}{r_{Ai}^3}$ term in equation (2) means that $\langle T_1 | \mathbf{H}_{SO} | S_0 \rangle^2$ increases when the heavy atoms are closer to the overlap site of the orbitals involved in the T_1 – S_0 transition (Fig. 3a(ii)). This is the contribution of the increase in $\langle T_1 | \mathbf{H}_{SO} | S_0 \rangle^2$ owing to the introduction of heavier atoms, that is, the heavy atom effect (HAE). The third factor that changes $\langle T_1 | \mathbf{H}_{SO} | S_0 \rangle^2$ is the relative orientation of the molecular orbitals. $\vec{l}_{Ai} \vec{s}_i$ in equation (2) means that $\langle T_1 | \mathbf{H}_{SO} | S_0 \rangle^2$ increases when the axes of the two orbitals involved in the overlapping region point in different directions (Fig. 3a(iii)). This enhancement of the SOC by this term is known as the El-Sayed rule³¹. When the two orbital axes are in the same direction, no torque is generated to flip the spin because the electron spin and magnetic moment are parallel (left of Fig. 3a(iii), supplementary Section 5-2, and supplementary Fig. S25). Conversely, when the axes of the two orbitals are in different directions, a torque that flips the spin is generated because the electron spin and magnetic moment are not parallel (Fig. 3a(iii), right)³². Thus, the three factors of the orbital overlap, HAE, and relative orientation of the molecular orbitals can be individually considered to discuss the difference in $\langle T_1 | \mathbf{H}_{SO} | S_0 \rangle^2 \propto k_{nr}^T$ between **3** and **5**.”

Additionally, we added the relationship between orbital orientation and \vec{l} in Section 5-2 in revised Supplementary information.

After revision: Section 5-2 in revised supplementary information

“5-2. Relationship between orbital orientation characteristics and spin flip (Figure

S25)

Transitions between singlet and triplet states necessarily involve a change in the spin angular momentum (\vec{s}). According to the conservation of total angular momentum, such a spin conversion must be accompanied by the change (ΔL) in the orbital angular momentum (\vec{l}). The resulting ΔL exerts a torque on the electron, thereby promoting spin flipping. The magnitude of ΔL depends on the vector cross product of the respective \vec{l} . Therefore, understanding the direction of the \vec{l} is essential when discussing the electronic transitions that require spin flipping.

The orbital orientation axis is directly related to orientation of \vec{l} . In case of p_z orbital, \vec{l} exhibits an effective component within the xy -plane (Fig. S25a(i)). When the projection component is considered, \vec{l} is oriented perpendicular to the orbital orientation axis (Fig. S25a(ii)). When two orbitals are oriented along the same axis, their corresponding \vec{l} are aligned in the same direction (Fig. S25b(i)). In this case, ΔL becomes negligible, and no torque is generated to induce spin flip (Fig. S25b(ii)). Conversely, ΔL increases when the two orbitals are oriented in different directions and reaches a maximum when their orbital orientation axes are perpendicular (Fig. S25b(iii)). Under these conditions, the electron experiences a large torque, which strongly promotes the spin-flip process (Fig. S25b(iv)). Therefore, discussing changes in orbital orientation axis is effectively equivalent to discussing the orientation and magnitude of ΔL , providing an intuitive approach for visualizing the magnitude of SOC.

Fig. S25. Relationship between orbital orientation characteristics and spin flip. **a**, Relationship between (i) atomic orbital and effective component of \vec{l} and (ii) orbital orientation axis and projected component of \vec{l} . **b**, Schematic illustrations of (i) two orbitals having the same orientation axis, (ii) the absence of torque generation for spin flipping, (iii) two orbitals oriented along different axes, and (iv) the electron experiencing a large torque that promotes spin flipping. In panels (ii) and (iv) of **b**, B_{eff} and μ_s represent the effective magnetic field generated by \vec{l} and the electron spin magnetic momentum, respectively.”

Next, we changed Fig. 3b(i) in original manuscript into following Fig. 3b in revised manuscript to explain the differences in $\langle T_1 | H_{\text{SO}} | S_0 \rangle^2$ between **3** and **5** using the orbital overlap, HAE, and the orbital orientation factors.

Before revision: Fig. 3b(i) and (ii) in original manuscript

Caption of Figure 3b, Molecular orbitals and their calculated values relating (i) and (ii) $\langle T_1 | H_{SO} | S_0 \rangle^2$, (iii) and (iv) $\langle S_n | H_{SO} | T_1 \rangle^2$, and (v) and (vi) $\mu_{S_n-S_0}^2$. In (i) and (ii), the red-blue orbitals and orange-light blue orbitals are the highest occupied molecular orbital (HOMO) and lowest unoccupied molecular orbital (LUMO), respectively. In (iii) and (iv), the red-blue orbitals are Ω_{S_3} or Ω_{S_5} and the orange-light blue orbitals are Ω_{T_1} . In (i)–(iv), the white arrows represent the orbital angular momentum (OAM) vectors. The isovalues of the HOMO of **3**, LUMO of **3**, and HOMO of **5** are 0.02. The isovalue of 0.0085 is used for the LUMO of **5**. The isovalues of Ω_{S_3} , Ω_{S_5} , and Ω_{T_1} are 0.00016.”

After revision: Fig. 3b in revised manuscript

Caption of Figure 3b, Molecular orbitals and schematic illustrations related to the (i) orbital overlap, (ii) HAE, and (ii) orbital orientation to explain the smaller k_{nr}^T of **5** than **3**. In (iii) of **a** and **b**, the white arrows represent the orbital orientation axes. The isovalues of the highest occupied molecular orbital (HOMO) of **3**, lowest unoccupied molecular orbital (LUMO) of **3**, and HOMO of **5** are 0.018. The isovalue of 0.0072 is used for the LUMO of **5**. Because the probability of the existence of the LUMO near Se in **5** is extremely small, the isovalue is set small to express the very small existence.”

Additionally, we changed the sentences in revised manuscript to discuss why $\langle T_1 | H_{SO} | S_0 \rangle^2$ of **3** is significantly smaller than that of **5** using the three factors governing SOC.

Before revision: From line 20 in p. 6 to line 7 in p. 7 in original manuscript

“First, compared with the horizontal hetero-substituents, the vertically introduced hetero-substituents suppress $\langle T_1 | H_{SO} | S_0 \rangle^2$ (Fig. 3b, A). In **3**, the highest occupied molecular orbital (HOMO) and lowest unoccupied molecular orbital (LUMO) greatly overlap on the Se atom and have different directions of the orbital angular momentum (OAM) vectors on Se (Fig. 3b(i)). Because the spin-orbit Hamiltonian involves the OAM operator that rotates the vector by 90° ²⁶, $\langle T_1 | H_{SO} | S_0 \rangle^2$ increases when the HOMO and LUMO have large overlap and different directions of the OAM vectors. Therefore, **3** has

large $\langle T_1 | H_{SO} | S_0 \rangle^2$ owing to such molecular orbitals on Se. In **5**, however, the spatial overlap of the HOMO and LUMO on Se is small, although the HOMO and LUMO have different directions of the OAM vectors on Se (Fig. 3b(ii) and supplementary Fig. S22). Owing to the small overlap on Se, $\langle T_1 | H_{SO} | S_0 \rangle^2$ of **5** is significantly smaller than that of **3**.”

After revision: From line 9 in p. 8 to line 4 in p. 9 in revised manuscript

“The suppression of the increase in k_{nr}^T with increasing number of vertical hetero-substitutions is due to the small overlap of the orbitals involved in the T_1-S_0 transition. Regarding the orbital overlap, the first factor that changes $\langle T_1 | H_{SO} | S_0 \rangle^2$, the T_1-S_0 transition is predominantly composed of the highest occupied molecular orbital (HOMO) and lowest unoccupied molecular orbital (LUMO) for both **3** and **5** (supplementary Fig. S26). Because the spatial overlap between the HOMO and LUMO on Se of **5** is much smaller than that of **3** (Fig. 3b(i)), the different amounts of orbital overlap contribute to the different $\langle T_1 | H_{SO} | S_0 \rangle^2$ values between **3** and **5**. Regarding the HAE, the second factor that changes $\langle T_1 | H_{SO} | S_0 \rangle^2$ for **3** and **5**, the Se atom is often considered to be the heavy atom source for the HAE, and it is widely used for organic phosphorescence materials³³⁻³⁵. However, the large difference in the $\langle T_1 | H_{SO} | S_0 \rangle^2$ values between **3** and **5** cannot be well explained by the HAE contribution because the distance between the HOMO and LUMO overlap sites and Se does not significantly differ between **3** and **5** (Fig. 3b(ii)). Regarding the orbital orientation factor, the third factor that changes $\langle T_1 | H_{SO} | S_0 \rangle^2$, the orbital orientation on the Se atoms differs between the HOMO and LUMO of **3** (Fig. 3b(iii), left). Because the HOMO and LUMO on the Se atoms also exhibit different orientations for **5** (Fig. 3b(iii), right), the orbital orientation factor does not well explain the different $\langle T_1 | H_{SO} | S_0 \rangle^2$ values between **3** and **5**. Therefore, the spatial overlap between the HOMO and LUMO on Se mainly causes the smaller $\langle T_1 | H_{SO} | S_0 \rangle^2$ of **5** than **3**, suppressing the increase of k_{nr}^T for **5** compared with that for **3**.”

In addition, we changed the calculation result in revised supplementary information regarding the molecular orbitals relating to $\langle T_1 | H_{SO} | S_0 \rangle^2$ because of the changes in the calculation methods.

Before revision: Section 5-2 in original supplementary information

“5-2. Molecular orbitals relating to $\langle T_1 | H_{SO} | S_0 \rangle^2$ ”

The T_1-S_0 transition of **3** and **5** are mainly constructed from HOMO and LUMO (Figs. S22a and S22b). Because **3** and **5** have similar nature in both HOMO and LUMO at DBC unit, the orbital’s nature around a Se atom were focused to explain the large difference in

$\langle T_1 | H_{SO} | S_0 \rangle^2$ between **3** and **5**. In **3**, HOMO and LUMO delocalize to Se atom (Fig. S22c). Similarly, HOMO of **5** is delocalized on Se atoms. In LUMO of **5**, however, a significantly small electron density at Se atom was observed and could not be confirmed using isovalue of more than 0.01 (Fig. S22d). Therefore, the spatial overlap between HOMO and LUMO of **5** is significantly small. Although the direction of the orbital angular momentum (OAM) vectors of HOMO and LUMO are comparable in both **3** and **5**, the $\langle T_1 | H_{SO} | S_0 \rangle^2$ of **5** becomes smaller than that of **3** due to the small overlap between HOMO and LUMO as shown in (i) and (ii) of Fig. 3b in the main text.

Fig. S22. Differences in molecular orbitals relating to $\langle T_1 | H_{SO} | S_0 \rangle^2$. **a**, HOMO and LUMO of **3**. **b**, HOMO and LUMO of **5**. **c**, HOMO (left) and LUMO (right) around a Se atom of **3**. **d**, HOMO (left) and LUMO (right) around a Se atom of **5**. In **a**) and **b**), isovalue of 0.02 is used for HOMO and LUMO.”

After revision: Section 5-3 in revised supplementary information

“5-3. Molecular orbitals relating to $\langle T_1 | H_{SO} | S_0 \rangle^2$ (Figure S26)

The T_1-S_0 transition of **3** and **5** are mainly constructed from HOMO and LUMO (Fig. S26a, top and S26b, top). Because **3** and **5** have similar nature in both the HOMO and LUMO at DBC unit, the orbital’s nature around the Se atom were focused to explain the large difference in $\langle T_1 | H_{SO} | S_0 \rangle^2$ between **3** and **5**. For **3**, HOMO and LUMO delocalize to Se atom (Fig. S26a, bottom). Similarly, HOMO of **5** is delocalized on Se atom. However, in LUMO of **5**, a significantly small electron density at Se atom was observed and could not be confirmed using isovalue of more than 0.008 (Fig. S26b, bottom).

Fig. S26. Molecular orbitals relating to $\langle T_1 | H_{SO} | S_0 \rangle^2$ for **3** and **5**. **a**, HOMO and LUMO of **3**. **b**, HOMO and LUMO of **5**.”

Next, we changed Fig. 3b(iii) and (iv) in original manuscript into Fig. 5 in revised manuscript to add the T_m contribution and to discuss the $\langle S_n | H_{SO} | T_1 \rangle^2$ and $\langle T_m | H_{SO} | S_0 \rangle^2$ using the orbital overlap, HAE, and the orbital orientation factors.

Before revision: Fig. 3b(iii) and (iv) in original manuscript

Caption of Figure 3b, Molecular orbitals and their calculated values relating (i) and (ii) $\langle T_1 | H_{SO} | S_0 \rangle^2$, (iii) and (iv) $\langle S_n | H_{SO} | T_1 \rangle^2$, and (v) and (vi) $\mu_{S_n-S_0}^2$. In (i) and (ii), the red-blue orbitals and orange-light blue orbitals are the highest occupied molecular orbital (HOMO) and lowest unoccupied molecular orbital (LUMO), respectively. In (iii) and (iv), the red-blue orbitals are Ω_{S3} or Ω_{S5} and the orange-light blue orbitals are Ω_{T1} . In (i)–(iv), the white arrows represent the orbital angular momentum (OAM) vectors. The isovalues of the HOMO of **3**, LUMO of **3**, and HOMO of **5** are 0.02. The isovalue of 0.0085 is used for the LUMO of **5**. The isovalues of Ω_{S3} , Ω_{S5} , and Ω_{T1} are 0.00016.”

After revision: Fig. 5 in revised manuscript

Fig. 5 Difference in the factors governing $\langle S_n | H_{SO} | T_1 \rangle^2$ and $\langle T_m | H_{SO} | S_0 \rangle^2$ between horizontal and vertical hetero-substituents. **a**, Simplified expression of the k_r^T equation. **b**, Molecular orbitals and schematic illustrations showing that $\langle S_5 | H_{SO} | T_1 \rangle^2$ of **3** (top) and $\langle S_2 | H_{SO} | T_1 \rangle^2$ of **5** (bottom) have comparable magnitudes. The isovalues of the overlap density regarding (HOMO-2)×(LUMO) and (HOMO)×(LUMO) for **3** and (HOMO-1)×(LUMO) and (HOMO)×(LUMO) for **5** are 0.00011. **c**, Molecular orbitals and schematic illustrations explaining the comparable magnitudes of $\langle T_7 | H_{SO} | S_0 \rangle^2$ for **3** (top) and $\langle T_9 | H_{SO} | S_0 \rangle^2$ for **5** (bottom). The isovalue of 0.013 is used for HOMO-3 and LUMO of **3** and HOMO-2 and LUMO of **5**. In **b** and **c**, panels (i), (ii), and (iii) represent the orbital overlap, HAE, and orbital orientation effect, respectively. The white arrows in (iii) represent the orbital axis orientations.”

Additionally, we changed the sentences in revised manuscript to discuss why $\langle S_n | H_{SO} | T_1 \rangle^2$ and $\langle T_m | H_{SO} | S_0 \rangle^2$ of **3** were maintained up to that of **5** using the three factors governing SOC.

Before revision: Lines 8–18 in p. 7 in original manuscript

“Next, multiple vertically introduced hetero-substituents maintain $\langle S_n | H_{SO} | T_1 \rangle^2$, enhancing k_r^T to a comparable magnitude to $\langle T_1 | H_{SO} | S_0 \rangle^2$ (Fig. 3b, B). In **3** and **5**, $n = 5$

and $n = 3$ mainly enhance k_r^T , respectively (supplementary Fig. S23). In **3**, $\langle S_5 | \mathbf{H}_{SO} | T_1 \rangle^2$ for the driving force of k_r^T greatly decreases compared with the large $\langle T_1 | \mathbf{H}_{SO} | S_0 \rangle^2$ (Fig. 3b(iii)). In contrast, $\langle S_3 | \mathbf{H}_{SO} | T_1 \rangle^2$ for the driving force of k_r^T increases compared with the small $\langle T_1 | \mathbf{H}_{SO} | S_0 \rangle^2$ in **5**. In **5**, the overlap density for the S_3 – S_0 transition (\mathbf{Q}_{S3}) composed of HOMO-1 and LUMO forms a through-space interaction between Se and C of DBC (Fig. 3b(iv), green open circle, and supplementary Fig. S24b). Because this partially overlaps with the overlap density for the T_1 – S_0 transition (\mathbf{Q}_{T1}) of C on DBC, the spatial overlap with different OAM directions for \mathbf{Q}_{S3} and \mathbf{Q}_{T1} between the Se and C atoms results in the sufficient $\langle S_3 | \mathbf{H}_{SO} | T_1 \rangle^2$ in **5** (Fig. 3b(iv)).”

After revision: From line 21 in p. 10 to last line in p. 12 in revised manuscript

“ $\langle S_n | \mathbf{H}_{SO} | T_1 \rangle^2$ of the multiple vertical hetero-substituted chromophore approaching that of the multiple horizontal hetero-substituted chromophore can be understood by considering the following three factors, which are the same factors as those explained the SOC regarding k_{nr}^T in Fig. 3a. First, regarding the overlap of the orbitals before and after the transition that increases $\langle S_n | \mathbf{H}_{SO} | T_1 \rangle$ in equation (4), $\langle S_n | \mathbf{H}_{SO} | T_1 \rangle$ logically increases as the overlap between the orbital overlap density related to the S_n – S_0 transition and the orbital overlap density related to the T_1 – S_0 transition increases. For **3**, $\langle S_5 | \mathbf{H}_{SO} | T_1 \rangle$ significantly contributes to the increase in k_r^T , but the overlap density of the S_5 – S_0 transition [(HOMO-2)×(LUMO)] only slightly overlaps with that of the T_1 – S_0 transition [(HOMO)×(LUMO)] between C and Se (green open circles at the top of Fig. 5b(i) and supplementary Fig. S32a). Although $\langle S_2 | \mathbf{H}_{SO} | T_1 \rangle$ significantly contributes to the increase in k_r^T for **5**, similarly, the overlap density of the S_2 – S_0 transition [(HOMO-1)×(LUMO)] only has a small overlap with that of the T_1 – S_0 transition [(HOMO)×(LUMO)] as a through-space interaction between C and Se (green open circle at the bottom of Fig. 5b(i) and supplementary Fig. S32b). Therefore, the comparable poor orbital overlap hardly changes $\langle S_n | \mathbf{H}_{SO} | T_1 \rangle$ between **3** and **5**. Second, regarding the contribution of the HAE to increasing $\langle S_n | \mathbf{H}_{SO} | T_1 \rangle$ in equation (4), the distance between the overlapping site and Se does not significantly change between **3** and **5** (Fig. 5b(ii)). Therefore, the effect of the HAE does not result in a driving force that significantly differentiates between $\langle S_5 | \mathbf{H}_{SO} | T_1 \rangle$ of **3** and $\langle S_2 | \mathbf{H}_{SO} | T_1 \rangle$ of **5**. Third, regarding the contribution of the orbital axis orientation to the change in $\langle S_n | \mathbf{H}_{SO} | T_1 \rangle$, for **3**, the orbital axes of (HOMO-2)×(LUMO) and (HOMO)×(LUMO) are in different directions (Fig. 5b(iii), top). Similarly, for **5**, the orbital axis of the through-space type (HOMO-1)×(LUMO) formed between Se and C is different from that of the (HOMO)×(LUMO) on DBC (Fig. 5b(iii), bottom). Therefore, the orbital axis orientation hardly changes $\langle S_n | \mathbf{H}_{SO} | T_1 \rangle$ between **3** and **5**. Therefore, these similar tendencies regarding the spatial overlap characteristics,

HAE, and orbital orientation increases $\langle S_2 | \mathbf{H}_{SO} | T_1 \rangle^2$ of **5** to a comparable value to $\langle S_5 | \mathbf{H}_{SO} | T_1 \rangle^2$ of **3**. When the Se atoms in **5** are replaced with S atoms, the enhancement of $\langle S_n | \mathbf{H}_{SO} | T_1 \rangle^2$ becomes small owing to the lack of an orbital overlap factor (supplementary Figs. S13 and S33–S35, and Table S4).

The approach of $\langle T_m | \mathbf{H}_{SO} | S_0 \rangle$ of the multiple vertical hetero-substituted chromophore to that of the multiple horizontal hetero-substituted chromophore can also be understood from the three perspectives of the orbital overlap, HAE effect, and orbital orientation (Fig. 5c). Regarding the orbital overlap (Fig. 5c(i)), the T_7 – S_0 transition in **3** is mainly composed of HOMO-3 and LUMO (supplementary Fig. S36a), and the T_9 – T_1 transition in **5** is composed of HOMO-2 and LUMO (supplementary Fig. S36b). The spatial overlap density between HOMO-3 and LUMO on Se in **3** (green open circles at the top of Fig. 5c(i)) is larger than the through-space overlap density between HOMO-2 and LUMO formed between Se and C in **5** (green open circle at the bottom of Fig. 5c(i)). Therefore, the contribution of the orbital overlap term to the enhancement of $\langle T_7 | \mathbf{H}_{SO} | S_0 \rangle$ for **3** is larger than that to the enhancement of $\langle T_9 | \mathbf{H}_{SO} | S_0 \rangle$ for **5**. Regarding the contribution from the HAE, the distance between the HOMO-3 and LUMO overlap sites and Se for **3** is smaller than that for **5** (Fig. 5c(ii)). Therefore, for **3**, the HAE plays a role in increasing $\langle T_m | \mathbf{H}_{SO} | S_0 \rangle$. Regarding the orbital orientation (Fig. 5c(iii)), the orbital axis directions of the HOMO-3 and LUMO on the Se atoms slightly differ for **3** (Fig. 5c(iii), top). However, for **5**, the orbital axis direction of the HOMO-2 of Se significantly differs from that of the LUMO of C on DBC (Fig. 5c(iii), bottom). Therefore, the contribution of the orbital orientation term to the enhancement of $\langle T_7 | \mathbf{H}_{SO} | S_0 \rangle$ for **3** is smaller than that to the enhancement of $\langle T_9 | \mathbf{H}_{SO} | S_0 \rangle$ for **5**. Considering these three factors, the decrease in $\langle T_9 | \mathbf{H}_{SO} | S_0 \rangle$ due to the poor orbital overlap and HAE for **5** is compensated for by the increase in $\langle T_9 | \mathbf{H}_{SO} | S_0 \rangle$ due to the orbital orientation, which results in $\langle T_9 | \mathbf{H}_{SO} | S_0 \rangle$ for **5** being equivalent to $\langle T_7 | \mathbf{H}_{SO} | S_0 \rangle$ for **3**.”

In addition, we changed the calculation result regarding the molecular orbitals relating to $\langle S_n | \mathbf{H}_{SO} | T_1 \rangle^2$ in revised supplementary information according to the changes in the calculation methods.

Before revision: Section 5-4 in original supplementary information

“5-4. Molecular orbitals relating to k_r^T ”

In **3**, the S_5 – S_0 transition which mainly contributes to k_r^T is constructed from HOMO-2 and LUMO (Fig. S24a(i)). The HOMO-2 at Se atom and LUMO at C on DBC partially overlaps (Fig. S24a(ii), green open circle), resulting in the formation of trough-bond interactions at Se-C bond site in the overlap density of HOMO-2 and LUMO for S_5 – S_0

transition (Ω_{S5}) (Fig. S24a(iii), green open circle). In **5**, HOMO-1 and LUMO mainly constitute the S_3 - S_0 transition (Fig. S24b(i)). The HOMO-1 of Se site and LUMO of C on DBC partially overlap (Fig. S24b(ii), green open circle) and form the through-space interaction in the overlap density of HOMO-1 and LUMO for S_3 - S_0 transition (Ω_{S3}) (Fig. S24b(ii), green open circle).

Fig. S24. Molecular orbitals regarding through-bond and through-space interactions. **a**, Molecular orbitals relating to the (i) and (ii) S_5 - S_0 transition and (iii) Ω_{S5} of **3**. **b**, Molecular orbitals relating to the (i) and (ii) S_3 - S_0 transition and (iii) Ω_{S3} of **5**. The isovalue of 0.015 is used for HOMO-2, HOMO-1, and LUMO. The isovalue of Ω_{S5} and Ω_{S3} are 0.00016.”

After revision: Section 5-7 in revised supplementary information

“**5-7. Molecular orbitals relating to $\langle S_n | H_{SO} | T_1 \rangle^2$ (Figure S32)**

Fig. S32. Molecular orbitals relating to $\langle S_n | H_{SO} | T_1 \rangle^2$. **a**, Molecular orbitals involved in the $S_5 - S_0$ transition (top) and the $T_1 - S_0$ transition (bottom) of **3**. **b**, Molecular orbitals involved in the $S_2 - S_0$ transition (top) and the $T_1 - S_0$ transition (bottom) of **5**. The isovalue of 0.015 is used for HOMO-2, HOMO, and LUMO for **3** and HOMO-1, HOMO, and LUMO for **5**. The isovalues of the overlap density regarding $(\text{HOMO-2}) \times (\text{LUMO})$ and $(\text{HOMO}) \times (\text{LUMO})$ for **3** and $(\text{HOMO-1}) \times (\text{LUMO})$ and $(\text{HOMO}) \times (\text{LUMO})$ for **5** are 0.00011.”

Furthermore, we added the molecular orbitals relating to $\langle T_m | H_{SO} | S_0 \rangle^2$ in Section 5-9 in revised supplementary information.

After revision: Section 5-9 in revised supplementary information

“5-9. Molecular orbitals relating to $\langle T_m | H_{SO} | S_0 \rangle^2$ (Figure S36)

Fig. S36. Molecular orbitals relating to $\langle T_m | H_{SO} | S_0 \rangle^2$. **a**, Molecular orbitals involved in the T_7-S_0 transition of **3**. **b**, Molecular orbitals involved in the T_9-S_0 transition of **5**. Isovalue of 0.013 is used for HOMO-3 and LUMO for **3** and HOMO-2 and LUMO for **5**.”

Next, we added the synthesis procedure and characterization data of **R1** in Section 1 in revised supplementary information.

After revision: Section 1 in revised supplementary information

“2,7,10,15-tetrakis(8-(phenylthio)naphthalen-1-yl)dibenzo[*g,p*]chrysene (R1):

2,7,10,15-Tetrakis(8-bromonaphthalen-1-yl)dibenzo[*g,p*]chrysene (139.1 mg, 0.121 mmol), diphenyl disulfide (117.6 mg, 0.539 mmol), [1,1'-bis(diphenylphosphino)ferrocene]dichloropalladium(II) (12.3 mg, 0.0168 mmol), and Zinc (41.2 mg, 0.630 mmol) in anhydrous tetrahydrofuran (4.0 mL) was stirred at a reflux temperature for 24 h under a nitrogen atmosphere. After completion of the reaction, dichloromethane was added to the reaction solution and washed with brine three times. The organic phase was dried over anhydrous Na_2SO_4 , evaporated under a reduced pressure, purified via column chromatography (silica gel; dichloromethane/hexane (35:65 v/v) as the eluent) to obtain **R1** as pale-yellow powder (79.6 mg, 51.9%). ^1H NMR

(acetone-*d*₆, 500 MHz): δ = 8.83–8.71 (m, 4H), 8.66 (s, 1H), 8.38–8.33 (m, 2H), 8.09–7.96 (m, 8H), 7.70–7.54 (m, 14H), 7.50–7.39 (m, 7H), 7.10–6.82 (m, 15H), 6.74–6.67 (m, 5H) (Fig. S13a). ¹³C NMR (CDCl₃, 125 MHz): δ = 141.43, 141.35, 140.55, 140.48, 140.42, 140.34, 140.29, 138.73, 138.48, 138.14, 137.91, 137.83, 137.68, 135.53, 135.27, 134.78, 134.37, 134.15, 133.96, 133.86, 133.58, 133.41, 133.18, 133.12, 133.02, 132.61, 132.49, 132.06, 131.83, 131.68, 131.53, 131.38, 131.30, 131.22, 131.17, 131.12, 131.09, 131.06, 130.55, 130.50, 130.46, 130.39, 130.28, 129.96, 129.84, 129.58, 129.49, 129.14, 129.09, 128.93, 128.90, 128.86, 128.76, 128.64, 128.55, 128.51, 128.39, 128.32, 128.09, 127.94, 127.84, 127.76, 127.62, 127.53, 127.48, 126.69, 126.54, 126.34, 125.95, 125.85, 125.83, 125.78, 125.16, 124.71, 124.66, 124.59 (Fig. S13b). HRMS-MALDI (m/Z): [M]⁺ calcd. for C₉₀H₅₆S₄, 1264.325; Found 1264.326 (Fig. S13c).

Fig. S13. Characterization data of **R1**. **a**, ¹H NMR spectrum in acetone-*d*₆. **b**, ¹³C NMR spectrum in CDCl₃. **c**, HRMS-MALDI spectrum.”

Additionally, we added the calculation result and photophysical properties of **R1** in section 5-8 in revised supplementary information.

After revision: Section 5-8 in revised supplementary information

“5-8. Calculation results and photophysical properties of chromophore **R1** (Figures S33–S35 and Table S4)

To evaluate the HAE contribution, chromophore **R1** in which Se atoms in **5** are replaced with S atoms was synthesized (Fig. S33a). The **R1** was doped in the amorphous β -estradiol at a concentration with 0.3 wt%. The amorphous film of **R1**-doped β -estradiol showed blue fluorescence under excitation and red RTP after ceasing excitation (Fig. S33b). The Φ_p was measured to be 10.4%, which is lower than **5** ($\Phi_p = 27.0\%$). The τ_p of **R1** was determined to be 416 ms from RTP decay characteristics (Fig. S33c). From the equations $\Phi_p = \Phi_{isc}k_r^T\tau_p$, $\tau_p = 1/(k_r^T + k_{nr}^T)$, and $\Phi_{isc} = 1 - \Phi_f$, the values of k_r^T and k_{nr}^T for **R1** were determined to be 0.28 s^{-1} and 2.1 s^{-1} , respectively (Table S4). Since k_r^T and k_{nr}^T of **5** are 6.1 s^{-1} and 16 s^{-1} , respectively, k_r^T largely decreases compared with k_{nr}^T depending on the change from **5** to **R1** (Table S4).

Fig. S33. Optical properties of **R1** in amorphous β -estradiol. **a**, Chemical structure of **R1**. **b**, Emission spectrum under excitation (top) and RTP spectrum soon after ceasing excitation (bottom) of 0.3 wt% **R1**-doped amorphous β -estradiol film. **c**, RTP decay characteristics of **R1** in amorphous β -estradiol.

Table S4. Comparison of photophysical parameters relating RTP between **5** and **R1**.

Chromophores ^{a)}	Φ_f	$\Phi_{isc}^{b)}$	Φ_p	τ_p	k_r^T	k_{nr}^T
	[%]	[%]	[%]	[ms]	[s^{-1}]	[s^{-1}]
5	1.0	99.0	27.0	44.8	6.1	16
R1	11.8	88.2	10.4	416	0.28	2.1

^{a)}The concentration of chromophores was 0.3 wt% in amorphous β -estradiol. ^{b)} Values determined from $1 - \Phi_f$.

The decrease in k_r^T relative to k_{nr}^T of **R1** compared with **5** could be confirmed from the differences in the balance between $\langle T_1 | \mathbf{H}_{SO} | S_0 \rangle^2$ and $\langle S_n | \mathbf{H}_{SO} | T_1 \rangle^2$. Quantum chemical calculation confirmed that $\langle S_3 | \mathbf{H}_{SO} | T_1 \rangle^2$ mainly contributes to k_r^T of **R1** (Fig. S34a). In **R1**, the $\langle S_3 | \mathbf{H}_{SO} | T_1 \rangle^2$ is slightly smaller than $\langle T_1 | \mathbf{H}_{SO} | S_0 \rangle^2$ (arrow of left graph in Fig. S34b), while in **5**, $\langle S_2 | \mathbf{H}_{SO} | T_1 \rangle^2$ is much larger than $\langle T_1 | \mathbf{H}_{SO} | S_0 \rangle^2$ (arrow of right graph in Fig. S34b). This calculation results regarding balance of $\langle T_1 | \mathbf{H}_{SO} | S_0 \rangle^2$ and $\langle S_n | \mathbf{H}_{SO} | T_1 \rangle^2$ indicate that some factors are preventing $\langle S_3 | \mathbf{H}_{SO} | T_1 \rangle^2$ from increasing in **R1**.

Fig. S34. Calculated parameters relating k_r^T for **R1**. **a**, Histograms of calculated parameters of $\langle S_n | \mathbf{H}_{SO} | T_1 \rangle^2 E_{S_n-T_1}^{-2} \mu_{S_n-S_0}^2$ (top), $\langle S_n | \mathbf{H}_{SO} | T_1 \rangle^2$ (middle), and $\mu_{S_n-S_0}^2$ (bottom) for **R1**. **b**, Histograms of the $\langle T_1 | \mathbf{H}_{SO} | S_0 \rangle^2$ and $\langle S_3 | \mathbf{H}_{SO} | T_1 \rangle^2$ values for **R1** (left) and $\langle T_1 | \mathbf{H}_{SO} | S_0 \rangle^2$ and $\langle S_2 | \mathbf{H}_{SO} | T_1 \rangle^2$ values for **5**.

The perspective of the orbital overlap, HAE, orbital orientation factor provided a reason for the small $\langle S_3 | \mathbf{H}_{SO} | T_1 \rangle^2$ of **R1**. First, regarding orbital overlap as a factor that changing $\langle S_n | \mathbf{H}_{SO} | T_1 \rangle$, for **R1**, the overlap density of the S_3-S_0 transition [(HOMO)×(LUMO+1)] hardly overlaps with that of T_1-S_0 transition [(HOMO)×(LUMO)] at through-space region between S and DBC (green open circle at the top of Fig. S35(i)). On the other hand, for **5**, the overlap density of the S_2-S_0 transition [(HOMO-1)×(LUMO)] well overlaps with that of the T_1-S_0 transition [(HOMO)×(LUMO)] at through-space site between Se and DBC (green open circle at the bottom of Fig. S35(i)). Therefore, the contribution of the orbital overlapping term to enhancement of $\langle S_3 | \mathbf{H}_{SO} | T_1 \rangle$ for **R1** is significantly smaller than that to enhancement of $\langle S_2 | \mathbf{H}_{SO} | T_1 \rangle$ in **5**. Second, the contribution from HAE, the distance between the overlap site and Se atom does not significantly change between **R1** and **5** (Fig. S35(ii)). However, the effect of HAE on **5** is larger than that on **R1** due to the heavier atoms. Third, regarding

orbital orientation factor, for **R1**, the orbital axis of (HOMO) \times (LUMO+1) and (HOMO) \times (LUMO) are in different directions at through-space site (Fig. S35(iii), top). Similarly, for **5**, the orbital axis of the through-space type (HOMO-1) \times (LUMO) is different from that of the (HOMO) \times (LUMO) on DBC (Fig. S35(iii), bottom). Therefore, the orbital axis orientation hardly changes $\langle S_n | \mathbf{H}_{SO} | T_1 \rangle$ between **R1** and **5**. When considering these three factors, in **5**, a through-space orbital overlap forms to selectively increase $\langle S_n | \mathbf{H}_{SO} | T_1 \rangle$. Therefore, k_r^T increases relative to k_{nr}^T , acting as the driving force for the enhancement of the phosphorescence yield. On the other hand, in **R1**, the orbital overlap effect to selectively enhance $\langle S_n | \mathbf{H}_{SO} | T_1 \rangle$ is not fully achieved. Therefore, a large $\langle S_3 | \mathbf{H}_{SO} | T_1 \rangle$ is not obtained, and k_r^T does not increase significantly relative to k_{nr}^T . Therefore, the phosphorescence yield does not improve significantly. Thus, the experimental and computational results regarding the replacement of Se with S indicates that the orbital overlap term is a key to maintenance of large $\langle S_n | \mathbf{H}_{SO} | T_1 \rangle$.

Fig. S35. Molecular orbitals and schematic illustrations related to (i) orbital overlap, (ii) HAE, and (iii) orbital orientation to explain the smaller $\langle S_3 | \mathbf{H}_{SO} | T_1 \rangle^2$ of **R1** (top) compared with $\langle S_2 | \mathbf{H}_{SO} | T_1 \rangle^2$ of **5** (bottom). The white arrows in (iii) represent the orbital axes orientations. The isovalues of the overlap density regarding (HOMO) \times (LUMO+1) and (HOMO) \times (LUMO) for **R1** is 0.000055. The iso-value of 0.0001 is used for the overlap density regarding (HOMO-1) \times (LUMO) and (HOMO) \times (LUMO) for **5**. When iso-value = 0.0001 is used for the overlap density regarding (HOMO) \times (LUMO+1) and (HOMO) \times (LUMO) for **R1**, no overlap region appears between Se and C.”

(Comment 3 from Reviewer 1)

The vertical substitution strategy may be conceptually related to the “intramolecular external heavy atom effect” (EHE), where the heavy atom is not directly conjugated with the emissive core but exerts SOC through spatial proximity. It would be valuable for the authors to elaborate on this analogy and position their design within the broader context of heavy-atom effect literature.

(Response)

Thank you for your suggestion. In previous report, the enhancement of SOC through the external HAE has not been well discussed in terms of the orbital overlap and the orbital orientation factors (*Chem. Sci.* **8**, 6060–6065 (2017); *Adv. Funct. Mater.* **28**, 1705045 (2018)). Although several reports on internal HAE have addressed the role of orbital orientation factor (*Chem. Sci.* **13**, 789–797 (2022)), the SOC contributions have not been distinguished into k_r^T and k_{nr}^T . In this study, as mentioned in our previous response, the enhanced RTP observed in the vertically hetero-substituted system was attributed to the suppressed $\langle T_1 | \mathbf{H}_{SO} | S_0 \rangle$ and to the maintained $\langle S_n | \mathbf{H}_{SO} | T_1 \rangle$ and $\langle T_m | \mathbf{H}_{SO} | S_0 \rangle$. The former arises from the small orbital overlap associated with the T_1 – S_0 transition. The latter is achieved not only by HAE but also by favorable orbital overlap and orbital orientation factors. We note that we consider this HAE in vertical hetero-substituents is not external HAE but internal HAE because the orbital nature on Se directly relates to the $\langle T_1 | \mathbf{H}_{SO} | S_0 \rangle$, $\langle S_n | \mathbf{H}_{SO} | T_1 \rangle$, and $\langle T_m | \mathbf{H}_{SO} | S_0 \rangle$. However, as Reviewer 1 pointed out, evaluating the role of the intramolecular external HAE and the contribution of spatial proximity is essential for fully clarifying the origin of the enhanced phosphorescence in the vertically substituted system.

To clearly clarify the contribution of the intramolecular external HAE and spatial proximity, we synthesized a reference chromophore **R2** in which a phenylseleno substituent is horizontally introduced at the 3-position of the DBC core (Fig. R11a). In **R2**, Se atom is not directly conjugated with emissive core because of no electron density at 3-position of DBC. Therefore, **R2** has a potential characteristic of the intramolecular external HAE. Additionally, the distance between Se and the 3-position of DBC is 1.94 Å in T_1 optimized geometry (Fig. R11b). Because the distance between Se and C of DBC for **4** (3.19 Å) and **5** (3.05 Å) are more separated spatially, the intramolecular external

HAE may play significantly in **R2**. The Φ_p of **R2** was measured to be 11.5%, which is lower than that of **4** (= 17.6%) and **5** (= 27.0%). The k_r^T and k_{nr}^T of **R2** were determined to be 1.2 s^{-1} and 9.5 s^{-1} , respectively. Compared with **4** and **5**, the small enhancement of k_r^T relative to k_{nr}^T was observed in **R2** from **1** (green arrow in Fig. R11c). The small enhancement of k_r^T indicates that the intramolecular external HAE is very weak or the other factors are unfavorable in **R2**.

Fig. R11 **a**, Chemical structure of **R2**. **b**, Distance between Se atom and 3-position of DBC in T_1 optimized geometry. **c**, Relationship between optically determined k_r^T and k_{nr}^T among **1**, **4**, **5**, and **R2**.

The perspective of the orbital overlap factor that changes SOC strength provided a rational reason for the small enhancement of k_r^T of **R2**. Quantum chemical calculation confirmed that $n = 6$ is mainly contribute to k_r^T of **R2** among $\langle S_n | H_{SO} | T_1 \rangle$ term. The overlap density of the overlap density of the S_6-S_0 transition [(HOMO-2) \times (LUMO)] hardly overlaps with that of T_1-S_0 transition [(HOMO) \times (LUMO)] (Fig. R12). This lack of spatial overlap does not allow enhancement of $\langle S_6 | H_{SO} | T_1 \rangle$ in **R2**. Indeed, $\langle S_6 | H_{SO} | T_1 \rangle^2$ of **R2** is 6.97 cm^{-2} , which is lower than $\langle S_7 | H_{SO} | T_1 \rangle^2$ of **4** (14.1 cm^{-2}), and $\langle S_2 | H_{SO} | T_1 \rangle^2$ of **5** (361 cm^{-2}). Although intramolecular external HAE may contribute to $\langle S_n | H_{SO} | T_1 \rangle$ of **R2** to some degree, the resulting $\langle S_6 | H_{SO} | T_1 \rangle^2$ of **R2** does not reach to $\langle S_7 | H_{SO} | T_1 \rangle^2$ of **4** and $\langle S_2 | H_{SO} | T_1 \rangle^2$ of **5**. Therefore, the intramolecular external HAE makes only a minor contribution to the enhancement of $\langle S_n | H_{SO} | T_1 \rangle$. In contrast, as observed in vertical hetero substituents, modulation of the orbital overlap contribution is more crucial than simply reducing the spatial distance between heavy atom and the emissive core to obtain the large SOC.

Fig. R12 Molecular orbitals and schematic illustrations related to orbital overlap to explain small $\langle S_6 | H_{SO} | T_1 \rangle^2$ of **R2**. The isovalues of the overlap density regarding (HOMO-2) \times (LUMO) and (HOMO) \times (LUMO) are 0.00016.

From the above, we added the following sentence regarding properties of **R2** in revised manuscript.

After revision: Lines 10–12 in p. 10 in revised manuscript

“This characteristic was also confirmed for chromophores with a horizontal phenylseleno substituent at the 3-position of DBC, and Φ_p of the chromophore was indeed small (supplementary Figs. S14 and S29–S31, and Table S3).”

Additionally, we added the synthesis procedure and characterization data of **R2** in Section 1 in revised supplementary information.

After revision: Section 1 in revised supplementary information

“Dibenzo[*g,p*]chrysene-3-yl(phenyl)selane (R2):

9-Fluorenone (1.35 g, 7.50 mmol), 2-bromo-9-fluorenone (648 mg, 2.50 mmol), zinc (4.21 g, 64.4 mmol), and zinc chloride (2.73 g, 20.0 mmol) in tetrahydrofuran (25 ml) and pure water (25 ml) was stirred at room temperature for 4 h. After completion of the reaction, dichloromethane was added to the reaction solution and filtrated to remove the undissolved solid. Pure water was added to the solution and extracted with

dichloromethane. The organic layer was dried over anhydrous Na_2SO_4 and evaporated under reduced pressure to give a white powder (2.09 g). The powder (2.09 g) in acetic acid (35.5 ml) and sulfuric acid (1.8 ml) was heated at 85 °C for 4 h. After cooling to room temperature, dichloromethane was added to the reaction mixture and washed with pure water and extracted with dichloromethane. The organic layer was dried over anhydrous Na_2SO_4 and concentrated under reduced pressure. The residue was purified by column chromatography (silica gel; dichloromethane: hexane = 3: 7) to give $\text{C}_{26}\text{H}_{16}\text{BrO}$ (430 mg) as a white powder. Although the products of $\text{C}_{26}\text{H}_{16}\text{BrO}$ had potentially three kinds of isomers, the products were directly used for the next reaction without further purification. $\text{C}_{26}\text{H}_{16}\text{BrO}$ (430 mg, 1.02 mmol), sodium borohydride (67.3 mg, 1.78 mmol) in anhydrous tetrahydrofuran (15 ml) and methanol (15 ml) was stirred at 60 °C for 3 h under a nitrogen atmosphere. After completion of the reaction, dichloromethane was added to the solution and washed with pure water. The organic layer was dried over anhydrous Na_2SO_4 and concentrated under reduced pressure to give a white powder (391 mg). The powder (391 mg) in acetic acid (28 ml) and sulfuric acid (0.70 ml) was heated at 85 °C for 4 h. After cooling to room temperature, pure water was added to the reaction solution and extracted with dichloromethane. The extracted organic layer was dried over anhydrous Na_2SO_4 and evaporated under reduced pressure. The residue was purified by column chromatography (silica gel; dichloromethane: hexane = 1: 9) to yield 3-bromodibenzo[*g,p*]chrysene (212 mg) as a white powder. The reaction yield of 3-bromodibenzo[*g,p*]chrysene from 2-bromo-9-fluorenone was 20.8%. 3-Bromodibenzo[*g,p*]chrysene (142 mg, 0.349 mmol), diphenyl diselenide (172 mg, 0.552 mmol), [1,1'-bis(diphenylphosphino)ferrocene]palladium(II) dichloride (40.4 mg, 0.0552 mmol), and zinc (72.2 mg, 1.10 mmol) in anhydrous tetrahydrofuran (11 ml) was stirred at reflux temperature for 16 h under a nitrogen atmosphere. After completion of the reaction, chloroform was added to the reaction solution. The organic phase was washed with pure water three times, dried over anhydrous Na_2SO_4 , and concentrated under the reduced pressure to give a crude product. The crude product was purified by column chromatography (silica gel; dichloromethane: hexane = 12: 88) to give **R2** as a pale-yellow powder (73.7 mg, 43.7%). Chromophore **R2** was purified further by sublimation. ^1H NMR (CDCl_3 , 500 MHz): δ = 8.71 (sd, J = 1.5 Hz, 1H), 8.68–8.63 (m, 5H), 8.58 (d, J = 8.5 Hz, 1H), 8.36 (dd, J = 8.0, 1.0 Hz, 1H), 7.75 (dd, J = 8.5, 1.5 Hz, 1H), 7.69–7.57 (m, 7H), 7.45 (ddd, J = 7.5, 6.9, 1.4 Hz, 1H), 7.33–7.29 (m, 3H) (Fig. S14a). ^{13}C NMR (CDCl_3 , 125 MHz): δ = 133.70, 132.69, 131.01, 130.97, 130.78, 130.54, 130.36, 130.15, 129.99, 129.78, 129.51, 129.22, 129.06, 128.98, 128.91, 128.89, 128.47, 128.06, 127.65, 126.76, 126.75, 126.71, 126.69, 126.58, 126.56, 124.38, 123.61, 123.52, 123.45 (Fig.

S14b). HRMS-MALDI (m/z): $[M]^+$ calcd. for $C_{32}H_{20}Se$, 484.0730; found 484.0729 (Fig. S14c). Anal. Calcd. for $C_{32}H_{20}Se$: C, 79.50; H, 4.17; Se, 16.33. Found C, 79.62; H, 4.26.

Fig. S14. Characterization data of **R2**. **a**, 1H NMR spectrum in $CDCl_3$. **b**, ^{13}C NMR spectrum in $CDCl_3$. **c**, HRMS-MALDI spectrum.”

Next, we added the optical properties and calculation results of **R2** in Section 5-6 in revised supplementary information.

After revision: Section 5-6 in revised supplementary information

“5-6. Calculation results and photophysical properties of chromophore R2 (Figures S29–S31 and Table S3)

To clearly clarify the contribution of the intramolecular external heavy atom effect (HAE) derived from spatial proximity between heavy atom and central π -core, chromophore **R2** in which a phenylseleno substituent is horizontally introduced at the 3-position of the DBC core was synthesized (Fig. S29a). In **R2**, Se atom is not directly conjugated with emissive core because of no electron density at 3-position of DBC. Therefore, **R2** has a potential characteristic of the intramolecular external HAE. Additionally, the distance between Se and the 3-position of DBC is 1.94 Å in T₁ optimized geometry (Fig. S29b). Because the distance between Se and C of DBC for **4** (3.19 Å) and **5** (3.05 Å) are more separated spatially, the intramolecular external HAE may play significantly in **R2**.

The **R2** was doped in the amorphous β -estradiol at a concentration with 0.3 wt%. The amorphous film of **R2**-doped β -estradiol showed blue fluorescence under excitation and red RTP after ceasing excitation (Fig. S29c and Fig. S29d). The Φ_p of **R2** was measured to be 11.5%, which is lower than that of **4** (= 17.6%) and **5** (= 27.0%). The τ_p of **R2** was determined to be 92.9 ms from RTP decay characteristics (Fig. S29e). From the equations $\Phi_p = \Phi_{isc}k_r^T\tau_p$, $\tau_p = 1/(k_r^T + k_{nr}^T)$, and $\Phi_{isc} = 1 - \Phi_f$, the values of k_r^T and k_{nr}^T of **R2** were determined to be 1.2 s⁻¹ and 9.5 s⁻¹, respectively (Table S3). Compared with **4** and **5**, the small enhancement of k_r^T relative to k_{nr}^T was observed in **R2** from **1** (green arrow in Fig. S29f). The small enhancement of k_r^T relative to k_{nr}^T indicates that the contribution of spatial proximity between Se and DBC core to k_r^T enhancement is weak.

Fig. S29. Photophysical characteristics of **R2**. **a**, Chemical structure of **R2**. **b**, Distance between Se atom and C atom at 3-position of DBC. **c**, Emission spectrum of **R2** in amorphous β -estradiol under excitation. **d**, RTP spectrum of **R2** in amorphous β -estradiol soon after ceasing excitation. **e**, RTP decay characteristics of **R2** in amorphous β -estradiol. **f**, Relationship between optically determined k_r^T and k_{nr}^T among **1**, **4**, **5**, and **R2**.

Table S3. Photophysical parameters of **R2**.

Chromophores ^{a)}	Φ_f	$\Phi_{isc}^{b)}$	Φ_p	τ_p	k_r^T	k_{nr}^T
	[%]	[%]	[%]	[ms]	[s ⁻¹]	[s ⁻¹]
R2	0.66	99.3	11.5	92.9	1.2	9.5

^{a)}The concentration of chromophores was 0.3 wt% in amorphous β -estradiol. ^{b)} Values determined from $1 - \Phi_f$.

The relatively small enhancement of k_r^T compared with k_{nr}^T in **R2** could be confirmed from the different contributions of SOC to k_r^T and k_{nr}^T . Quantum chemical calculation confirmed that $n = 6$ is mostly contribute to k_r^T of **R2** (Fig. S30a). In **R2**, the $\langle S_6 | H_{SO} | T_1 \rangle^2$ that enhances k_r^T is smaller than $\langle T_1 | H_{SO} | S_0 \rangle^2 \propto k_{nr}^T$ (arrow in Fig. S30b).

Fig. S30. Calculated parameters relating to k_r^T in **R2**. **a**, Histograms of calculated parameters of $\langle S_n | H_{SO} | T_1 \rangle^2 E_{S_n - T_1}^{-2} \mu_{S_n - S_0}^2$ (top), $\langle S_n | H_{SO} | T_1 \rangle^2$ (middle), and $\mu_{S_n - S_0}^2$ (bottom) for **R2**. **b**, Histogram of the $\langle T_1 | H_{SO} | S_0 \rangle^2$ and $\langle S_6 | H_{SO} | T_1 \rangle^2$ values for **R2**.

The perspective of the orbital overlap factor that changes SOC strength provided a rational reason for the small enhancement of $\langle S_n | H_{SO} | T_1 \rangle$ of **R2**. The $S_6 - S_0$ transition of **R2** consists primarily of two components with 28.4% and 28.3% contributions, respectively. The former is composed of HOMO-2 and LUMO. The latter is composed of HOMO and LUMO+2. The overlap density between HOMO-2 and LUMO for $S_6 - S_0$ transition [(HOMO-2) \times (LUMO)] hardly overlaps with that between HOMO and LUMO for $T_1 - S_0$ transition [(HOMO) \times (LUMO)] (Fig. S31, left). Similarly, no spatial overlap was observed between the overlap density of HOMO and LUMO+2 for $S_6 - S_0$ transition [(HOMO) \times (LUMO+2)] and the (HOMO) \times (LUMO) for $T_1 - S_0$ transition (Fig. S31, right). These lack of spatial overlap does not allow enhancement of $\langle S_6 | H_{SO} | T_1 \rangle$ in **R2**. Indeed, $\langle S_6 | H_{SO} | T_1 \rangle^2$ of **R2** is 6.97 cm^{-2} , which is much lower than $\langle S_7 | H_{SO} | T_1 \rangle^2$ of **4** (14.1 cm^{-2}), and $\langle S_2 | H_{SO} | T_1 \rangle^2$ of **5** (361 cm^{-2}). Therefore, simply placing the heavy atom in close proximity to the central π -core is insufficient. The orbital overlap selectively working on $\langle S_n | H_{SO} | T_1 \rangle$, which is realized for **5**, is a key molecular design strategy for achieving a large k_r^T .

Fig. S31. Molecular orbitals and schematic illustrations related to orbital overlap to explain small $\langle S_6 | H_{SO} | T_1 \rangle^2$ of **R2**. The isovalues of the overlap density regarding (HOMO-2) \times (LUMO), (HOMO) \times (LUMO), and (HOMO) \times (LUMO+2) are 0.00016.”

(Comment 4 from Reviewer 1)

The definition of “cooperative effect” should be more precisely articulated—whether it refers to a simple multiplicative relationship between $\mu_{S_n-S_0}$ and $\langle S_n | H_{SO} | T_1 \rangle$, or if energy matching and symmetry considerations are also involved.

(Response)

Thank you for your comment. From equation (R1-2), first term involves the $E_{S_n-T_1}$ contribution and second term involves $E_{T_m-S_0}$ (Fig. R13a). Therefore, the energy term also relates to k_r^T . In the revised manuscript, the calculation results were analysed taking into account of $E_{S_n-T_1}$ and $E_{T_m-S_0}$ contributions. For instance, in the contribution in first term of equation (R1-2), the product of $\frac{\langle S_n | H_{SO} | T_1 \rangle^2}{E_{S_n-T_1}^2} \mu_{S_n-S_0}^2$ is the largest at $n = 2$ for **5** (Fig. R13b(i)). Compared with the differences in the magnitude of $\langle S_n | H_{SO} | T_1 \rangle^2$ (Fig. R13b(ii)) and $\mu_{S_n-S_0}^2$ among $n = 1$ to $n = 10$ (Fig. R13b(iii)), the $E_{S_n-T_1}^2$ does not drastically change among $n = 1$ to $n = 10$ (Fig. R13b(iv)). This indicates that contribution of the $E_{S_n-T_1}$ to the k_r^T is relatively minor. Such a trend was observed in the second term of equation (R1-2) (Fig. R13c(i)–(iv)). Therefore, the energy term is not included in the definition of cooperative effect in the manuscript.

Fig. R13 a, Equations of k_r^T . **b**, Histograms of (i) $\frac{\langle S_n | \mathbf{H}_{SO} | T_1 \rangle^2}{E_{S_n-T_1}^2} \mu_{S_n-S_0}^2$, (ii) $\langle S_n | \mathbf{H}_{SO} | T_1 \rangle^2$, (iii) $\mu_{S_n-S_0}^2$, and (iv) $E_{S_n-T_1}^2$ for each n . **c**, Histograms of (i) $\frac{\langle T_m | \mathbf{H}_{SO} | S_0 \rangle^2}{E_{T_m-S_0}^2} \mu_{T_m-T_1}^2$, (ii) $\langle T_m | \mathbf{H}_{SO} | S_0 \rangle^2$, (iii) $\mu_{T_m-T_1}^2$, and (iv) $E_{T_m-S_0}^2$ for each m .

For the symmetry consideration, the symmetry nature at S_n-S_0 transition and T_m-T_1 transition intrinsically relates to the cooperativity between spin-orbit coupling and transition dipoles. In **3**, as we mentioned in our response to first comment from Reviewer 1, S_5-S_0 and T_7-T_1 transitions are symmetrically forbidden along x axis that long π -conjugation direction extended by horizontal substituents (Fig. R14a). The symmetrical nature does not allow the large $\mu_{S_5-S_0}^2$ and $\mu_{T_7-T_1}^2$, resulting in the weak cooperativity between SOC and TDM in **3**. On the other hand, **5** has symmetry-allowed transition along long conjugated axis in both S_2-S_0 and T_9-T_1 transitions (Fig. R14b). These symmetrical features increase $\mu_{S_2-S_0}^2$ and $\mu_{T_9-T_1}^2$ of **5** relative to $\mu_{S_5-S_0}^2$ and $\mu_{T_7-T_1}^2$ of **3**, which promoting stronger cooperation between spin-orbit coupling and transition dipoles.

Fig. R14 a, Molecular orbitals relating S_5-S_0 transition (left) and T_7-T_1 transition (right) in **3**. **b**, Molecular orbitals relating S_2-S_0 transition (left) and T_9-T_1 transition in **5**. In **a** and **b**, (HOMO-2) \times (LUMO) for **3** and (HOMO-1) \times (LUMO) for **5** are calculated by the ADF2025 package using the PBE0 functional and TZP basis set. The molecular orbitals of β -LUMO and β -HOMO-2 for **3** and β -LUMO and β -HOMO-1 for **5** were calculated with the Gaussian09 program using the PBE/PBE functional and 6-311G(d,p) basis set. The isovalue of 0.00016 is used for (HOMO-2) \times (LUMO) in **3** and (HOMO-1) \times (LUMO) in **5**. The β -HOMO-2 and β -LUMO of **3** and β -HOMO-1 and β -LUMO of **5** are shown with an isovalue of 0.01.

From the above, we added Fig. R13 in Section 5-5 in revised supplementary information to clarify the contribution of energy term to k_r^T .

After revision: Section 5-5 in revised supplementary information

“5-5. Contribution of $E_{S_n-S_0}$ and $E_{T_m-S_0}$ to k_r^T (Figure S28)

From equation (4), first term involves the $E_{S_n-T_1}$ contribution and second term involves $E_{T_m-S_0}$ (Fig. S28a). Therefore, the energy term also relates to k_r^T . In the contribution in

first term of equation (4), the product of $\frac{(S_n|H_{SO}|T_1)^2}{E_{S_n-T_1}^2} \mu_{S_n-S_0}^2$ is the largest at $n = 2$ for

5 (Fig. S28b(i)). However, compared with the differences in the magnitude of $\langle S_n | \mathbf{H}_{SO} | T_1 \rangle^2$ (Fig. S28b(ii)) and $\mu_{S_n-S_0}^2$ among $n = 1$ to $n = 10$ (Fig. S28b(iii)), the $E_{S_n-T_1}^2$ does not drastically change among $n = 1$ to $n = 10$ (Fig. S28b(iv)). This indicates that contribution of the $E_{S_n-T_1}$ to the k_r^T is relatively minor. Such a trend was observed in the second term of equation (4) (Fig. S28c(i)–(iv)). Therefore, the energy term is not included in the definition of cooperative effect.

Fig. S28. Parameters affecting k_r^T . **a**, Equations of k_r^T . **b**, Histograms of (i) $\frac{\langle S_n | \mathbf{H}_{SO} | T_1 \rangle^2}{E_{S_n-T_1}^2} \mu_{S_n-S_0}^2$, (ii) $\langle S_n | \mathbf{H}_{SO} | T_1 \rangle^2$, (iii) $\mu_{S_n-S_0}^2$, and (iv) $E_{S_n-T_1}^2$ for each n . **c**, Histograms of (i) $\frac{\langle T_m | \mathbf{H}_{SO} | S_0 \rangle^2}{E_{T_m-S_0}^2} \mu_{T_m-T_1}^2$, (ii) $\langle T_m | \mathbf{H}_{SO} | S_0 \rangle^2$, (iii) $\mu_{T_m-T_1}^2$, and (iv) $E_{T_m-S_0}^2$ for each m .

(Comment 5 from Reviewer 1)

The discussion would benefit from a clearer comparison with previous “horizontal substitution” strategies, highlighting why they fail to enhance RTP while the vertical substitution succeeds.

(Response)

Thank you for your comment. Conventional horizontal substituent strategies increase k_r^T , but they also induce a much larger rise in k_{nr}^T . This occurs because the spatial overlap factor of SOC enhances not only $\langle S_n | \mathbf{H}_{SO} | T_1 \rangle$ and $\langle T_m | \mathbf{H}_{SO} | S_0 \rangle$, but also $\langle T_1 | \mathbf{H}_{SO} | S_0 \rangle$ simultaneously. In contrast, the vertical substituent strategy maintains a large k_r^T without significant increase in k_{nr}^T . This is because the orbital overlap factor of SOC does not specifically contribute to $\langle T_1 | \mathbf{H}_{SO} | S_0 \rangle$. Furthermore, increasing the number of vertical substituents produces the strong cooperativity between $\langle S_n | \mathbf{H}_{SO} | T_1 \rangle^2$ and TDM as well as $\langle T_m | \mathbf{H}_{SO} | S_0 \rangle^2$ and TDM for enhanced phosphorescence. In previous strategies featuring horizontal hetero-substitutions, the extended π -conjugation could not be effectively utilized for the triplet radiation. The vertical hetero-substituents strategy can activate the long π conjugation system for triplet radiation, which serves as a promising design guideline for organic phosphorescent molecules.

Accordingly, we added the sentences in discussion section in revised manuscript.

After revision: Lines 1–6 in p. 15 in revised manuscript

“this study demonstrates that vertically introduced through-space substituents selectively enhance $\langle S_n | \mathbf{H}_{SO} | T_1 \rangle^2$ and $\langle T_m | \mathbf{H}_{SO} | S_0 \rangle^2$, and they are not directly related to $\langle S_1 | \mathbf{H}_{SO} | T_m \rangle^2 \propto \Phi_{isc}$. Additionally, increasing the number of vertical hetero-substituents generates strong cooperativity between $\langle S_n | \mathbf{H}_{SO} | T_1 \rangle^2$ and the TDM, as well as between $\langle T_m | \mathbf{H}_{SO} | S_0 \rangle^2$ and the TDM, for enhanced phosphorescence.”

Response to Reviewer 2

(Comments from Reviewer 2):

In this paper, the author presents that enhanced phosphorescence was achieved by adjusting vertical hetero-substituents into π -conjugated structures. Multiple hetero-substituents vertical to the π plane allow for the effective use of the transition dipole extended to the π plane to cooperate with the spin-orbit coupling to enhance the phosphorescence yield. Considering these thorough results and the potential importance of this research, I believe this work should be published with improved organization and a clear summary.

(Response)

We sincerely appreciate your constructive comments. We have addressed your comments, as outlined in the following responses. Below, our response to the comments from Reviewer 2 is in blue, and the specific text and supporting information that we revised are in red.

(Comment 1 from Reviewer 2)

The phosphorescence radiative rate (k_r) and lifetime (τ_P) are intrinsically governed by the energy gap (ΔE_{ST}) between the triplet state (T_1) and the ground state (S_0), as well as the magnitude of the transition dipole moment for the $T_1 \rightarrow S_0$ transition. A larger transition dipole moment and a smaller ΔE_{ST} generally favor a higher k_r , leading to a shorter τ_P . This fundamental relationship should be explicitly discussed in the manuscript to provide a more complete photophysical rationale for the observed phosphorescence performance.

(Response)

Thank you for your suggestion. As Reviewer 2 pointed out, k_r^T and τ_p are governed by the energy gap between T_1 and S_0 ($E_{T_1-S_0}$). However, smaller $E_{T_1-S_0}$ theoretically causes a not a higher but a lower k_r^T . This is because k_r^T based on the first order perturbation is expressed as (*Chem. Rev.* **66**, 199–241 (1966); *J. Phys. Chem. A* **111**, 10490–10499 (2007); *Chem. Rev.* **117**, 6500–6537 (2017)),

$$k_r^T = \frac{E_{T_1-S_0}^3}{3\varepsilon_0\pi\hbar^4c^3} |\mu_{T_1-S_0}|^2, \quad (\text{R2-1})$$

$$\mu_{T_1-S_0} = \sum_{n \geq 1} \frac{\langle S_n | \mathbf{H}_{SO} | T_1 \rangle}{E_{S_n-T_1}} \mu_{S_n-S_0} + \sum_{m \geq 2} \frac{\langle T_m | \mathbf{H}_{SO} | S_0 \rangle}{E_{T_m-S_0}} \mu_{T_m-T_1}, \quad (\text{R2-2})$$

where ε_0 is the vacuum permittivity, \hbar is the Dirac constant, c is the speed of light, $\mu_{T_1-S_0}$ is the transition dipole moment (TDM) between T_1 and S_0 , $\langle S_n | \mathbf{H}_{SO} | T_1 \rangle$ is the spin-orbit coupling (SOC) between the n th-order singlet excited state (S_n , $n \geq 1$) and T_1 , $E_{S_n-T_1}$ is the energy gap between S_n and T_1 , $\mu_{S_n-S_0}$ is the TDM between S_n and S_0 , $\langle T_m | \mathbf{H}_{SO} | S_0 \rangle$ is the SOC between the m th-order triplet excited state (T_m , $m \geq 2$) and S_0 , $E_{T_m-S_0}$ is the energy gap between T_m and S_0 , $\mu_{T_m-T_1}$ is the TDM between T_m and T_1 . These parameters are visualized in Fig. R15a. Equation (R2-1) indicates that k_r^T becomes smaller as decreasing in $E_{T_1-S_0}$. On the other hand, the rate constant of nonradiative transition from T_1 (k_{nr}^T) become larger as decreasing in $E_{T_1-S_0}$. k_{nr}^T is generally expressed as (*Chem. Phys. Lett.* **16**, 353–358 (1972)),

$$k_{nr}^T = \frac{2\pi}{\hbar} \langle T_1 | \mathbf{H}_{SO} | S_0 \rangle^2 FC, \quad (\text{R2-3})$$

where $\langle T_1 | \mathbf{H}_{SO} | S_0 \rangle$ is the SOC between T_1 and S_0 , \mathbf{H}_{SO} is the spin-orbit Hamiltonian, and FC is the Franck–Condon factor between T_1 and S_0 . FC contains the energy gap law, where k_{nr}^T increases as the phosphorescence is red-shifted. Because τ_p is expressed as $\tau_p = 1 / (k_r^T + k_{nr}^T)$, increase in k_{nr}^T is responsible for the shorter τ_p in longer-wavelength region. However, phosphorescence energies of chromophores **1–5** are comparable, the $E_{T_1-S_0}$ term in equations (R2-1) and (R2-3) hardly affect to the difference in the τ_p among **1–5** (Fig. R15b).

Fig. R15 a, Schematic energy diagram illustrating the key factors contributing to k_r^T . **b**, Equations of k_r^T and k_{nr}^T (top) and RTP spectra of 1–5 in amorphous β -estradiol (bottom). The insert photograph was captured after ceasing excitation.

Accordingly, first of all, we added the equation of k_r^T and sentences regarding the relationship between k_r^T and phosphorescence energy in revised manuscript.

After revision: From line 9 in p. 9 to line 1 in p. 10 in revised manuscript

“The equation for k_r^T based on first-order perturbation theory is expressed as^{6,29,30}

$$k_r^T = \frac{E_{T_1-S_0}^3}{3\varepsilon_0\pi\hbar^4c^3} |\mu_{T_1-S_0}|^2, \quad (3)$$

$$\mu_{T_1-S_0} = \sum_{n \geq 1} \frac{\langle S_n | \mathbf{H}_{SO} | T_1 \rangle}{E_{S_n-T_1}} \mu_{S_n-S_0} + \sum_{m \geq 2} \frac{\langle T_m | \mathbf{H}_{SO} | S_0 \rangle}{E_{T_m-S_0}} \mu_{T_m-T_1}, \quad (4)$$

where $E_{T_1-S_0}$ is the energy gap between T_1 and S_0 , ε_0 is the vacuum permittivity, \hbar is the Dirac constant, c is the speed of light, $\mu_{T_1-S_0}$ is the TDM between T_1 and S_0 , $\langle S_n | \mathbf{H}_{SO} | T_1 \rangle$ is the SOC between the n th-order singlet excited state (S_n , $n \geq 1$) and T_1 , $E_{S_n-T_1}$ is the energy gap between S_n and T_1 , $\mu_{S_n-S_0}$ is the TDM between S_n and S_0 , $\langle T_m | \mathbf{H}_{SO} | S_0 \rangle$ is the SOC between the m th-order triplet excited state (T_m , $m \geq 2$) and S_0 , $E_{T_m-S_0}$ is the energy gap between T_m and S_0 , and $\mu_{T_m-T_1}$ is the TDM between T_m and T_1 . Equation (3) indicates that the enhancement of k_r^T becomes increasingly difficult as the phosphorescence is red-shifted. From equation (4), however, k_r^T can still be facilitated by the cooperative

contributions of $\langle S_n | \mathbf{H}_{SO} | T_1 \rangle^2$ and $\mu_{S_n-S_0}^2$, and of $\langle T_m | \mathbf{H}_{SO} | S_0 \rangle^2$ and $\mu_{T_m-T_1}^2$ even in the long-wavelength region (Fig. 4a).”

Next, we added the equation of k_{nr}^T and sentences regarding the relationship between k_{nr}^T and phosphorescence energy in revised manuscript.

After revision: From line 17 in p. 6 to line 3 in p. 7 in revised manuscript

“ k_{nr}^T is generally expressed as^{22,27}

$$k_{nr}^T = \frac{2\pi}{\hbar} \langle T_1 | \mathbf{H}_{SO} | S_0 \rangle^2 FC, \quad (1)$$

where $\langle T_1 | \mathbf{H}_{SO} | S_0 \rangle$ is the spin-orbit coupling (SOC) between T_1 and the ground state (S_0), \mathbf{H}_{SO} is the spin-orbit Hamiltonian, and FC is the Franck-Condon factor between T_1 and S_0 . FC contains the energy gap law, where k_{nr}^T increases as the phosphorescence is red-shifted²⁸. Because the phosphorescence energies of **1–5** are comparable (Fig. 2c), the variation of the k_{nr}^T values among **1–5** is expected to originate from differences in the $\langle T_1 | \mathbf{H}_{SO} | S_0 \rangle^2$ values.”

(Comment 2 from Reviewer 2)

The superior performance of compound **5** against compound **3** is likely attributable to a stronger through-space spin-orbit coupling effect based on the heavy atom. This enhancement is probably due to the optimized spatial proximity and orientation of the heavy selenium (Se) atom relative to the central π -core, allowing its potent heavy-atom effect to more effectively perturb the spin densities and facilitate the $T_1 \rightarrow S_0$ transition. It seems that explanations regarding this aspect are not mentioned in the article.

(Response)

We are sorry that we did not well explain about the SOC in original manuscript. As Reviewer 2 pointed out, through-space SOC effect is largely related to enhanced phosphorescence in the vertical hetero-substituents. As advised by Reviewer 2, we did not explain what factors change the SOC in the original manuscript. Therefore, we first revised it to explain the effects of increasing the SOC separately from three perspectives: the orbital overlap term, the heavy atom effect (HAE) term, and the orbital orientation term related to the orbital angular momentum vector.

In this paragraph, we explain the contents of the SOC related to k_{nr}^T . From equation (R2-3), k_{nr}^T is proportional to the $\langle T_1 | \mathbf{H}_{SO} | S_0 \rangle^2$. Regarding the first factor that changes $\langle T_1 | \mathbf{H}_{SO} | S_0 \rangle^2$, $\langle T_1 | \mathbf{H}_{SO} | S_0 \rangle^2$ becomes larger when the overlap of the orbitals involved in the two transitions is large (Fig. R16(i)) because the SOC element contains the integration of two wavefunctions involved in the T_1 – S_0 transition. In $\langle T_1 | \mathbf{H}_{SO} | S_0 \rangle^2$, \mathbf{H}_{SO} is expressed as (*Chem. Rev.* **66**, 199–241 (1966); *J. Phys. Chem. A* **111**, 10490–10499 (2007); *Chem. Rev.* **117**, 6500–6537 (2017)),

$$\mathbf{H}_{SO} = \alpha^2 \sum_A \sum_i \frac{Z_A}{r_{Ai}^3} \vec{l}_{Ai} \vec{s}_i, \quad (\text{R2-4})$$

where α is the fine structure constant, A indicates the nucleus, i indicates the electron, Z is the atomic number, r is the electron-nucleus distance, \vec{l} is the orbital angular momentum, \vec{s} is the spin angular momentum. The second factor that changes $\langle T_1 | \mathbf{H}_{SO} | S_0 \rangle^2$ is the distance between the heavy atoms and the overlap site of the orbitals involved in the T_1 – S_0 transition. The $\frac{Z_A}{r_{Ai}^3}$ term in equation (R1-4) means that $\langle T_1 | \mathbf{H}_{SO} | S_0 \rangle^2$ increases when the heavy atoms are closer to the overlap site of the orbitals involved in the T_1 – S_0 transition (Fig. R16(ii)). This is the contribution of the increase in $\langle T_1 | \mathbf{H}_{SO} | S_0 \rangle^2$ owing to the introduction of heavier atoms, that is, HAE. The third factor that changes $\langle T_1 | \mathbf{H}_{SO} | S_0 \rangle^2$ is the relative orientation of the molecular orbitals. $\vec{l}_{Ai} \vec{s}_i$ in equation (R2-4) means that $\langle T_1 | \mathbf{H}_{SO} | S_0 \rangle^2$ increases when the axes of the two orbitals involved in the overlapping region point in different directions (Fig. R16(iii)). This enhancement of the SOC by this term is known as the El-Sayed rule (*J. Chem. Phys.* **38**, 2384–2938 (1963)). Thus, the three factors of the orbital overlap, HAE, and the orbital orientation characteristics govern the magnitude of $\langle T_1 | \mathbf{H}_{SO} | S_0 \rangle^2$.

Fig. R16 Schematic illustration of the parameters affecting $\langle T_1 | \mathbf{H}_{SO} | S_0 \rangle^2$: the (i) orbital overlap, (ii) heavy atom effect (HAE), and (iii) orbital orientation.

Although we mentioned that the orbital orientation feature is an important factor for the SOC strength, the following points should be clarified to avoid potential misinterpretation. Transitions between singlet and triplet states necessarily involve a change in \vec{s} . According to the conservation of total angular momentum, such a spin conversion must be accompanied by the change in \vec{l} (ΔL). The resulting ΔL exerts a torque on the electron, thereby promoting spin flipping. The magnitude of ΔL depends on the vector cross product of the respective \vec{l} (*Acc. Chem. Res.* **55**, 1573–1585 (2022)). Therefore, understanding the direction of the \vec{l} is essential when discussing the electronic transitions that require spin flipping.

The orbital orientation axis is directly related to orientation of \vec{l} . In case of p_z orbital, \vec{l} exhibits an effective component within the xy -plane (Fig. R17a(i)). When the projection component is considered, \vec{l} is oriented perpendicular to the orbital orientation axis (Fig. R17a(ii)). When two orbitals are oriented along the same axis, their corresponding \vec{l} are aligned in the same direction (Fig. R17b(i)). In this case, ΔL becomes negligible, and no torque is generated to induce spin flip (Fig. R17b(ii)). Conversely, ΔL increases when the two orbitals are oriented in different directions and reaches a maximum when their orbital orientation axes are perpendicular (Fig. R17b(iii)). Under these conditions, the electron experiences a large torque, which strongly promotes the spin-flip process (Fig. R17b(iv)). Therefore, discussing changes in orbital orientation

axis is effectively equivalent to discussing the orientation and magnitude of ΔL , providing an intuitive approach for visualizing the magnitude of SOC.

Fig. R17 a, Relationship between (i) atomic orbital and effective component of \vec{l} and (ii) orbital orientation axis and projected component of \vec{l} . **b**, Schematic illustrations of (i) two orbitals have the same orientation axis, (ii) the absence of torque generation for spin flipping, (iii) two orbitals oriented along different axes, and (iv) the electron experiencing a large torque that promotes spin flipping. In panels (ii) and (iv), B_{eff} and μ_S represent the effective magnetic field generated by \vec{l} and the electron spin magnetic momentum, respectively.

The use of the three factors of the orbital overlap, HAE, and the orbital orientation, helps us understand the reason for the significantly suppressed k_{nr}^T of **5** compared with that of **3** (energy diagrams in Fig. R18, left). First, in terms of the orbital overlap, different amounts of orbital overlap are confirmed between **3** and **5**. In both **3** and **5**, HOMO and LUMO mainly contribute to the T_1 - S_0 transition. In **3**, HOMO and LUMO largely overlap around Se atom (Fig. R18(i), top). Meanwhile, the much smaller overlap between HOMO and LUMO was confirmed in **5** (Fig. R18(i), bottom). Therefore, the contribution of orbital overlap factor to $\langle T_1 | H_{SO} | S_0 \rangle^2$ is suppressed in **5** compared with **3**. Second, regarding HAE factor, the Se atom is often considered to be the heavy atom source for the HAE and it is widely utilized for organic phosphorescence materials. However, the

difference in the $\langle T_1 | H_{SO} | S_0 \rangle^2$ values between **3** and **5** cannot be well explained by the HAE contribution because both the distance between Se atom and HOMO-LUMO overlap sites does not significantly change between **3** and **5** (Fig. R18(ii)). Third, regarding the orbital orientation factor that changes $\langle T_1 | H_{SO} | S_0 \rangle$, the orbital orientation on the Se atoms differs between the HOMO and LUMO of **3** (Fig. R18(iii), top). Because the HOMO and LUMO on the Se atoms also exhibit different orientations for **5** (Fig. R18(iii), bottom), the orbital orientation factor does not well explain the different $\langle T_1 | H_{SO} | S_0 \rangle^2$ values between **3** and **5**. Therefore, the spatial overlap between the HOMO and LUMO on Se mainly causes the smaller $\langle T_1 | H_{SO} | S_0 \rangle^2$ of **5** than **3**, suppressing the increase of k_{nr}^T for **5** compared with that for **3**. Thus, consideration of the orbital overlap, HAE, and orbital orientation factors provide us with an understanding of the different characteristics of $\langle T_1 | H_{SO} | S_0 \rangle^2$.

Fig. R18 a, Molecular orbitals and schematic illustrations related to the (i) orbital overlap, (ii) HAE, and (iii) orbital orientation to explain the smaller k_{nr}^T of **5** than **3**. In (iii), white arrows represent the orbital orientation axes. The isovalues of the HOMO of **3**, LUMO of **3**, and HOMO of **5** are 0.018. The isovalue of 0.0072 is used for the LUMO of **5**. Because the probability of the existence of the LUMO near Se in **5** is extremely small, the isovalue is set small to express the very small existence.

The argument of dividing SOC into three parts, orbital overlap, HAE, and orbital orientation, also worked for $\langle S_n | \mathbf{H}_{SO} | T_1 \rangle$, which is related to k_r^T based on equations (R2-1) and (R2-2). $\langle S_n | \mathbf{H}_{SO} | T_1 \rangle^2$ of **5** is comparable to that of **3** (energy diagrams in Fig. R19, left). First, regarding orbital overlap factor, magnitude of $\langle S_n | \mathbf{H}_{SO} | T_1 \rangle$ logically increases as the overlap between the orbital overlap density related to the S_n-S_0 transition and the orbital overlap density related to the T_1-S_0 transition increases. For **3**, $\langle S_5 | \mathbf{H}_{SO} | T_1 \rangle$ significantly contributes to the increase in k_r^T , but the overlap density of the S_5-S_0 transition [(HOMO-2) \times (LUMO)] only slightly overlaps with that of the T_1-S_0 transition [(HOMO) \times (LUMO)] between C and Se (green open circle in top of Fig. R19(i)). Although $\langle S_2 | \mathbf{H}_{SO} | T_1 \rangle$ significantly contributes to the increase in k_r^T for **5**, similarly, the overlap density of the S_2-S_0 transition [(HOMO-1) \times (LUMO)] only has a small overlap with that of the T_1-S_0 transition [(HOMO) \times (LUMO)] as a through-space interaction between C and Se (green open circle at the bottom of Fig. R19(i)). Therefore, the comparable poor orbital overlap hardly changes $\langle S_n | \mathbf{H}_{SO} | T_1 \rangle$ between **3** and **5**. Second, regarding the contribution of HAE to increasing $\langle S_n | \mathbf{H}_{SO} | T_1 \rangle$, the distance between the overlapping site and Se does not significantly change between **3** and **5** (Fig. R19(ii)). Therefore, the effect of the HAE does not result in a driving force that significantly differentiates between $\langle S_5 | \mathbf{H}_{SO} | T_1 \rangle$ of **3** and $\langle S_2 | \mathbf{H}_{SO} | T_1 \rangle$ of **5**. Third, regarding the contribution of the orbital axis orientation to the change in $\langle S_n | \mathbf{H}_{SO} | T_1 \rangle$, for **3**, the orbital axes of the (HOMO-2) \times (LUMO) and (HOMO) \times (LUMO) are in different directions (Fig. R19(iii), top). Similarly, for **5**, the orbital axis of the through-space type (HOMO-1) \times (LUMO) formed between Se and C is different from that of the (HOMO) \times (LUMO) on DBC (Fig. R19b(iii), bottom). Therefore, the orbital axis orientation factor similarly increases the $\langle S_n | \mathbf{H}_{SO} | T_1 \rangle$ of **3** and **5**. Because all three factors of the orbital overlap, HAE, and orbital orientation are present at the through-space site between Se and DBC, the $\langle S_n | \mathbf{H}_{SO} | T_1 \rangle$ of **5** becomes as large as that of **3**.

Fig. R19 Molecular orbitals and schematic illustrations showing that $\langle S_5 | H_{SO} | T_1 \rangle^2$ of **3** (top) and $\langle S_2 | H_{SO} | T_1 \rangle^2$ of **5** (bottom) have comparable magnitudes. Panels (i), (ii), and (iii) represent the orbital overlap, HAE, and orbital orientation effect, respectively. The white arrows in (iii) represent the orbital axis orientations. The isovalues of the overlap density regarding (HOMO-2) \times (LUMO) and (HOMO) \times (LUMO) for **3** and (HOMO-1) \times (LUMO) and (HOMO) \times (LUMO) for **5** are 0.00011.

To summarize characteristics of SOC term, both $\langle T_1 | H_{SO} | S_0 \rangle$ and $\langle S_n | H_{SO} | T_1 \rangle$ could be analysed by individually discussing the three factors of the orbital overlap, HAE, and orbital orientation. In through-space type molecule, the $\langle T_1 | H_{SO} | S_0 \rangle$ that relates to k_{nr}^T was suppressed due to the small orbital-overlapping regarding T_1-S_0 transition. In contrast, the $\langle S_n | H_{SO} | T_1 \rangle$ that relates to k_r^T was maintained because of the sufficient orbital-overlapping contribution. However, the orbital overlap is not necessarily generated solely by a close spatial proximity between the heavy atom (Se) and the central π -core.

To clarify the contribution of spatial proximity effect, we synthesized a reference chromophore (**R2**) in which phenylseleno substituent is horizontally introduced at the 3-position of DBC core (Fig. R20a). In **R2**, Se atom is not directly conjugated with central π -core because of no electron density at 3-position of DBC. Therefore, **R2** is a through-

space type chromophore. Additionally, the distance between Se and the 3-position of DBC for **R2** is 1.94 Å (Fig. R20b), which is shorter than that for **4** (3.19 Å) and that for **5** (3.05 Å). The Φ_p of **R2** was measured to be 11.5%, which is lower than that of **4** (= 17.6%) and **5** (= 27.0%). The k_r^T and k_{nr}^T of **R2** were determined to be 1.2 s⁻¹ and 9.5 s⁻¹, respectively. Compared with **4** and **5**, the small enhancement of k_r^T relative to k_{nr}^T was observed in **R2** from **1** (green arrow in Fig. R20c). The small enhancement of k_r^T indicates that the contribution of spatial proximity to k_r^T is weak, while the other factors are unfavorable in **R2**.

Fig. R20 **a**, Chemical structure of **R2**. **b**, Distance between Se atom and 3-position of DBC in T_1 optimized geometry. **c**, Relationship between optically determined k_r^T and k_{nr}^T among **1**, **4**, **5**, and **R2**.

The perspective of the orbital overlap factor that changes SOC strength provided a rational reason for the small enhancement of k_r^T of **R2**. Quantum chemical calculation confirmed that $n = 6$ is mainly contribute to k_r^T of **R2** among $\langle S_n | H_{SO} | T_1 \rangle$ term. The overlap density of the overlap density of the S_6-S_0 transition [(HOMO-2) \times (LUMO)] hardly overlaps with that of T_1-S_0 transition [(HOMO) \times (LUMO)] (Fig. R21). This lack of spatial overlap does not allow enhancement of $\langle S_6 | H_{SO} | T_1 \rangle$ in **R2**. Indeed, $\langle S_6 | H_{SO} | T_1 \rangle^2$ of **R2** is 6.97 cm⁻², which is much lower than $\langle S_7 | H_{SO} | T_1 \rangle^2$ of **4** (14.1 cm⁻²), and $\langle S_2 | H_{SO} | T_1 \rangle^2$ of **5** (361 cm⁻²). Therefore, simply placing the heavy atom in close proximity to the central π -core is insufficient. The orbital overlap selectively working on $\langle S_n | H_{SO} | T_1 \rangle$ is a key molecular design strategy for achieving a large k_r^T .

Fig. R21 Molecular orbitals and schematic illustrations related to orbital overlap to explain small $\langle S_6 | H_{SO} | T_1 \rangle^2$ of **R2**. The isovalues of the overlap density regarding (HOMO-2) \times (LUMO) and (HOMO) \times (LUMO) are 0.00016.

From the above, first of all, we added following Fig. 3a in revised manuscript to illustrate the three factors governing SOC.

After revision: Fig. 3a in revised manuscript

Caption of Figure 3a, Schematic illustration of the parameters affecting $\langle T_1 | \mathbf{H}_{SO} | S_0 \rangle^2$: the (i) orbital overlap, (ii) heavy atom effect (HAE), and (iii) orbital orientation.”

Furthermore, we added the sentences in revised manuscript to explain the three factors governing SOC.

After revision: From line 6 in p. 7 to line 8 in p. 8 in revised manuscript

“The first factor that changes $\langle T_1 | \mathbf{H}_{SO} | S_0 \rangle^2$ is the overlap of the orbitals. $\langle T_1 | \mathbf{H}_{SO} | S_0 \rangle^2$ increases when the overlap of the orbitals involved in the two transitions is large (Fig. 3a(i)). \mathbf{H}_{SO} is expressed as^{6,29,30}

$$\mathbf{H}_{SO} = \alpha^2 \sum_A \sum_i \frac{Z_A}{r_{Ai}^3} \vec{l}_{Ai} \vec{s}_i, \quad (2)$$

where α is the fine structure constant, A indicates the nucleus, i indicates the electron, Z is the atomic number, r is the electron–nucleus distance, \vec{l} is the orbital angular momentum, and \vec{s} is the spin angular momentum. The second factor that changes $\langle T_1 | \mathbf{H}_{SO} | S_0 \rangle^2$ is the distance between the heavy atoms and the overlap site of the orbitals involved in the T_1 – S_0 transition. The $\frac{Z_A}{r_{Ai}^3}$ term in equation (2) means that $\langle T_1 | \mathbf{H}_{SO} | S_0 \rangle^2$

increases when the heavy atoms are closer to the overlap site of the orbitals involved in the T_1-S_0 transition (Fig. 3a(ii)). This is the contribution of the increase in $\langle T_1 | \mathbf{H}_{SO} | S_0 \rangle^2$ owing to the introduction of heavier atoms, that is, the heavy atom effect (HAE). The third factor that changes $\langle T_1 | \mathbf{H}_{SO} | S_0 \rangle^2$ is the relative orientation of the molecular orbitals. $\vec{l}_{Ai} \vec{s}_i$ in equation (2) means that $\langle T_1 | \mathbf{H}_{SO} | S_0 \rangle^2$ increases when the axes of the two orbitals involved in the overlapping region point in different directions (Fig. 3a(iii)). This enhancement of the SOC by this term is known as the El-sayed rule³¹. When the two orbital axes are in the same direction, no torque is generated to flip the spin because the electron spin and magnetic moment are parallel (left of Fig. 3a(iii), supplementary Section 5-2, and supplementary Fig. S25). Conversely, when the axes of the two orbitals are in different directions, a torque that flips the spin is generated because the electron spin and magnetic moment are not parallel (Fig. 3a(iii), right)³². Thus, the three factors of the orbital overlap, HAE, and relative orientation of the molecular orbitals can be individually considered to discuss the difference in $\langle T_1 | \mathbf{H}_{SO} | S_0 \rangle^2 \propto k_{nr}^T$ between **3** and **5**.”

Additionally, we added the relationship between orbital orientation and \vec{l} in Section 5-2 in revised supplementary information.

After revision: Section 5-2 in revised supplementary information

“5-2. Relationship between orbital orientation characteristics and spin flip (Figure S25)

Transitions between singlet and triplet states necessarily involve a change in the spin angular momentum (\vec{s}). According to the conservation of total angular momentum, such a spin conversion must be accompanied by the change (ΔL) in the orbital angular momentum (\vec{l}). The resulting ΔL exerts a torque on the electron, thereby promoting spin flipping. The magnitude of ΔL depends on the vector cross product of the respective \vec{l} . Therefore, understanding the direction of the \vec{l} is essential when discussing the electronic transitions that require spin flipping.

The orbital orientation axis is directly related to orientation of \vec{l} . In case of p_z orbital, \vec{l} exhibits an effective component within the xy-plane (Fig. S25a(i)). When the projection component is considered, \vec{l} is oriented perpendicular to the orbital orientation axis (Fig. S25a(ii)). When two orbitals are oriented along the same axis, their corresponding \vec{l} are aligned in the same direction (Fig. S25b(i)). In this case, ΔL becomes negligible, and no torque is generated to induce spin flip (Fig. S25b(ii)). Conversely, ΔL increases when the two orbitals are oriented in different directions and reaches a maximum when their orbital orientation axes are perpendicular (Fig. S25b(iii)). Under these conditions, the electron experiences a large torque, which strongly promotes

the spin-flip process (Fig. S25b(iv)). Therefore, discussing changes in orbital orientation axis is effectively equivalent to discussing the orientation and magnitude of ΔL , providing an intuitive approach for visualizing the magnitude of SOC.

Fig. S25. Relationship between orbital orientation characteristics and spin flip. **a**, Relationship between (i) atomic orbital and effective component of \vec{l} and (ii) orbital orientation axis and projected component of \vec{l} . **b**, Schematic illustrations of (i) two orbitals having the same orientation axis, (ii) the absence of torque generation for spin flipping, (iii) two orbitals oriented along different axes, and (iv) the electron experiencing a large torque that promotes spin flipping. In panels (ii) and (iv) of **b**, B_{eff} and μ_s represent the effective magnetic field generated by \vec{l} and the electron spin magnetic momentum, respectively.”

Next, we changed Fig. 3b(i) in original manuscript into following Fig. 3b in revised manuscript to explain the differences in $\langle T_1 | H_{\text{SO}} | S_0 \rangle^2$ between **3** and **5** using the orbital overlap, HAE, and the orbital orientation factors.

Before revision: Fig. 3b(i) and (ii) in original manuscript

Caption of Figure 3b, Molecular orbitals and their calculated values relating (i) and (ii) $\langle T_1 | H_{SO} | S_0 \rangle^2$, (iii) and (iv) $\langle S_n | H_{SO} | T_1 \rangle^2$, and (v) and (vi) $\mu_{S_n-S_0}^2$. In (i) and (ii), the red-blue orbitals and orange-light blue orbitals are the highest occupied molecular orbital (HOMO) and lowest unoccupied molecular orbital (LUMO), respectively. In (iii) and (iv), the red-blue orbitals are Ω_{S_3} or Ω_{S_5} and the orange-light blue orbitals are Ω_{T_1} . In (i)–(iv), the white arrows represent the orbital angular momentum (OAM) vectors. The isovalues of the HOMO of **3**, LUMO of **3**, and HOMO of **5** are 0.02. The isovalue of 0.0085 is used for the LUMO of **5**. The isovalues of Ω_{S_3} , Ω_{S_5} , and Ω_{T_1} are 0.00016.”

After revision: Fig. 3b in revised manuscript

Caption of Figure 3b, Molecular orbitals and schematic illustrations related to the (i) orbital overlap, (ii) HAE, and (ii) orbital orientation to explain the smaller k_{nr}^T of **5** than **3**. In (iii) of **a** and **b**, the white arrows represent the orbital orientation axes. The isovalues of the highest occupied molecular orbital (HOMO) of **3**, lowest unoccupied molecular orbital (LUMO) of **3**, and HOMO of **5** are 0.018. The isovalue of 0.0072 is used for the LUMO of **5**. Because the probability of the existence of the LUMO near Se in **5** is extremely small, the isovalue is set small to express the very small existence.”

Additionally, we changed the sentences in revised manuscript to discuss why $\langle T_1 | H_{SO} | S_0 \rangle^2$ of **3** is significantly smaller than that of **5** using the three factors governing SOC.

Before revision: From line 20 in p. 6 to line 7 in p. 7 in original manuscript

“First, compared with the horizontal hetero-substituents, the vertically introduced hetero-substituents suppress $\langle T_1 | H_{SO} | S_0 \rangle^2$ (Fig. 3b, A). In **3**, the highest occupied molecular orbital (HOMO) and lowest unoccupied molecular orbital (LUMO) greatly overlap on the Se atom and have different directions of the orbital angular momentum (OAM) vectors on Se (Fig. 3b(i)). Because the spin-orbit Hamiltonian involves the OAM operator that rotates the vector by 90° ²⁶, $\langle T_1 | H_{SO} | S_0 \rangle^2$ increases when the HOMO and LUMO have large overlap and different directions of the OAM vectors. Therefore, **3** has

large $\langle T_1 | H_{SO} | S_0 \rangle^2$ owing to such molecular orbitals on Se. In **5**, however, the spatial overlap of the HOMO and LUMO on Se is small, although the HOMO and LUMO have different directions of the OAM vectors on Se (Fig. 3b(ii) and supplementary Fig. S22). Owing to the small overlap on Se, $\langle T_1 | H_{SO} | S_0 \rangle^2$ of **5** is significantly smaller than that of **3**.”

After revision: From line 9 in p. 8 to line 4 in p. 9 in revised manuscript

“The suppression of the increase in k_{nr}^T with increasing number of vertical hetero-substitutions is due to the small overlap of the orbitals involved in the T_1 – S_0 transition. Regarding the orbital overlap, the first factor that changes $\langle T_1 | H_{SO} | S_0 \rangle^2$, the T_1 – S_0 transition is predominantly composed of the highest occupied molecular orbital (HOMO) and lowest unoccupied molecular orbital (LUMO) for both **3** and **5** (supplementary Fig. S26). Because the spatial overlap between the HOMO and LUMO on Se of **5** is much smaller than that of **3** (Fig. 3b(i)), the different amounts of orbital overlap contribute to the different $\langle T_1 | H_{SO} | S_0 \rangle^2$ values between **3** and **5**. Regarding the HAE, the second factor that changes $\langle T_1 | H_{SO} | S_0 \rangle^2$ for **3** and **5**, the Se atom is often considered to be the heavy atom source for the HAE, and it is widely used for organic phosphorescence materials^{33–35}. However, the large difference in the $\langle T_1 | H_{SO} | S_0 \rangle^2$ values between **3** and **5** cannot be well explained by the HAE contribution because the distance between the HOMO and LUMO overlap sites and Se does not significantly differ between **3** and **5** (Fig. 3b(ii)). Regarding the orbital orientation factor, the third factor that changes $\langle T_1 | H_{SO} | S_0 \rangle^2$, the orbital orientation on the Se atoms differs between the HOMO and LUMO of **3** (Fig. 3b(iii), left). Because the HOMO and LUMO on the Se atoms also exhibit different orientations for **5** (Fig. 3b(iii), right), the orbital orientation factor does not well explain the different $\langle T_1 | H_{SO} | S_0 \rangle^2$ values between **3** and **5**. Therefore, the spatial overlap between the HOMO and LUMO on Se mainly causes the smaller $\langle T_1 | H_{SO} | S_0 \rangle^2$ of **5** than **3**, suppressing the increase of k_{nr}^T for **5** compared with that for **3**.”

Next, we changed Fig. 3b(iii) and (iv) in original manuscript into Fig. 5 in revised manuscript to add the T_m contribution and to discuss the $\langle S_n | H_{SO} | T_1 \rangle^2$ and $\langle T_m | H_{SO} | S_0 \rangle^2$ using the orbital overlap, HAE, and the orbital orientation factors. Because Reviewer 1 requested some considerations related to k_r^T in higher triplet states, we have included that information in Figure 5c in revised manuscript.

Before revision: Fig. 3b(iii) and (iv) in original manuscript

Caption of Figure 3b, Molecular orbitals and their calculated values relating (i) and (ii) $\langle T_1 | H_{SO} | S_0 \rangle^2$, (iii) and (iv) $\langle S_n | H_{SO} | T_1 \rangle^2$, and (v) and (vi) $\mu_{S_n-S_0}^2$. In (i) and (ii), the red-blue orbitals and orange-light blue orbitals are the highest occupied molecular orbital (HOMO) and lowest unoccupied molecular orbital (LUMO), respectively. In (iii) and (iv), the red-blue orbitals are Ω_{S3} or Ω_{S5} and the orange-light blue orbitals are Ω_{T1} . In (i)–(iv), the white arrows represent the orbital angular momentum (OAM) vectors. The isovalues of the HOMO of **3**, LUMO of **3**, and HOMO of **5** are 0.02. The isovalue of 0.0085 is used for the LUMO of **5**. The isovalues of Ω_{S3} , Ω_{S5} , and Ω_{T1} are 0.00016.”

After revision: Fig. 5 in revised manuscript

Fig. 5 Difference in the factors governing $\langle S_n | H_{SO} | T_1 \rangle^2$ and $\langle T_m | H_{SO} | S_0 \rangle^2$ between horizontal and vertical hetero-substituents. **a**, Simplified expression of the k_r^T equation. **b**, Molecular orbitals and schematic illustrations showing that $\langle S_5 | H_{SO} | T_1 \rangle^2$ of **3** (top) and $\langle S_2 | H_{SO} | T_1 \rangle^2$ of **5** (bottom) have comparable magnitudes. The isovalues of the overlap density regarding (HOMO-2)×(LUMO) and (HOMO)×(LUMO) for **3** and (HOMO-1)×(LUMO) and (HOMO)×(LUMO) for **5** are 0.00011. **c**, Molecular orbitals and schematic illustrations explaining the comparable magnitudes of $\langle T_7 | H_{SO} | S_0 \rangle^2$ for **3** (top) and $\langle T_9 | H_{SO} | S_0 \rangle^2$ for **5** (bottom). The isovalue of 0.013 is used for HOMO-3 and LUMO of **3** and HOMO-2 and LUMO of **5**. In **b** and **c**, panels (i), (ii), and (iii) represent the orbital overlap, HAE, and orbital orientation effect, respectively. The white arrows in (iii) represent the orbital axis orientations.”

Additionally, we changed the sentences in revised manuscript to discuss why $\langle S_n | H_{SO} | T_1 \rangle^2$ of **3** were maintained up to that of **5** using the three factors governing SOC. Before revision: Lines 8–18 in p. 7 in original manuscript

“Next, multiple vertically introduced hetero-substituents maintain $\langle S_n | H_{SO} | T_1 \rangle^2$, enhancing k_r^T to a comparable magnitude to $\langle T_1 | H_{SO} | S_0 \rangle^2$ (Fig. 3b, B). In **3** and **5**, $n = 5$ and $n = 3$ mainly enhance k_r^T , respectively (supplementary Fig. S23). In **3**, $\langle S_5 | H_{SO} | T_1 \rangle^2$ for the driving force of k_r^T greatly decreases compared with the large $\langle T_1 | H_{SO} | S_0 \rangle^2$ (Fig. 3b(iii)). In contrast, $\langle S_3 | H_{SO} | T_1 \rangle^2$ for the driving force of k_r^T increases compared with the

small $\langle T_1 | \mathbf{H}_{SO} | S_0 \rangle^2$ in **5**. In **5**, the overlap density for the S_3 – S_0 transition ($\mathbf{\Omega}_{S_3}$) composed of HOMO-1 and LUMO forms a through-space interaction between Se and C of DBC (Fig. 3b(iv), green open circle, and supplementary Fig. S24b). Because this partially overlaps with the overlap density for the T_1 – S_0 transition ($\mathbf{\Omega}_{T_1}$) of C on DBC, the spatial overlap with different OAM directions for $\mathbf{\Omega}_{S_3}$ and $\mathbf{\Omega}_{T_1}$ between the Se and C atoms results in the sufficient $\langle S_3 | \mathbf{H}_{SO} | T_1 \rangle^2$ in **5** (Fig. 3b(iv)).”

After revision: From line 21 in p. 10 to line 2 in p. 12 in revised manuscript

“ $\langle S_n | \mathbf{H}_{SO} | T_1 \rangle^2$ of the multiple vertical hetero-substituted chromophore approaching that of the multiple horizontal hetero-substituted chromophore can be understood by considering the following three factors, which are the same factors as those explained the SOC regarding k_{nr}^T in Fig. 3a. First, regarding the overlap of the orbitals before and after the transition that increases $\langle S_n | \mathbf{H}_{SO} | T_1 \rangle$ in equation (4), $\langle S_n | \mathbf{H}_{SO} | T_1 \rangle$ logically increases as the overlap between the orbital overlap density related to the S_n – S_0 transition and the orbital overlap density related to the T_1 – S_0 transition increases. For **3**, $\langle S_5 | \mathbf{H}_{SO} | T_1 \rangle$ significantly contributes to the increase in k_{nr}^T , but the overlap density of the S_5 – S_0 transition [(HOMO-2)×(LUMO)] only slightly overlaps with that of the T_1 – S_0 transition [(HOMO)×(LUMO)] between C and Se (green open circles at the top of Fig. 5b(i) and supplementary Fig. S32a). Although $\langle S_2 | \mathbf{H}_{SO} | T_1 \rangle$ significantly contributes to the increase in k_{nr}^T for **5**, similarly, the overlap density of the S_2 – S_0 transition [(HOMO-1)×(LUMO)] only has a small overlap with that of the T_1 – S_0 transition [(HOMO)×(LUMO)] as a through-space interaction between C and Se (green open circle at the bottom of Fig. 5b(i) and supplementary Fig. S32b). Therefore, the comparable poor orbital overlap hardly changes $\langle S_n | \mathbf{H}_{SO} | T_1 \rangle$ between **3** and **5**. Second, regarding the contribution of the HAE to increasing $\langle S_n | \mathbf{H}_{SO} | T_1 \rangle$ in equation (4), the distance between the overlapping site and Se does not significantly change between **3** and **5** (Fig. 5b(ii)). Therefore, the effect of the HAE does not result in a driving force that significantly differentiates between $\langle S_5 | \mathbf{H}_{SO} | T_1 \rangle$ of **3** and $\langle S_2 | \mathbf{H}_{SO} | T_1 \rangle$ of **5**. Third, regarding the contribution of the orbital axis orientation to the change in $\langle S_n | \mathbf{H}_{SO} | T_1 \rangle$, for **3**, the orbital axes of (HOMO-2)×(LUMO) and (HOMO)×(LUMO) are in different directions (Fig. 5b(iii), top). Similarly, for **5**, the orbital axis of the through-space type (HOMO-1)×(LUMO) formed between Se and C is different from that of the (HOMO)×(LUMO) on DBC (Fig. 5b(iii), bottom). Therefore, the orbital axis orientation hardly changes $\langle S_n | \mathbf{H}_{SO} | T_1 \rangle$ between **3** and **5**. Therefore, these similar tendencies regarding the spatial overlap characteristics, HAE, and orbital orientation increases $\langle S_2 | \mathbf{H}_{SO} | T_1 \rangle^2$ of **5** to a comparable value to $\langle S_5 | \mathbf{H}_{SO} | T_1 \rangle^2$ of **3**. When the Se atoms in **5** are replaced with S atoms, the enhancement

of $\langle S_n | \mathbf{H}_{SO} | T_1 \rangle^2$ becomes small owing to the lack of an orbital overlap factor (supplementary Figs. S13 and S33–S35, and Table S4).”

Next, we added the following sentence regarding properties of **R2** in revised manuscript.

After revision: Lines 10–12 in p. 10 in revised manuscript

“This characteristic was also confirmed for chromophores with a horizontal phenylseleno substituent at the 3-position of DBC, and Φ_p of the chromophore was indeed small (supplementary Figs. S14 and S29–S31, and Table S3).”

Additionally, we added the synthesis procedure and characterization data of **R2** in Section 1 in revised supplementary information.

After revision: Section 1 in revised supplementary information

“Dibenzo[*g,p*]chrysene-3-yl(phenyl)selane (R2):

9-Fluorenone (1.35 g, 7.50 mmol), 2-bromo-9-fluorenone (648 mg, 2.50 mmol), zinc (4.21 g, 64.4 mmol), and zinc chloride (2.73 g, 20.0 mmol) in tetrahydrofuran (25 ml) and pure water (25 ml) was stirred at room temperature for 4 h. After completion of the reaction, dichloromethane was added to the reaction solution and filtrated to remove the undissolved solid. Pure water was added to the solution and extracted with dichloromethane. The organic layer was dried over anhydrous Na₂SO₄ and evaporated under reduced pressure to give a white powder (2.09 g). The powder (2.09 g) in acetic acid (35.5 ml) and sulfuric acid (1.8 ml) was heated at 85 °C for 4 h. After cooling to room temperature, dichloromethane was added to the reaction mixture and washed with pure water and extracted with dichloromethane. The organic layer was dried over anhydrous Na₂SO₄ and concentrated under reduced pressure. The residue was purified by column chromatography (silica gel; dichloromethane: hexane = 3: 7) to give C₂₆H₁₆BrO (430 mg) as a white powder. Although the products of C₂₆H₁₆BrO had potentially three kinds of isomers, the products were directly used for the next reaction without further purification. C₂₆H₁₆BrO (430 mg, 1.02 mmol), sodium borohydride (67.3 mg, 1.78 mmol) in anhydrous tetrahydrofuran (15 ml) and methanol (15 ml) was stirred at 60 °C for 3 h under a nitrogen atmosphere. After completion of the rection, dichloromethane was added to the solution and washed with pure water. The organic layer was dried over anhydrous Na₂SO₄ and concentrated under reduced pressure to give a white powder (391 mg). The powder (391 mg) in acetic acid (28 ml) and sulfuric acid (0.70 ml) was heated at 85 °C for 4 h. After cooling to room temperature, pure water was added to the reaction solution

and extracted with dichloromethane. The extracted organic layer was dried over anhydrous Na₂SO₄ and evaporated under reduced pressure. The residue was purified by column chromatography (silica gel; dichloromethane: hexane = 1: 9) to yield 3-bromodibenzo[*g,p*]chrysene (212 mg) as a white powder. The reaction yield of 3-bromodibenzo[*g,p*]chrysene from 2-bromo-9-fluorenone was 20.8%. 3-Bromodibenzo[*g,p*]chrysene (142 mg, 0.349 mmol), diphenyl diselenide (172 mg, 0.552 mmol), [1,1'-bis(diphenylphosphino)ferrocene]palladium(II) dichloride (40.4 mg, 0.0552 mmol), and zinc (72.2 mg, 1.10 mmol) in anhydrous tetrahydrofuran (11 ml) was stirred at reflux temperature for 16 h under a nitrogen atmosphere. After completion of the reaction, chloroform was added to the reaction solution. The organic phase was washed with pure water three times, dried over anhydrous Na₂SO₄, and concentrated under the reduced pressure to give a crude product. The crude product was purified by column chromatography (silica gel; dichloromethane: hexane = 12: 88) to give **R2** as a pale-yellow powder (73.7 mg, 43.7%). Chromophore **R2** was purified further by sublimation. ¹H NMR (CDCl₃, 500 MHz): δ = 8.71 (sd, *J* = 1.5 Hz, 1H), 8.68–8.63 (m, 5H), 8.58 (d, *J* = 8.5 Hz, 1H), 8.36 (dd, *J* = 8.0, 1.0 Hz, 1H), 7.75 (dd, *J* = 8.5, 1.5 Hz, 1H), 7.69–7.57 (m, 7H), 7.45 (ddd, *J* = 7.5, 6.9, 1.4 Hz, 1H), 7.33–7.29 (m, 3H) (Fig. S14a). ¹³C NMR (CDCl₃, 125 MHz): δ = 133.70, 132.69, 131.01, 130.97, 130.78, 130.54, 130.36, 130.15, 129.99, 129.78, 129.51, 129.22, 129.06, 128.98, 128.91, 128.89, 128.47, 128.06, 127.65, 126.76, 126.75, 126.71, 126.69, 126.58, 126.56, 124.38, 123.61, 123.52, 123.45 (Fig. S14b). HRMS-MALDI (*m/z*): [*M*]⁺ calcd. for C₃₂H₂₀Se, 484.0730; found 484.0729 (Fig. S14c). Anal. Calcd. for C₃₂H₂₀Se: C, 79.50; H, 4.17; Se, 16.33. Found C, 79.62; H, 4.26.

Fig. S14. Characterization data of **R2**. **a**, ^1H NMR spectrum in CDCl_3 . **b**, ^{13}C NMR spectrum in CDCl_3 . **c**, HRMS-MALDI spectrum.”

Next, we added the optical properties and calculation results of **R2** in Section 5-6 in revised supplementary information.

After revision: Section 5-6 in revised supplementary information

“5-6. Calculation results and photophysical properties of chromophore **R2** (Figures S29–S31 and Table S3)

To clearly clarify the contribution of the intramolecular external heavy atom effect (HAE) derived from spatial proximity between heavy atom and central π -core, chromophore **R2** in which a phenylseleno substituent is horizontally introduced at the 3-position of the DBC core was synthesized (Fig. S29a). In **R2**, Se atom is not directly conjugated with emissive core because of no electron density at 3-position of DBC. Therefore, **R2** has a potential characteristic of the intramolecular external HAE. Additionally, the distance between Se and the 3-position of DBC is 1.94 Å in T_1 optimized geometry (Fig. S29b). Because the distance between Se and C of DBC for **4** (3.19 Å) and **5** (3.05 Å) are more separated spatially, the intramolecular external HAE may play significantly in **R2**.

The **R2** was doped in the amorphous β -estradiol at a concentration with 0.3 wt%. The amorphous film of **R2**-doped β -estradiol showed blue fluorescence under excitation and red RTP after ceasing excitation (Fig. S29c and Fig. S29d). The Φ_p of **R2** was measured to be 11.5%, which is lower than that of **4** (= 17.6%) and **5** (= 27.0%). The τ_p of **R2** was determined to be 92.9 ms from RTP decay characteristics (Fig. S29e). From the equations $\Phi_p = \Phi_{isc} k_r^T \tau_p$, $\tau_p = 1/(k_r^T + k_{nr}^T)$, and $\Phi_{isc} = 1 - \Phi_f$, the values of k_r^T and k_{nr}^T of **R2** were determined to be 1.2 s⁻¹ and 9.5 s⁻¹, respectively (Table S3). Compared with **4** and **5**, the small enhancement of k_r^T relative to k_{nr}^T was observed in **R2** from **1** (green arrow in Fig. S29f). The small enhancement of k_r^T relative to k_{nr}^T indicates that the contribution of spatial proximity between Se and DBC core to k_r^T enhancement is weak.

Fig. S29. Photophysical characteristics of **R2**. **a**, Chemical structure of **R2**. **b**, Distance between Se atom and C atom at 3-position of DBC. **c**, Emission spectrum of **R2** in amorphous β -estradiol under excitation. **d**, RTP spectrum of **R2** in amorphous β -estradiol soon after ceasing excitation. **e**, RTP decay characteristics of **R2** in amorphous β -estradiol. **f**, Relationship between optically determined k_r^T and k_{nr}^T among **1**, **4**, **5**, and **R2**.

Table S3. Photophysical parameters of **R2**.

Chromophores ^{a)}	Φ_f	$\Phi_{isc}^{b)}$	Φ_p	τ_p	k_r^T	k_{nr}^T
	[%]	[%]	[%]	[ms]	[s ⁻¹]	[s ⁻¹]
R2	0.66	99.3	11.5	92.9	1.2	9.5

^{a)}The concentration of chromophores was 0.3 wt% in amorphous β -estradiol. ^{b)} Values determined from $1 - \Phi_f$.

The relatively small enhancement of k_r^T compared with k_{nr}^T in **R2** could be confirmed from the different contributions of SOC to k_r^T and k_{nr}^T . Quantum chemical calculation confirmed that $n = 6$ is mostly contribute to k_r^T of **R2** (Fig. S30a). In **R2**, the $\langle S_6 | \mathbf{H}_{SO} | T_1 \rangle^2$ that enhances k_r^T is smaller than $\langle T_1 | \mathbf{H}_{SO} | S_0 \rangle^2 \propto k_{nr}^T$ (arrow in Fig. S30b).

Fig. S30. Calculated parameters relating to k_r^T in **R2**. **a**, Histograms of calculated parameters of $\langle S_n | H_{SO} | T_1 \rangle^2 E_{S_n-T_1}^{-2} \mu_{S_n-S_0}^2$ (top), $\langle S_n | H_{SO} | T_1 \rangle^2$ (middle), and $\mu_{S_n-S_0}^2$ (bottom) for **R2**. **b**, Histogram of the $\langle T_1 | H_{SO} | S_0 \rangle^2$ and $\langle S_6 | H_{SO} | T_1 \rangle^2$ values for **R2**.

The perspective of the orbital overlap factor that changes SOC strength provided a rational reason for the small enhancement of $\langle S_n | H_{SO} | T_1 \rangle$ of **R2**. The S_6-S_0 transition of **R2** consists primarily of two components with 28.4% and 28.3% contributions, respectively. The former is composed of HOMO-2 and LUMO. The latter is composed of HOMO and LUMO+2. The overlap density between HOMO-2 and LUMO for S_6-S_0 transition [(HOMO-2)×(LUMO)] hardly overlaps with that between HOMO and LUMO for T_1-S_0 transition [(HOMO)×(LUMO)] (Fig. S31, left). Similarly, no spatial overlap was observed between the overlap density of HOMO and LUMO+2 for S_6-S_0 transition [(HOMO)×(LUMO+2)] and the (HOMO)×(LUMO) for T_1-S_0 transition (Fig. S31, right). These lack of spatial overlap does not allow enhancement of $\langle S_6 | H_{SO} | T_1 \rangle$ in **R2**. Indeed, $\langle S_6 | H_{SO} | T_1 \rangle^2$ of **R2** is 6.97 cm^{-2} , which is much lower than $\langle S_7 | H_{SO} | T_1 \rangle^2$ of **4** (14.1 cm^{-2}), and $\langle S_2 | H_{SO} | T_1 \rangle^2$ of **5** (361 cm^{-2}). Therefore, simply placing the heavy atom in close proximity to the central π -core is insufficient. The orbital overlap selectively working on $\langle S_n | H_{SO} | T_1 \rangle$, which is realized for **5**, is a key molecular design strategy for achieving a large k_r^T .

Fig. S31. Molecular orbitals and schematic illustrations related to orbital overlap to explain small $\langle S_6 | H_{SO} | T_1 \rangle^2$ of **R2**. The isovalues of the overlap density regarding (HOMO-2) \times (LUMO), (HOMO) \times (LUMO), and (HOMO) \times (LUMO+2) are 0.00016.”

(Comment 3 from Reviewer 2)

The proposed cooperation between strong SOC from high-lying singlets (S5 and S3) and a large transition dipole moment is an intriguing and less common mechanism for enhancing phosphorescence. However, is there any experimental observation evidence to support this hypothesis?

(Response)

As previously mentioned in our response to first comment from Reviewer 2, k_r^T based on the first order perturbation is expressed as following equations (*Chem. Rev.* **66**, 199 – 241 (1966); *J. Phys. Chem. A* **111**, 10490–10499 (2007); *Chem. Rev.* **117**, 6500–6537 (2017)),

$$k_r^T = \frac{E_{T_1-S_0}^3}{3\varepsilon_0\pi\hbar^4c^3} |\mu_{T_1-S_0}|^2, \quad (\text{R2-1})$$

$$\mu_{T_1-S_0} = \sum_{n \geq 1} \frac{\langle S_n | H_{SO} | T_1 \rangle}{E_{S_n-T_1}} \mu_{S_n-S_0} + \sum_{m \geq 2} \frac{\langle T_m | H_{SO} | S_0 \rangle}{E_{T_m-S_0}} \mu_{T_m-T_1}. \quad (\text{R2-2})$$

Therefore, k_r^T is facilitated via multiplying product of $\langle S_n | H_{SO} | T_1 \rangle$ and $\mu_{S_n-S_0}$ as well as $\langle T_m | H_{SO} | S_0 \rangle$ and $\mu_{T_m-T_1}$. However, experimental determination using magnetic field of $\langle S_n | H_{SO} | T_1 \rangle$ and $\langle T_m | H_{SO} | S_0 \rangle$ are challenging, because higher-lying excited states decay

on extremely fast timescales, typically in less than 1 ps regime. Therefore, in the current state, the magnitudes of these higher-order SOC could be discussed by quantum chemical calculations. As shown in Fig. R22(i), a good correlation between the experimental and calculated values of k_r^T was obtained in not only the molecules in the original manuscript but also the newly synthesized molecules (**R1** and **R2**). In addition, using the same geometry, the calculated $\langle T_1 | H_{SO} | S_0 \rangle^2$, which theoretically proportional to the k_{nr}^T , correlated well with optically measured k_{nr}^T (Fig. R22(ii)). Therefore, using computational results to discuss the cooperative effects is considered reasonable.

Fig. R22 Relationship (i) between optically measured and calculated k_r^T and (ii) between optically measured k_{nr}^T and calculated $\langle T_1 | H_{SO} | S_0 \rangle^2$.

Accordingly, we updated the calculation results in revised supplementary information regarding the correlation between the calculated and measured k_r^T as well as the correlation between the measured k_{nr}^T and the calculated $\langle T_1 | H_{SO} | S_0 \rangle^2$, owing to the inclusion of the newly synthesized molecules into the correlation.

Before revision: Section 5-1 in original supplementary information

“5-1. Calculated values of k_r^T and $\langle T_1 | H_{SO} | S_0 \rangle^2$ ”

Calculation procedure of T_1 optimized structure for **1–5** were described in supplementary section 2. Using the T_1 optimized structure (Fig. S13), single-point calculations were performed using the time-dependent (TD) DFT with the Amsterdam Density Functional (ADF) 2018 package to calculate k_r^T and the spin-orbit coupling (SOC) between T_1 and S_0 ($\langle T_1 | \mathbf{H}_{SO} | S_0 \rangle$). The parameter $\langle S_n | \mathbf{H}_{SO} | T_1 \rangle$ was treated as a perturbation based on scalar relativistic orbitals using the PBE0 as a functional and TZP as a basis sets. The scalar relativistic-time-dependent DFT calculations included 10 singlet and 10 triplet excitations, which were used as the basis for the perturbative expansions in the calculations.

From the above calculations, the good correlation between calculated k_r^T and measured k_r^T (Fig. S21a) and between calculated $\langle T_1 | \mathbf{H}_{SO} | S_0 \rangle^2$ and measured k_{nr}^T (Fig. S21b) were observed. From **2** to **3**, the magnitude of calculated k_r^T enhancement (Fig. S21a, blue arrow) was smaller than that of $\langle T_1 | \mathbf{H}_{SO} | S_0 \rangle^2$ enhancement (Fig. S21b, blue arrow). In contrast, from **4** to **5**, calculated k_r^T was more enhanced compared with $\langle T_1 | \mathbf{H}_{SO} | S_0 \rangle^2$ (Figs. S21a and S21b, red arrows). Thus, the more selective enhancement of k_r^T compared with k_{nr}^T of **5** was observed from theoretical calculations as well as optical measurements. Therefore, the discussion on the differences in measured k_r^T and k_{nr}^T based on the quantum chemical calculation was reasonable.

Fig. S21. Relationship between measured and calculated parameters relating phosphorescence process. **a**, Correlation between measured and calculated k_r^T . **b**, Correlation between measured k_{nr}^T and calculated $\langle T_1 | \mathbf{H}_{SO} | S_0 \rangle^2$.

After revision: Section 5-1 in revised supplementary information

“5-1. Calculated values of k_r^T and $\langle T_1 | \mathbf{H}_{SO} | S_0 \rangle^2$ (Figure S24)

Calculation procedure of T_1 optimized structure for **1–5** were described in supplementary Section 2. Using the T_1 optimized structure (Fig. S15), single-point calculations were performed using the time-dependent (TD) DFT with the Amsterdam Density Functional (ADF) 2025 package to calculate k_r^T and the spin-orbit coupling (SOC) between T_1 and S_0 ($\langle T_1 | \mathbf{H}_{SO} | S_0 \rangle$). The parameter $\langle S_n | \mathbf{H}_{SO} | T_1 \rangle$ and $\langle T_m | \mathbf{H}_{SO} | S_0 \rangle$ were treated as a perturbation based on scalar relativistic orbitals using the PBE0 as a functional and TZP as a basis sets. The scalar relativistic-time-dependent DFT calculations included 10 singlet and 10 triplet excitations, which were used as the basis for the perturbative expansions in the calculations.

From the above calculations, the good correlation between calculated $\langle T_1 | \mathbf{H}_{SO} | S_0 \rangle^2$ and measured k_{nr}^T (Fig. S24a) and between calculated k_r^T and measured k_r^T (Fig. S24b) were observed among synthesized chromophores. From **2** to **3**, the magnitude of calculated k_r^T enhancement (Fig. S24b, blue arrow) was smaller than that of $\langle T_1 | \mathbf{H}_{SO} | S_0 \rangle^2$ enhancement (Fig. S24a, blue arrow). In contrast, from **4** to **5**, calculated k_r^T was more enhanced compared with $\langle T_1 | \mathbf{H}_{SO} | S_0 \rangle^2$ (Figs. S24a and S24b, red arrows). Thus, the more selective enhancement of k_r^T relative to k_{nr}^T of **5** was observed from theoretical calculations as well as optical measurements. Therefore, the discussion on the differences in measured k_r^T and k_{nr}^T based on the quantum chemical calculation is reasonable.

Fig. S24. Relationship between measured and calculated parameters relating phosphorescence process. **a**, Correlation between measured k_{nr}^T and calculated $\langle T_1 | \mathbf{H}_{SO} | S_0 \rangle^2$. **b**, Correlation between measured and calculated k_r^T .

(Comment 4 from Reviewer 2)

The claim regarding the vertical distribution of substituents relative to the central plane

should ideally be corroborated by single-crystal X-ray diffraction data. If such crystal structures are available for the discussed compounds, referencing them would provide direct and unambiguous experimental evidence for this asserted conformation.

(Response)

Thank you for your comments. As Reviewer 2 pointed out, the single-crystal X-ray diffraction data well helps us understand geometry. Because the vertically substituted chromophores did not crystallize due to their twisted nature, we could not determine the geometry in crystalline state. However, the geometries obtained from single crystal X-ray analysis may not accurately represent the molecular geometry in amorphous or other crystalline matrixes due to the intermolecular forces and packing effects. Furthermore, highly pure single crystals occasionally do not exhibit long-lived RTP (*Nat. Mater.* **20**, 1539–1544 (2021); *Angew. Chem. Int. Ed.* **62**, e202315911 (2023)), indicating that the molecular geometry in the single crystal is not necessarily essential for discussing photophysical parameters of RTP process. Moreover, as exemplified by biphenyl, the molecular geometry in the excited state often differs from that in the ground state (*Chem. Phys. Lett.* **376** 201–206 (2003)).

As mentioned in the above response, however, the k_r^T and $\langle T_1 | H_{SO} | S_0 \rangle^2$ values calculated by the optimized T_1 geometries of the synthesized molecules correlate well with the experimentally measured k_r^T and k_{nr}^T (Fig. R22). Therefore, the use of parameters calculated by T_1 optimized geometry is a reasonable approach for discussing the photophysical behavior of the synthesized molecules.

(Comment 5 from Reviewer 2)

The manuscript's language will be thoroughly refined to enhance precision, clarity, and conciseness.

(Response)

We asked a professional company to proofread the revised manuscript and submitted a certificate of proofreading when we submitted the revised version.

Response to Reviewer 3

(Comments from Reviewer 3):

In this work, Hayashi and colleagues present an innovative strategy to enhance organic RTP: cooperative spin-orbit and transition dipoles effect. They point out that the horizontally introduced substituent strategies commonly used in fluorescent materials are not suitable for phosphorescence. By employing a unique Se-containing DBC molecular system, the authors found that multiple vertical hetero-substituents lead to more efficient phosphorescence than horizontal substitution. Theoretical calculations reveal that the reason lies in the larger transition dipole moment along the π plane enabled by vertical substitution, which facilitates the cooperative effect of a large transition dipole moment (TDM) and strong spin-orbit coupling (SOC). They further utilize their molecule in multi-label bioimaging to show its practical application. This work presents an effective and compelling strategy, making it well-suited for publication in Nat Commun. However, some minor revisions, particularly regarding theoretical analysis, should be addressed prior to acceptance.

(Response)

We sincerely appreciate your constructive comments. We have addressed your comments, as outlined in the following responses. Below, our response to the comments from Reviewer 3 is in blue, and the specific text and supporting information that we revised are in red.

(Comment 1 from Reviewer 3)

The author may consider add the term through-space substitution or vertical substitution in the title to emphasize its importance in bringing cooperative spin-orbit and transition dipole and enhancing RTP.

(Response)

Thank you for your important suggestion. We added “vertical substitution” in revised title to emphasize our organic phosphorescence design.

Before revision: Title in original manuscript

“Cooperative spin-orbit and transition dipoles for organic phosphorescence”

After revision: Title in revised manuscript

“Vertical substitution strategy to enable cooperation between spin–orbit coupling and transition dipoles for organic phosphorescence”

(Comment 2 from Reviewer 3)

On page 2, line 25, the authors claim that this molecular design can enhance the efficiency of organic phosphorescence to a level comparable to that of fluorescence across the entire visible spectrum. However, there is a lack of discussion regarding fluorescence enhancement in the article. Please clarify this point.

(Response)

Thank you for your comment. We included in the fluorescence yield (Φ_f) in original supplementary information. However, as Reviewer 3 pointed out, the fluorescence yield (Φ_f) was not mentioned in original manuscript. The Φ_f values of **1–5d** in amorphous β -estradiol were determined to be 20%, 1.1%, 1.7%, 1.2%, 1.0%, and 0.8%, respectively. Therefore, increasing number of horizontal hetero-substituents enhances Φ_f , whereas vertical hetero-substituents suppress Φ_f . The Φ_p of **1–5d** were 2.9%, 18%, 15%, 18%, 27%, and 35%, respectively. Therefore, the multiple introductions of horizontal hetero-substituents led to a reduction in Φ_p , while the Φ_p was improved by multiple introductions of vertical hetero-substituents.

Accordingly, we revised the sentences to include the Φ_f values and to clarify the different behaviours of Φ_f and Φ_p depending on the substitution pattern in the revised manuscript.

Before revision: Lines 11–13 in p. 5 in original manuscript

“Integration sphere measurements confirmed that the Φ_p values of **1–5d** were 2.9%, 18%, 15%, 18%, 27%, and 35%, respectively (supplementary Figs. S15 and S16).”

After revision: Lines 11–16 in p. 5 in revised manuscript

“Integration sphere measurements showed that the Φ_f values of **1–5d** were 20%, 1.1%, 1.7%, 1.2%, 1.0%, and 0.8%, respectively. The Φ_p values of **1–5d** were 2.9%, 18%, 15%, 18%, 27%, and 35%, respectively (supplementary Figs. S17–S19). These results confirm that increasing the number of horizontal hetero-substituents enhances Φ_f while reducing Φ_p , whereas Φ_p is selectively enhanced with increasing number of vertical hetero-substituents.”

(Comment 3 from Reviewer 3)

In Figure 2c, comparing RTP intensity through photoluminescence (PL) measurements for solid samples may be challenging due to multiple variables such as film inhomogeneity, doping concentration, and measurement distance. Therefore, it is recommended to supplement the figure with phosphorescence quantum yield values.

(Response)

Thank you for your comments and suggestions. The height of spectral intensity of Fig. 2c in original manuscript was set by taking Φ_p into consideration as follows. The recorded RTP spectra were normalized by the integrated spectral intensity at RTP region to remove the influence of absolute intensity variations such as measurement distance and/or absorbance of sample (Fig. R23a(i)). The normalized spectra were then multiplied by the Φ_p to rescale the height of spectral intensity, providing emission profiles whose integrated areas corresponding to the Φ_p (Fig. R23a(ii)). Although the height of spectral intensity in Fig. 2c was normalized by the peak intensity of **5d**, the spectral shape and relative RTP intensity were preserved. To examine the influence of the film inhomogeneity and the doping concentration, the Φ_p of amorphous film of 0.3 wt% **2**-doped β -estradiol was remeasured using different sample of 0.3 wt% **2**-doped β -estradiol. The value of Φ_p for 0.3 wt% **2**-doped amorphous β -estradiol film hardly changed among Lot. 1, 2, and 3 (Fig. R23b). Therefore, the value of Φ_p used in this paper has reproducibility.

Fig. R23 a, RTP spectra of 1–5d with a vertical axis of (i) RTP intensity normalized by integrated spectral intensity and (ii) RTP intensity adjusted by Φ_p . **b**, Photon intensity spectra measured in the integrating sphere to check the reproducibility of Φ_p of 2 in amorphous β -estradiol.

From suggestion of Reviewer 3, we have first added the Φ_p value to Fig. 2c in revised manuscript.

Before revision: Fig. 2c in original manuscript.

Caption of Figure 2c, Room-temperature phosphorescence (RTP) spectra soon after ceasing excitation.”

After revision: Fig. 2c in revised manuscript.

Caption of Figure 2c, Room-temperature phosphorescence (RTP) spectra soon after ceasing excitation.”

Additionally, we added Section 4-2 to revised supplementary information regarding the procedure for making Fig. 2c and reproducibility of Φ_p .

After revision: Section 4-2 in revised supplementary information.

“4-2. Procedure for making Fig. 2c and reproducibility of Φ_p (Figure S19)

The height of spectral intensity of Fig. 2c was set by taking Φ_p into consideration as follows. The recorded RTP spectra were normalized by the integrated spectral intensity at RTP region to remove the influence of absolute intensity variations such as the distance between sample and photodetector as well as the absorbance of sample (Fig. S19a(i)). The normalized spectra were then multiplied by the Φ_p to rescale the height of spectral intensity, providing emission profiles whose integrated areas corresponding to the Φ_p (Fig. S19a(ii)). Although the height of spectral intensity in Fig. 2c is normalized by the peak intensity of **5d**, the spectral shape and relative RTP intensity are preserved. To examine the influence of the film inhomogeneity and the doping concentration, the Φ_p of amorphous film of 0.3 wt% **2**-doped β -estradiol was remeasured using different sample of 0.3 wt% **2**-doped β -estradiol. The value of Φ_p for 0.3 wt% **2**-doped amorphous β -estradiol film hardly changed among Lot. 1, 2, and 3 (Fig. S19b).

Fig. S19. Procedure for making Fig. 2c and reproducibility of Φ_p . **a**, RTP spectra of **1–5d** in amorphous β -estradiol with a vertical axis of (i) RTP spectral intensity normalized by the integrated spectral intensity and (ii) RTP spectral intensity adjusted by Φ_p . **b**, Photon intensity spectra measured in the integrating sphere equipment with (light blue) and without (black) **2**-doped amorphous β -estradiol film.”

(Comment 4 from Reviewer 3)

Although the authors cite some references in the Supporting Information (Section 5-3), which empirically illustrate that $k_r T$ is related to TDM(Sn-S0) and SOC(Sn-T1), the structure-property relationship remains unclear. Why the transition dipole moment of Sn and the SOC between Sn and T1 will affect the rate constant of T1? Could the authors provide an explanation from the electronic effect perspective instead of an empirical rule? (Response)

Thank you for your comment. The rate constant of the radiative transition from T₁ (k_r^T) based on the first order perturbation is expressed as (*Chem. Rev.* **66**, 199–241 (1966); *J. Phys. Chem. A* **111**, 10490–10499 (2007); *Chem. Rev.* **117**, 6500–6537 (2017)),

$$k_r^T = \frac{E_{T_1-S_0}^3}{3\epsilon_0\pi\hbar^4c^3} |\mu_{T_1-S_0}|^2, \quad (R3-1)$$

$$\mu_{T_1-S_0} = \sum_{n \geq 1} \frac{\langle S_n | H_{SO} | T_1 \rangle}{E_{S_n-T_1}} \mu_{S_n-S_0} + \sum_{m \geq 2} \frac{\langle T_m | H_{SO} | S_0 \rangle}{E_{T_m-S_0}} \mu_{T_m-T_1}, \quad (R3-2)$$

where $E_{T_1-S_0}$ is the energy gap between T_1 and S_0 , ϵ_0 is the vacuum permittivity, \hbar is the Dirac constant, c is the speed of light, $\mu_{T_1-S_0}$ is the transition dipole moment (TDM) between T_1 and S_0 , $\langle S_n | \mathbf{H}_{SO} | T_1 \rangle$ is the spin-orbit coupling (SOC) between the n th-order singlet excited state (S_n , $n \geq 1$) and T_1 , $E_{S_n-T_1}$ is the energy gap between S_n and T_1 , $\mu_{S_n-S_0}$ is the TDM between S_n and S_0 , $\langle T_m | \mathbf{H}_{SO} | S_0 \rangle$ is the SOC between the m th-order triplet excited state (T_m , $m \geq 2$) and S_0 , $E_{T_m-S_0}$ is the energy gap between T_m and S_0 , $\mu_{T_m-T_1}$ is the TDM between T_m and T_1 . These parameters are visualized in Fig. R24. The $\mu_{T_1-S_0}$ is strictly zero in the absence of perturbations because T_1-S_0 transition is spin-forbidden. However, when SOC is introduced, $\mu_{T_1-S_0}$ becomes finite value through intensity borrowing from spin-allowed transitions. Specifically, the $\frac{\langle S_n | \mathbf{H}_{SO} | T_1 \rangle}{E_{S_n-T_1}}$ site in first term of equation (R3-2) represents the probability for mixing the spin-allowed TDM $\mu_{S_n-S_0}$ into the T_1-S_0 transition. Similarly, the $\frac{\langle T_m | \mathbf{H}_{SO} | S_0 \rangle}{E_{T_m-S_0}}$ site in second term of equation (R3-2) represents the probability for mixing the spin-allowed TDM $\mu_{T_m-T_1}$ into the T_1-S_0 transition.

Fig. R24 Schematic energy diagram illustrating the key factors contributing to k_r^T .

Accordingly, we have added the equation of k_r^T in revised manuscript.

After revision: Lines 9–20 in p. 9 in revised manuscript

“The equation for k_r^T based on first-order perturbation theory is expressed as^{6,29,30}

$$k_r^T = \frac{E_{T_1-S_0}^3}{3\varepsilon_0\pi\hbar^4c^3} |\mu_{T_1-S_0}|^2, \quad (3)$$

$$\mu_{T_1-S_0} = \sum_{n \geq 1} \frac{\langle S_n | \mathbf{H}_{SO} | T_1 \rangle}{E_{S_n-T_1}} \mu_{S_n-S_0} + \sum_{m \geq 2} \frac{\langle T_m | \mathbf{H}_{SO} | S_0 \rangle}{E_{T_m-S_0}} \mu_{T_m-T_1}, \quad (4)$$

where $E_{T_1-S_0}$ is the energy gap between T_1 and S_0 , ε_0 is the vacuum permittivity, \hbar is the Dirac constant, c is the speed of light, $\mu_{T_1-S_0}$ is the TDM between T_1 and S_0 , $\langle S_n | \mathbf{H}_{SO} | T_1 \rangle$ is the SOC between the n th-order singlet excited state (S_n , $n \geq 1$) and T_1 , $E_{S_n-T_1}$ is the energy gap between S_n and T_1 , $\mu_{S_n-S_0}$ is the TDM between S_n and S_0 , $\langle T_m | \mathbf{H}_{SO} | S_0 \rangle$ is the SOC between the m th-order triplet excited state (T_m , $m \geq 2$) and S_0 , $E_{T_m-S_0}$ is the energy gap between T_m and S_0 , and $\mu_{T_m-T_1}$ is the TDM between T_m and T_1 .”

(Comment 5 from Reviewer 3)

In Figure 3b, panels iii and iv show overlap between ΩS_n and ΩT_1 , but only ΩS_n (red and blue) is clearly visible, while ΩT_1 (orange and blue) is difficult to distinguish. Please clarify this more explicitly and include additional representations of ΩT_1 in the Supporting Information. Moreover, for consistency, it is suggested that overlap density be used uniformly in panels i–iv, rather than switching between HOMO/LUMO and ΩS_n representations when comparing compounds 3 and 5.

(Response)

In the revised manuscript, we have introduced a figure corresponding to the content of Figure 3b in Figure 5b. For **3** in Figure 5b, the overlap density between the HOMO-2 and LUMO involved in the S_5-S_0 transition is shown in red. The overlap density between the HOMO and LUMO involved in the T_1-S_0 transition is shown in green. This overlap between the red and green colors can be well seen in the open green circle. For **5** in Figure 5b, the overlap density between the HOMO-1 and LUMO involved in the S_2-S_0 transition is shown in red. The overlap density between the HOMO and LUMO involved in the T_1-S_0 transition is shown in green. This overlap between the red and green colors can be seen in the open green circle. Following Reviewer 3's suggestion, we have discontinued the symbolic representation of the overlap density and instead used the HOMO and LUMO notations to show the orbital overlap. In Figure 5b, to compare the difference in the effect

of orbital overlap on $\langle S_n | H_{SO} | T_1 \rangle^2$ between **3** and **5**, the isovalues for **3** and **5** have been made consistent.

Before revision: Fig. 3 in original manuscript

Caption of Figure 3b, Molecular orbitals and their calculated values relating (i) and (ii) $\langle T_1 | H_{SO} | S_0 \rangle^2$, (iii) and (iv) $\langle S_n | H_{SO} | T_1 \rangle^2$, and (v) and (vi) $\mu_{S_n-S_0}^2$. In (i) and (ii), the red-blue orbitals and orange-light blue orbitals are the highest occupied molecular orbital (HOMO) and lowest unoccupied molecular orbital (LUMO), respectively. In (iii) and (iv), the red-blue orbitals are Ω_{S_5} or Ω_{S_3} and the orange-light blue orbitals are Ω_{T_1} . In (i)–(iv), the white arrows represent the orbital angular momentum (OAM) vectors. The isovalues of the HOMO of **3**, LUMO of **3**, and HOMO of **5** are 0.02. The isovalue of 0.0085 is used for the LUMO of **5**. The isovalues of Ω_{S_3} , Ω_{S_5} , and Ω_{T_1} are 0.00016.”

After revision: Fig. 5 in revised manuscript

Fig. 5 Difference in the factors governing $\langle S_n | H_{SO} | T_1 \rangle^2$ and $\langle T_m | H_{SO} | S_0 \rangle^2$ between horizontal and vertical hetero-substituents. **a**, Simplified expression of the k_r^T equation. **b**, Molecular orbitals and schematic illustrations showing that $\langle S_5 | H_{SO} | T_1 \rangle^2$ of **3** (top) and $\langle S_2 | H_{SO} | T_1 \rangle^2$ of **5** (bottom) have comparable magnitudes. The isovalues of the overlap density regarding (HOMO-2)×(LUMO) and (HOMO)×(LUMO) for **3** and (HOMO-1)×(LUMO) and (HOMO)×(LUMO) for **5** are 0.00011. **c**, Molecular orbitals and schematic illustrations explaining the comparable magnitudes of $\langle T_7 | H_{SO} | S_0 \rangle^2$ for **3** (top) and $\langle T_9 | H_{SO} | S_0 \rangle^2$ for **5** (bottom). The isovalue of 0.013 is used for HOMO-3 and LUMO of **3** and HOMO-2 and LUMO of **5**. In **b** and **c**, panels (i), (ii), and (iii) represent the orbital overlap, HAE, and orbital orientation effect, respectively. The white arrows in (iii) represent the orbital axis orientations.”

Because Reviewer 1 requested some considerations related to k_r^T in higher triplet states, we have included that information in Figure 5c.

(Comment 6 from Reviewer 3)

Multi-label imaging is an interesting application. Have the authors conducted cytotoxicity tests to demonstrate the biocompatibility of the material?

(Response)

Thank you for your comment. To evaluate the biocompatibility of the material, changes in cellular morphology were monitored by transmittance imaging after the injection of 0.3 wt% **5d**-doped crystalline benzophenone particles. Compared to the control condition in the absence of the particles (Fig. R25(i)), the morphology of the cell membrane remained essentially unchanged throughout the observation period in the presence of the particles (Fig. R25(ii)). Meanwhile, the afterglow images confirmed the presence of the particles (Fig. R25(iii)), indicating that the particles did not induce any noticeable cytotoxic effects under the experimental conditions.

Fig. R25 (i,ii) Transmittance images of HEK293 cell culture sample in the absence (i) and the presence (ii) of 0.3 wt% **5d**-doped crystalline benzophenone particles, and (iii) afterglow images captured after ceasing excitation in the presence of the crystalline particles.

Accordingly, we added the sentence regarding changes in cell morphology over the time.

After revision: Lines 5–7 in p. 14 in revised manuscript

“and the morphology of the cell membrane remained essentially unchanged for at least 90 min in the presence of the particles (supplementary Fig. S37).”

Additionally, we added Section 6-2 to revised supplementary information regarding the biocompatibility of the nanocrystals.

After revision: Section 6-2 in revised supplementary information.

“6-2. Biocompatibility of the water-dispersible nanocrystal (Figure S37)

To evaluate the biocompatibility of the water-dispersible crystalline particle, changes in cellular morphology were monitored by transmittance imaging after the injection of 0.3 wt% **5d**-doped crystalline benzophenone particles. Compared to the control condition in the absence of the particles (Fig. S37(i)), the morphology of the cell membrane remained essentially unchanged throughout the observation period of at least 90 min in the presence of the particles (Fig. S37(ii)). Meanwhile, the afterglow images confirmed the presence of the particles (Fig. S37(iii)), indicating that the particles did not induce any noticeable cytotoxic effects under the experimental conditions.

Fig. S37. (i,ii) Transmittance images of HEK293 cell culture sample in the absence (i) and the presence (ii) of 0.3 wt% **5d**-doped crystalline benzophenone particles, and (iii) afterglow images captured after ceasing excitation in the presence of the crystalline particles.”

(Comment 7 from Reviewer 3)

An efficient RTP system achieved through similar through-space substitution of main-group elements was recently reported (10.1021/jacs.4c17142), which should be cited. In addition, other recently developed high-performance red RTP systems (e.g., 10.1002/agt2.70124, 10.1038/s41467-025-61714-0) should also be referenced in the Introduction.

(Response)

Thank you for your suggestion. We have added the paper describing red afterglow RTP (10.1038/s41467-025-61714-0) to the revised manuscript in ref 26. However, Reviewer 3's suggested paper (10.1021/jacs.4c17142) does not include the word for through-space interaction. Therefore, we consider that the paper is not directly relevant to this work. Instead, we cite a recent report on organic phosphorescent materials that focuses on multiple through-space interactions (*J. Am. Chem. Soc.* **147**, 10803–10814 (2025)). In this report, through-space interactions have been discussed as a factor relating to the rigidity and Φ_{isc} , where Φ_{isc} has been discussed only in terms of calculated values of $\langle S_1 | \mathbf{H}_{SO} | T_m \rangle^2$. On the other hand, this study demonstrates that vertically introduced through-space substituents selectively enhance $\langle S_n | \mathbf{H}_{SO} | T_1 \rangle^2$ and $\langle T_m | \mathbf{H}_{SO} | S_0 \rangle^2$, where they are not directly related to the $\langle S_1 | \mathbf{H}_{SO} | T_m \rangle^2 \propto \Phi_{isc}$. Additionally, this study founded that increasing the number of vertical hetero-substituents generates the strong cooperativity between $\langle S_n | \mathbf{H}_{SO} | T_1 \rangle^2$ and TDM as well as $\langle T_m | \mathbf{H}_{SO} | S_0 \rangle^2$ and TDM for enhanced phosphorescence.

Accordingly, we have added the sentences regarding differences in the our approach and the previous through-space approaches in the revised manuscript

After revision: From line 23 in p. 14 to line 6 in p. 15 in revised manuscript

“So far, through-space interactions have been discussed as a factor related to the rigidity and Φ_{isc} ^{36,37}, where Φ_{isc} has been discussed only in terms of calculated values of $\langle S_1 | \mathbf{H}_{SO} | T_m \rangle^2$. In contrast, this study demonstrates that vertically introduced through-space substituents selectively enhance $\langle S_n | \mathbf{H}_{SO} | T_1 \rangle^2$ and $\langle T_m | \mathbf{H}_{SO} | S_0 \rangle^2$, and they are not directly related to $\langle S_1 | \mathbf{H}_{SO} | T_m \rangle^2 \propto \Phi_{isc}$. Additionally, increasing the number of vertical hetero-substituents generates strong cooperativity between $\langle S_n | \mathbf{H}_{SO} | T_1 \rangle^2$ and the TDM, as well as between $\langle T_m | \mathbf{H}_{SO} | S_0 \rangle^2$ and the TDM, for enhanced phosphorescence.”